# Metabolic targeting of cancer associated fibroblasts overcomes T-cell exclusion and chemoresistance in soft-tissue sarcomas

Marina T. Broz[1], Emily Y. Ko [2], Kristin Ishaya[2], Jinfen Xiao [2], Marco De Simone[2], Xen Ping Hoi [1], Roberta Piras [1], Basia Gala[1], Fernando H. G. Tessaro[2], Anja Karlstaedt[1,3,4], Sandra Orsulic [4,5], Amanda W. Lund [6], Keith Syson Chan[7] & Jlenia Guarnerio [1,2,4,8] ✉

T cell-based immunotherapies have exhibited promising outcomes in tumor control; however, their efficacy is limited in immune-excluded tumors. Cancer-associated fibroblasts (CAFs) play a pivotal role in shaping the tumor microenvironment and modulating immune infiltration. Despite the identification of distinct CAF subtypes using single-cell RNA-sequencing (scRNA-seq), their functional impact on hindering T-cell infiltration remains unclear, particularly in soft-tissue sarcomas (STS) characterized by low response rates to T cell-based therapies. In this study, we characterize the STS microenvironment using murine models (in female mice) with distinct immune composition by scRNA-seq, and identify a subset of CAFs we termed glycolytic cancer-associated fibroblasts (glyCAF). GlyCAF rely on GLUT1-dependent expression of CXCL16 to impede cytotoxic T-cell infiltration into the tumor parenchyma. Targeting glycolysis decreases T-cell restrictive glyCAF accumulation at the tumor margin, thereby enhancing T-cell infiltration and augmenting the efficacy of chemotherapy. These findings highlight avenues for combinatorial therapeutic interventions in sarcomas and possibly other solid tumors. Further investigations and clinical trials are needed to validate these potential strategies and translate them into clinical practice.

In recent years, T cell-based immunotherapies, including anti-PD-1/PD-L1 agents and CAR-T cell therapies, have shown considerable success in controlling tumor growth and inducing its regression across various malignancies[1,2]. However, their efficacy is limited in immune-excluded tumors lacking substantial T-cell infiltration[3]. Therefore, understanding the mechanisms governing T-cell recruitment and devising strategies to enhance their infiltration into the tumor parenchyma—

where direct interactions with malignant cells provoke potent anti-tumor immune responses—is of utmost importance.

Cancer-associated fibroblasts (CAFs) have been recognized as pivotal contributors to the physical blockade of T-cell infiltration by creating a dense extracellular matrix and forging an immunosuppressive tumor microenvironment through the secretion of factors such as CXCL12 and TGFβ[4–7]. Because of these and other functions,

[1]Department of Biomedical Sciences, Cedars-Sinai Medical Center, Los Angeles, CA, USA. [2]Department of Radiation Oncology, Cedars-Sinai Medical Center, Los Angeles, CA, USA. [3]Department of Cardiology, Smidt Heart Institute, Cedars-Sinai Medical Center, Los Angeles, CA, USA. [4]David Geffen Medical School, Department of Medicine, University of California, Los Angeles, CA, USA. [5]Department of Obstetrics and Gynecology, David Geffen School of Medicine, University of California, Los Angeles, CA, USA. [6]Ronald O. Perelman Department of Dermatology, NYU Grossman School of Medicine, New York, NY, USA. [7]Department of Urology, Neal Cancer Center, Houston Methodist Research Institute, Houston, TX, USA. [8]Board of Governors Regenerative Medicine Institute, Cedars-Sinai Medical Center, Los Angeles, CA, USA. ✉e-mail: Jlenia.guarnerio@cshs.org

CAFs have been historically labeled as tumor-promoting cells; however, therapeutic targeting of the αSMA+ CAFs has not demonstrated significant benefits in inhibiting tumor growth[8,9], while FAP targeting strategies such as the use of CAR-T have been inconclusive[10–14]. Recent advancements, such as single-cell RNA-sequencing (scRNA-seq), have enabled in-depth characterization of the CAFs in tumors with different histology, revealing the existence of distinct CAF subtypes[8,15,16]. These subtypes, including myofibroblastic CAF (myCAF), inflammatory CAF (iCAF), and antigen-presenting CAF (apCAF)[17], exhibit unique transcriptional profiles, functional activities, and spatial localization within the tumor mass[5,18,19]. For example, iCAFs predominantly express inflammatory cytokines (e.g., IL-6, LIF, CXCL12), while myCAFs are associated with focal adhesion and extracellular matrix interactions[9,19], and apCAFs exhibit enrichment in antigen processing and presentation pathways[18]. Despite these findings, the specific CAF subtype responsible for limiting T-cell infiltration into the tumor parenchyma, thus compromising the effectiveness of T cell-based therapies, remains unclear. Targeted interventions that selectively inhibit tumor-promoting and T-cell excluding CAFs or strategies aimed at CAF reprogramming hold promise for enhancing the efficacy of T cell-based immunotherapies[20].

While these investigations may benefit multiple tumor types, they are especially important in the context of soft-tissue sarcoma (STS), which are tumors of mesenchymal origin, and remain understudied in terms of microenvironment composition. First-line treatment options for STS are radiotherapy and surgery, which result in local tumor recurrence in roughly 20% of patients within 5 years[21,22]. Additionally, ~50% of STS patients also develop aggressive metastatic disease following surgery, most frequently to the lung, leading to a 5-year survival rate of 20%[23]. Whether anthracycline based chemotherapy as neo-adjuvant/adjuvant therapies can reduce recurrence and metastasis in high-grade sarcomas has shown mixed results; moreover, difficulties identifying high grade tumors at biopsy and heterogeneity among STS tumors have limited the conclusions of these studies[24,25]. The response rate to existing T cell-based immunotherapies such as anti-PD-1 is limited with response rates observed in only ~15% of patients[26,27], likely because many STS present extensive areas that are T-cell excluded. Thus, understanding the mechanisms of T-cell exclusion for these tumors, including the role of CAFs in this respect, becomes crucial. Yet, these investigation have been relatively limited and lacks a comprehensive characterization of the CAFs associated with STS and their functional roles[28].

In this study, we employ immune-competent murine models replicating genetic alterations observed in STS patients, as well as the immune tumor microenvironments of immune-excluded and immune-infiltrated tumors. Through scRNA-seq characterization of fibroblasts associated with sarcomas in mice and humans, we delineate the expression programs of CAFs, including the mechanisms employed by CAFs to hinder T-cell infiltration into the tumor parenchyma. Functional studies in vitro and in vivo identify a specific subset of CAFs termed glycolytic cancer-associated fibroblasts (glyCAF), which rely on glycolysis to impede cytotoxic T-cell entry into the tumor mass via the *Cxcl16/Cxcr6* axis. Importantly, inhibiting glycolytic properties of glyCAF increases infiltration of cytotoxic CD8+ T cells into the tumor parenchyma and improves chemotherapy efficacy. These findings provide the foundation for the development of combinatorial therapeutic interventions for sarcomas and, possibly, other solid tumors to be evaluated in future clinical trials.

## Results

### Sarcoma mouse models recapitulate immune- excluded and infiltrated TME

The amplification or over-expression of the Cyclin E1 (*CCNE1*) and the Vestigial Like Growth Factor 3 (*VGLL3*), in addition to the functional loss of *TRP53*, are among the most common genetic alterations found in STS patients[29]. GSEA analysis of *CCNE1*-high (z-score > 1.5 vs. unaltered) patients show upregulation of pathways related to cell cycle and DNA replication; a signature typically associated with tumors poorly infiltrated by T cells[30] (Fig. 1a). On the contrary, *VGLL3*-high tumors are enriched in genes related to immune responses and inflammation (Fig. 1a). These STS human data provided the rationale for us to determine if these two oncogenes could drive the formation of sarcoma in mice with "cold" and "hot" tumor immune microenvironments, as suggested from the analysis of STS patients[27], and if so, whether we could use these models to define the molecular mechanisms behind immune exclusion. Accordingly, we employed a previously established protocol to generate immunocompetent STS mouse models[31] and FACS-sorted non-malignant mesenchymal stromal cells from the bone marrow of p53-null mice (*p53*^KO) for genetic manipulation. After a brief in vitro expansion at 1% oxygen, MSCs were transformed into sarcoma cells via overexpression of *Ccne1* or *Vgll3*, using viral vectors which also encode a red fluorescent protein (dsRED) reporter for tracing (Supplementary Fig. 1a, b). The resulting p53^KO*Ccne1*+ and p53^KO*Vgll3*+ cells expressed increased levels of *Ccne1* (~5-fold) and *Vgll3* (~4-fold) compared to the p53^KO cells transduced with an empty control vector (Supplementary Fig. 1c). The overexpression of *Ccne1* or *Vgll3* was sufficient to transform the cells, which exhibit morphological changes in vitro (Supplementary Fig. 1d) and, in the case of p53^KO*Ccne1*+ cells, showed higher proliferation rates compared to the p53^KO cells (Supplementary Fig. 1e). To model tumorigenesis in vivo, p53^KO*Ccne1*+ or p53^KO*Vgll3*+ cells were seeded onto 3D polyurethane scaffolds and implanted subcutaneously in syngeneic mice for two subsequent generations to select for clones with tumor generating potential (Fig. 1b). The secondary recipients generated tumors within 1 month with 100% penetrance for both the genetics analyzed. Pathological analysis of the tumors showed a similar histology between Ccne1+ and Vgll3+ genetics, consistent with High Grade Undifferentiated Sarcoma with Spindle Cell morphology, which recapitulates the histological features of human Undifferentiated Pleiomorphic Sarcoma (UPS)[32] (Supplementary Fig. 1f). The differences of in vitro proliferation were also maintained in vivo; mice bearing scaffolds with p53^KO*Ccne1*+ cells exhibited a 34% greater tumor mass at endpoint (0.2080 ± 0.09167 g, *p* = 0.0530), compared to tumors generated by p53^KO*Vgll3*+ cells (Supplementary Fig. 1g).

We first characterized the immune infiltration of p53^KO*Ccne1*+ and p53^KO*Vgll3*+ (herein referred to as Ccne1+ and Vgll3+) tumors by flow cytometry. In line with gene expression analysis from *VGLL3*-high patient tumors exhibiting upregulated inflammatory pathways, Vgll3+ murine tumors are infiltrated with overall higher amounts of CD45+ immune cells, including increased relative levels CD4+ and CD8+ T cells, macrophages, and NK cells compared to Ccne1+ tumors that are overall poorly infiltrated by immune cells (Fig. 1c and Supplementary Fig. 1h). By examining spatial distribution of the T cells in these two models, we observed that both CD8+ and CD4+ T cells are significantly more abundant in the inner parenchyma of Vgll3+ tumors than in Ccne1+ tumors, in which CD8+ T cells are instead especially restricted to the tumor margin (Fig. 1d–f). As immune-infiltrated tumors are associated with better responses to chemotherapy[33,34] we next measured the responsiveness to doxorubicin (DOX) of the Ccne1+ and Vgll3+ tumors. Accordingly, the immune-infiltrated Vgll3+ model showed a more robust response to the standard regimen of doxorubicin compared to the immune-excluded Ccne1+ model (Fig. 1g, h), suggesting that an inflamed sarcoma microenvironment may be favorable for chemotherapy responses. These data indicate that Ccne1+ and Vgll3+ STS models recapitulate two distinct versions of the immune TME as found in the patients ("immune-excluded" vs. "immune-infiltrated") and serve as unique tools for investigating molecular mechanisms behind immune exclusion and chemoresistance in sarcoma.

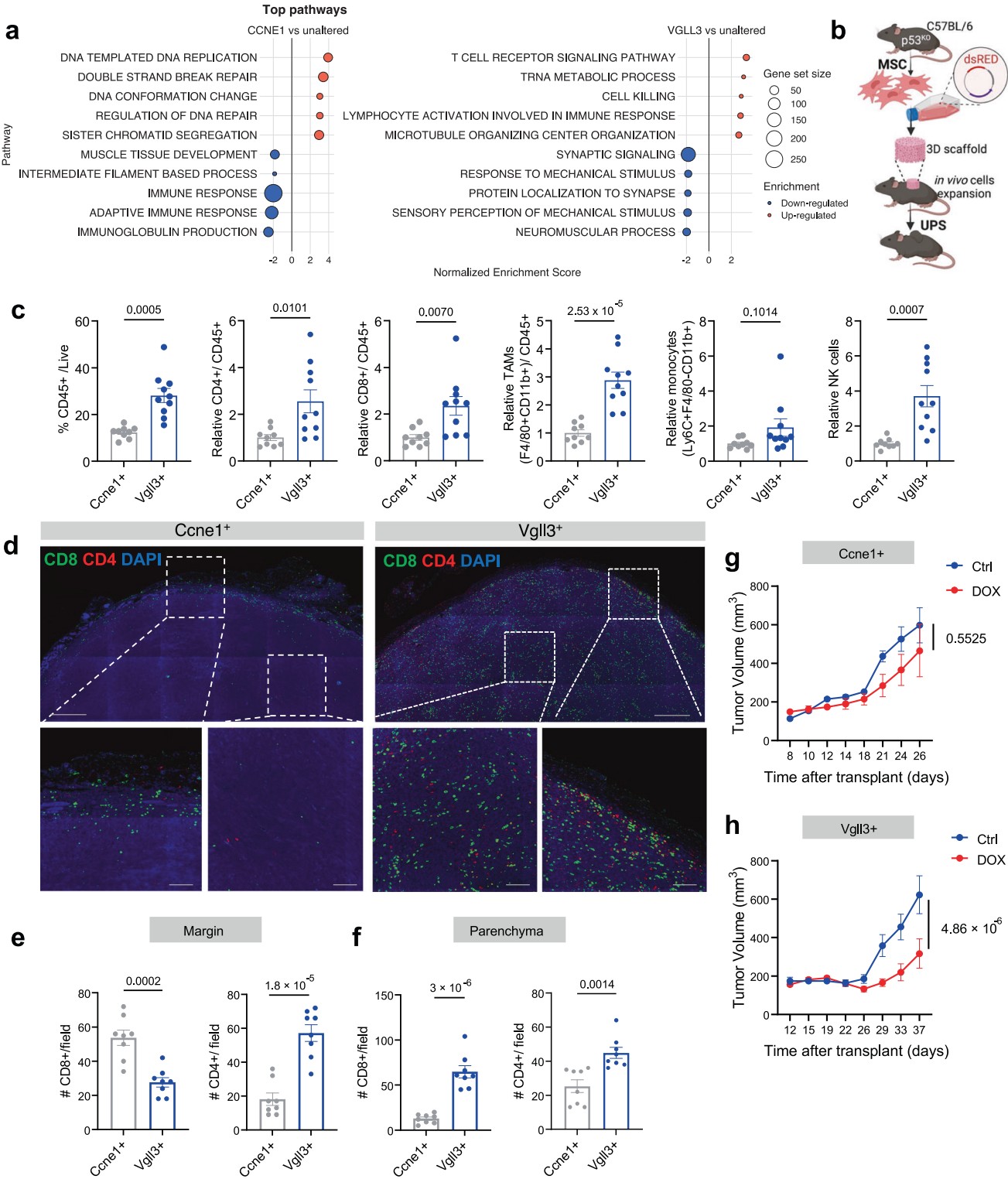

## Cancer-Associated Fibroblasts (CAFs) populate mouse and human sarcoma

Cancer associated fibroblasts (CAF) are the prevailing stromal cell in carcinomas[35,36] and CAF-abundant tumors are often T-cell excluded[37,38]. Accordingly, we hypothesized that different CAF content in Ccne1+ and Vgll3+ tumors is responsible for their distinct immune TME, and especially for blocking T cells at the tumor border in the Ccne1+

tumors. The shared mesenchymal origin of sarcoma cells and CAFs has hindered the ability to study the contribution of CAF to sarcoma growth and immune exclusion. Therefore, we first used a reporter to distinguish sarcoma cells from the TME cells and profiled 12 independent Ccne1+ tumors by scRNA-seq to identify specific CAF markers (Fig. 2a). Using the reads mapping to the transcript encoding for the reporter (*dsRED*) we selectively identified the tumor cells (Fig. 2a and

**Fig. 1 | Sarcoma mouse models recapitulate immune-excluded and -infiltrated TME. a** GSEA Pathway enrichment analysis of RNA-seq data from 251 sarcoma patient samples which were stratified based on *CCNE1*-high vs. unaltered expression ($n = 16$; 235) or *VGLL3*-high vs. unaltered expression ($n = 8$; 243). **b** Overview of platform for modeling tumorigenesis utilizing murine mesenchymal stem cells (MSCs) isolated from p53[KO] mice. **c** Flow cytometry analysis of the proportion of infiltrating CD45$^+$ immune cells of total cells. Relative proportions of tumor infiltrating immune cells: CD4$^+$ and CD8$^+$ T cells, tumor associated macrophage (TAM), monocytes, and NK cells ($n = 9$ Ccne1$^+$; $n = 10$ Vgll3$^+$ mice). **d** Immunofluorescence staining of CD8$^+$ T cells (green) and CD4$^+$ T cells (red) (Scale bars 500 µm (top) and 100 µm (bottom)). Images are representative of three independent experiments with $n = 4$ mice. **e, f** Quantifications of CD8$^+$ and CD4$^+$ cells/field in ROIs encompassing the tumor margin or the tumor parenchyma ($n = 8$ ROIs from $n = 3$ mice). **g, h** Tumor growth following treatment with doxorubicin (DOX, 6 mg/kg) in Ccne1$^+$ and Vgll3$^+$ tumor bearing mice. Two-way ANOVA with Tukey's multiple comparison was used to determine differences between groups ($n = 5$ mice). Unless otherwise indicated, results are presented as mean ± SEM and $p$-values are derived by a two-tailed unpaired Student's $t$ test. Source data are provided as a Source Data file.

Supplementary Fig. 2a), which we found to express a mesenchymal signature of transcripts encoding for extracellular matrix genes and collagens (*Col1a1, Col1a2, Cald1, Calu*). In addition to the tumor cells (*dsRED*+), we also found another cell cluster lacking the expression of *dsRED* that similarly express mesenchymal genes, but which was enriched for additional genes previously associated with CAFs, such as *Rarres2, Pi16, Fap, Lum, Sfrp2, Dpt*, and *Col14a1* (Fig. 2a, b)[5,35]. Additionally, this cluster expressed *Thy1*, encoding the surface protein CD90 (Fig. 2a, b and Supplementary Fig. 2a), which aligns with previous reports of CD90 as a CAF marker in pancreatic and breast tumors[39,40]; accordingly, we observed that 87.9% of cells in the cluster annotated as CAF express *Thy1*, thus providing us with the rationale to adopt CD90 as a CAF enrichment marker for sarcoma for further CAF characterization and functional studies. Accordingly, histological analysis in murine sarcomas illustrates CD3$^-$ CD90$^+$ CAFs localized at the tumor margin and, to a lesser extent, diffusely throughout the tumors (Fig. 2c).

Having distinguished the transcriptomic profile of CAFs versus sarcoma cells in mouse, we next sought to use this same gene signature to identify CAFs and cancer cells in human sarcoma samples. We performed single-cell RNA-seq of four human sarcomas: two primary Leiomyosarcomas (LMS) samples (Fig. 2d and Supplementary Fig. 2b, c), which share genetic similarities with UPS[29], one metastatic high grade Leiomyosarcoma (met-LMS) (Fig. 2d and Supplementary Fig. 2d), and one Myxofibrosarcoma (MFS) (Fig. 2d and Supplementary Fig. 2e), which shares pathologic similarities with UPS[41]. Cells from each sample were computationally clustered and the gene expression profile of all clusters was analyzed. In each patient sample, common microenvironment cell types were identified, including immune cells (*PTPRC, ITGAM, S100A8, CD163, NKG7, CD79A*), endothelial cells (*PECAM1, CDH5*) and pericytes (*RGS5, NOTCH3*). To label the remaining cells, which we reasoned were either sarcoma cells or fibroblasts based on expression of mesenchymal markers such as *COL1A1, COL1A2, CALD1, CALU*, and negativity for other TME markers, we scored the cells for average expression of the top 30 genes that were upregulated in mouse CAFs compared to sarcoma cells (Fig. 2b). Interestingly, the mouse-derived CAF gene signature was expressed at high levels in only a subset of the remaining mesenchymal cell clusters, which we reasoned were CAFs (Fig. 2d, e). In addition, these same cells were relatively enriched for a 25-gene universal fibroblast signature previously reported by Buechler and colleagues[42] (Fig. 2e), further suggesting CAF identity. To test whether the remaining cells could be sarcoma cells, we next scored all mesenchymal cells from primary LMS tumors for the expression of an LMS signature previously published by Lee and colleagues[43]. Indeed, within the mesenchymal compartment, the LMS signature was most highly expressed in the non-CAF cell clusters (Fig. 2f), and expression of the LMS and CAF gene signatures was modestly negatively correlated (Spearman's rho = −0.33 (LMS-A32) and −0.42 (LMS-C10); $p < 2.2e-16$). The CAF gene signature thus suggested a criterion that could distinguish CAFs and sarcoma cells, despite shared mesenchymal lineage. Histological analysis of human sarcomas also captured the presence of CD3$^-$ CD90$^+$ CAF with spindle shape morphology (Fig. 2g) in the tumor mass, like those populating the mouse models (Fig. 2c).

## Glycolytic CAF (glyCAF) are the dominant CAF subtype in immune-excluded tumors

After having defined CAF-specific markers, we investigated whether immune-excluded and -infiltrated tumors have a proportionally different CAF content. When we compared the total number of CAFs we did not find a significant difference ($p = 0.380$) between the Ccne1$^+$ (mean = 45,027; SD = 9725) and Vgll3$^+$ (mean = 68,194 mean; SD = 48,912) models (Fig. 3a) and the thickness of the stromal barrier quantified by the average distance of the CD90$^+$ fibrotic stroma from the tumor invasive margin (Fig. 3b) was similar across the two models, suggesting that the number of fibrotic cells or thickness of the stromal barrier does not explain the differential distribution of CD8$^+$ T cells between the Ccne1$^+$ and Vgll3$^+$ models. Next, we sought to characterize the CAF composition in immune-infiltrated and immune-excluded tumors at the transcriptional level. Single cell RNA-sequencing has revealed heterogeneity in the expression profile of CAFs and shown that transcriptionally different sub-types of CAFs can play distinct pro- or anti-tumor roles in the tumor microenvironment[15,16,18,44]. Whether differences in the CAF composition underlie the differences in immune infiltration has been only marginally investigated[45,46]. To further investigate this possibility, we performed a second experiment to enrich for the CAFs in the Ccne1$^+$ immune-excluded model by FACS-sorting dsRED$^-$ CD45$^-$ CD31$^-$ CD90$^+$ (non-tumor, non-immune, non-endothelial) cells and analyzing the purified CAFs by scRNA-seq (Fig. 3c, d and Supplementary Fig. 3a). As a quality check of the enriched population, we assessed that the isolated cells lacked expression of tumor (*dsRED*), immune (*Ptprc*) and endothelial markers (*Pecam1*) and expressed the mouse CAF-specific signature we previously identified (e.g. *Rarres2, Fap, Lum, Thy1*) (Fig. 3e). We identified 4 subclusters of CAFs: inflammatory CAF (iCAF), matrix CAF (mCAF), antigen-presenting CAF (apCAF), and glycolytic CAF (glyCAF) (Fig. 3d−f and Supplementary Fig. 3b). Inflammatory (iCAF) were identified based on their expression of *Ly6c1, Clec3b, Cd34, Dpt*[18]. Matrix CAF (mCAF) were identified by their expression of genes involved in extracellular matrix components and collagens (*Col12a1, Tnc, Thbs2, Postn*)[47]. We identified a cluster of antigen-presenting CAF (apCAF) enriched in genes involved in MHC-class II antigen presentation (*Cd74, H2-Ab1, H2-Eb1*), which has been previously reported in mouse models of PDAC[18]. Lastly, we identified a cluster of glycolytic CAF (glyCAF) expressing genes related to glucose metabolism (*Slc2a1, Pgk1, Pkm, Pgam1, Hk2*). The glyCAF cluster also specifically expressed *Nt5e*, encoding the surface protein CD73, which has been reported to be upregulated during Warburg metabolism and hypoxia[48] (Fig. 3f). KEGG pathway analysis corroborated the observation that glyCAF are enriched in genes involved in glycolysis and HIF-1 signaling (Fig. 3g). Interestingly, glyCAF express genes associated with the contractile phenotype of myofibroblastic CAF (myCAF) identified in carcinomas (*Acta2, Tagln, Myl9, Tpm1, Tpm2*) (Supplementary Fig. 3c), suggesting that glyCAF share myCAF features, but ultimately represent a specific metabolic cell state that is distinct from the myofibroblastic CAF previously described in carcinomas[15,18].

To validate the scRNA-seq results and gain functional data, we performed further experiments aimed at characterizing the glyCAF in the immune-excluded setting. By measuring the expression of the

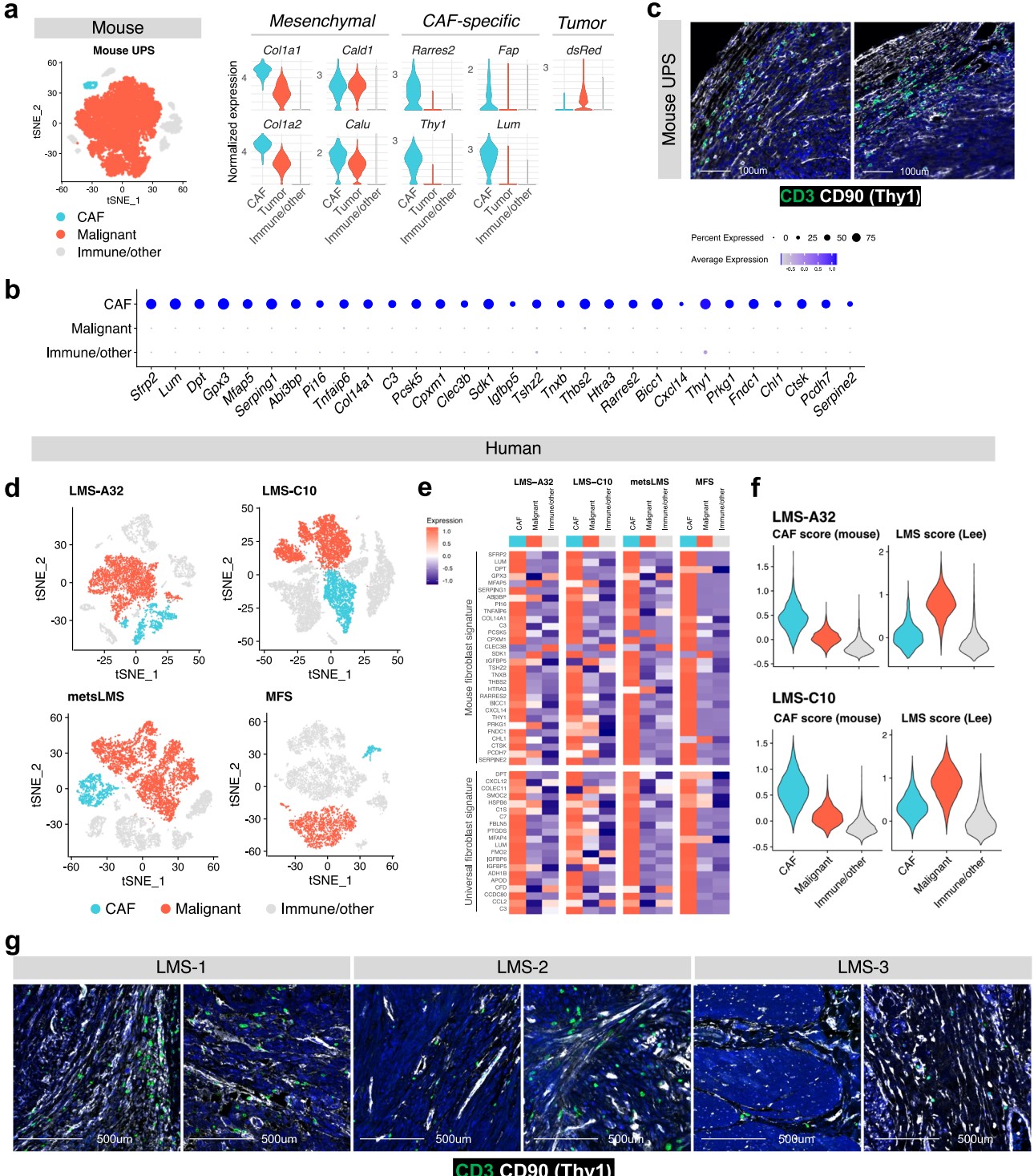

**Fig. 2 | CAF identified in human sarcoma share similarities with murine sarcoma-associated CAF. a** t-distributed stochastic neighbor embedding (t-SNE) depicting the immune, malignant, and fibroblastic compartments of the mouse UPS tumor mass (Ccne1+, *n* = 12 individually hashed mice) (left panel). Expression of mesenchymal markers and CAF-specific markers (right panel). **b** Genes enriched in CAF vs dsRED+ sarcoma cells in murine Ccne1+ tumors. **c** Multiplex immunohistochemistry staining of CD3 and CD90 at the tumor margin of two Ccne1+ tumors. Images are representative of three independent experiments.

**d** t-distributed stochastic neighbor embedding (t-SNE) depicting the immune, malignant, and CAF compartments of the human sarcomas including two primary uterine leiomyosarcoma (A32, C10), one high grade metastatic leiomyosarcoma (metsLMS), and one primary myxofibrosarcoma (MFS). **e** Heatmap of the mouse CAF signature and universal fibroblast signature in 4 human sarcoma samples. **f** Averaged expression of the mouse CAF signature and the LMS signature in the primary LMS samples. **g** Multiplex immunohistochemistry staining of CD3 and CD90 in three independent human leiomyosarcoma (LMS) cases.

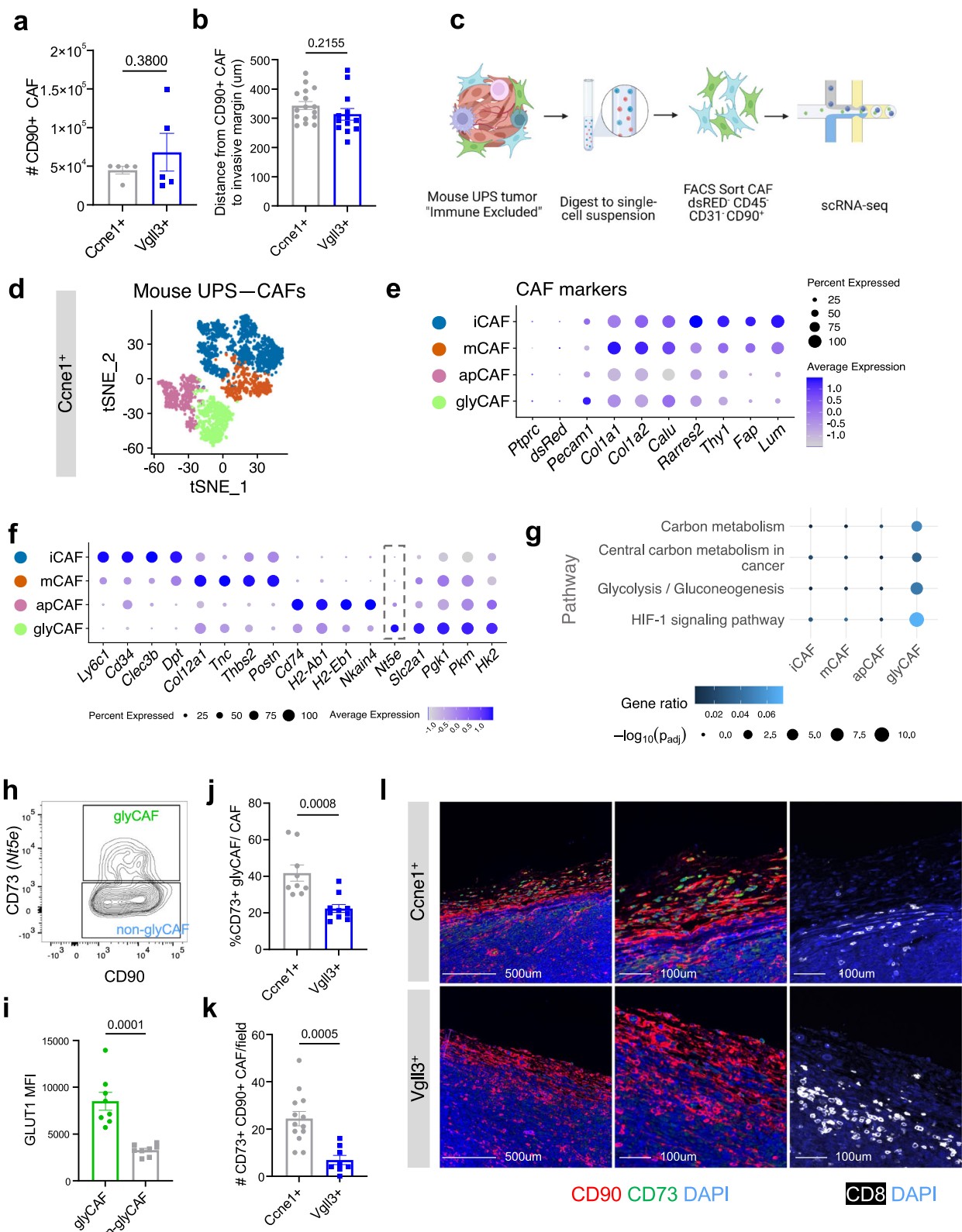

glucose transporter GLUT1 in flow cytometry, we ascertained that glyCAF (CD90⁺ CD73⁺) express ~2.5-fold higher levels of GLUT1 relative to the non-glyCAF (CD90⁺ CD73⁻) (Fig. 3h, i). Moreover, we FACS-sorted the CD73⁺ and CD73⁻ CAFs and validated their glycolytic signature by RT-qPCR quantification of *Slc2a1* and *Nt5e* as well as downstream glycolytic enzymes *Pgk1* and *Pkm* (Supplementary Fig. 3d). Functionally, we measured in vivo glucose transport in the CAFs by

injecting in the tumor-bearing mice the fluorescent glucose analog 2-NBDG and confirmed that CD73⁺ glyCAF exhibit increased glucose uptake compared to CD73⁻ non-glyCAF (Supplementary Fig. 3e). Given the previous association of specific CAF-subtypes in promoting immune exclusion[5,45,46], we asked if glyCAF are enriched in immune-excluded tumors compared to the highly infiltrated ones. Interestingly, we observed a striking increase in the proportions of glyCAF

**Fig. 3 | Glycolytic CAF (glyCAF) are the dominant CAF-subtype in immune excluded tumors. a** Absolute counts of CD90$^+$ CAF by flow cytometry ($n = 5$ mice). **b** Quantification of the distance from CD90$^+$ CAF stroma to tumor invasive margin assessed by immunofluorescence staining. Dots represent individual distances from independent CD90+ cells to the tumor margin ($n = 16$ Ccne1$^+$, $n = 13$ Vgll3$^+$) taken from $n = 3$ independent mice. **c** Schematic of isolation of mouse CAF from Ccne1$^+$ tumors for scRNA-seq ($n = 4$ mice). **d** t-SNE depicting 4 major CAF clusters; inflammatory CAF (iCAF), matrix CAF (mCAF), antigen-presenting CAF (apCAF), and glycolytic CAF (glyCAF). **e** Dot plot of cell-typing markers used to confirm fibroblast identity in CAF clusters. **f** Expression of top marker genes differentially expressed by the CAF sub-clusters. Box denotes *Nt5e* (CD73) expression in CAF sub-clusters. **g** KEGG pathways enriched in the glyCAF cluster. **h** Flow cytometry gating strategy for glyCAF (CD90$^+$CD73$^+$) and non-glyCAF (CD90$^+$CD73$^-$) gated on dsRED$^-$ CD45$^-$ CD31$^-$ cells. **i** Median fluorescence intensity (MFI) of GLUT1 by flow cytometry ($n = 8$ mice). **j** Proportions of glyCAF (CD73$^+$ CD90$^+$) by flow cytometry ($n = 9$ mice). **k** Quantification of CD73$^+$ CD90$^+$ glyCAF in ROIs encompassing the tumor margin (Ccne1$^+$: $n = 13$ ROI, Vgll3$^+$: $n = 8$ ROI from $n = 3$ mice). **l** Immunofluorescence of glyCAF (top) and CD8$^+$ T cells (right panel) at the tumor margin. Images are representative of two independent experiments. Unless otherwise indicated, results are presented as mean ± SEM and *p*-values are derived by a two-tailed unpaired Student's *t* test. Source data are provided as a Source Data file.

(CD90$^+$ CD73$^+$) in the immune-excluded (Ccne1$^+$) model compared to the highly infiltrated (Vgll3$^+$) one (Fig. 3j), while absolute numbers of total CD90$^+$ CAF were unchanged (Fig. 3a). Immunofluorescence staining confirmed flow cytometry results and showed a net enrichment of glyCAF at the tumor margin, near CD8$^+$ T cells in the Ccne1$^+$ model (Fig. 3k, l). These data indicate that immune-infiltrated and -excluded tumors present similar amount of CAFs but different subtypes, with CAFs exhibiting glycolytic metabolism being enriched at the border of the immune-excluded tumors.

## GLUT1 inhibition targets glycolytic CAF and promotes intratumoral T-cell infiltration

Given the differential accumulation of glyCAF in the immune-excluded tumors, we next tested if GLUT1 inhibition (GLUT1i, BAY-876) is sufficient to reprogram the glyCAF metabolism and promote the infiltration of CD8$^+$ T cells in the inner tumor parenchyma. GLUT1i treatment significantly reduced the CAF glycolytic metabolism (Fig. 4a and Supplementary Fig. 4a) and the number of CD73$^+$ CD90$^+$ glyCAF both at the tumor margin (Fig. 4b, c) and in the tumor mass (Fig. 4d). However, while we observed a reduction in glyCAF following GLUT1 inhibition, an overall net increase in total CD90$^+$ CAF was noted (Fig. 4e), prompting the investigation on how the remaining non-glyCAF compartment responds to GLUT1i. Accordingly, we FACS-sorted the total CAFs (dsRED$^-$, CD45$^-$ CD31$^-$ CD90$^+$) from GLUT1i treated tumors and analyzed the transcriptional states of the sorted cells by scRNA-seq. Upon GLUT1i, we observed a reduction only in the glyCAF gene signature, while iCAF, mCAF and apCAF states were unaltered (Fig. 4f), suggesting that among the CAF sub-types, only the glyCAF are affected by GLUT1i. Concomitantly to the reduction of the glyCAF proportion and metabolic features, we observed a net overall increase in CD8$^+$ T-cell infiltration in the tumor mass, with a significant increase of CD8$^+$ T cells localizing in the tumor parenchyma, as opposed to the tumor margin (Fig. 4g–i).

T cells and tumor associated macrophages (TAMs) can also exhibit glycolytic metabolism[49,50], thus GLUT1 inhibition could potentially alter their functional properties. To investigate whether T cells and TAMs are directly affected by GLUT1i, we FACS-sorted CD45$^+$ cells immune cells from GLUT1i treated tumors and profiled them by scRNA-seq. First, we observed that unlike the glyCAF, GLUT1 inhibition did not reduce the glycolytic signature of the myeloid or lymphoid cells (Supplementary Fig. 4b, c), suggesting that possible compensatory mechanisms to GLUT1 inhibition exist in immune cells. Because myeloid cells are the most frequent immune cells in the sarcoma mass, we also assessed changes in the proportions of pro- and anti-tumorigenic macrophages, as previously profiled[51]. Among the myeloid cells, we identified three populations of macrophages (MΦ *Thbs1*$^+$, MΦ *C1qa*$^+$, MΦ *Spp1*$^+$), monocytes, inflammatory monocytes/macrophages (Mono/MΦ Ifi), and dendritic cells (moDC/cDC) (Supplementary Fig. 4d, e), consistent with previous reports[51]. Proportions of dendritic cells, monocytes, Mono/MΦ Ifi, MΦ*Thbs1*$^+$ and MΦ *C1qa*$^+$ were not altered by GLUT1i, although the proportion of MΦ *Spp1*$^+$ may be slightly increased upon GLUT1i (Supplementary Fig. 4f). All together these observations suggest that no or only minor changes occur in the myeloid and lymphoid compartments upon application of GLUT1i, while glyCAF are primarily by the treatment.

Lastly, we asked if blocking glycolysis specifically in the malignant tumor cells could impact T-cell trafficking to the tumor mass. GLUT1 knock-down in the Ccne1$^+$ tumor cells (Fig. 4j, k and Supplementary Fig. 4g, h) did not resolve the exclusion of the CD8$^+$ T cells and did not phenocopy the results obtained with GLUT1 inhibition by BAY-876, suggesting once again that changes in the glycolytic properties of the glyCAF are primarily responsible for the observed increase in T cell tumor infiltration. Furthermore, in line with these observations we assessed that glyCAF are in closer proximity to CD8$^+$ T cells (Fig. 4l, m) relative to non-glyCAF and CSF1R$^+$ TAMs, thus suggesting that the nature of the glyCAF-CD8 interaction may restrict T cells to the peritumoral region, and this interaction may be reversed by the GLUT1i-dependent decrease of glyCAF. Together, these observations suggest that the lymphoid, myeloid and tumor-cell intrinsic glycolytic metabolism is not as critical as that of glyCAF in determining T-cell infiltration.

## CXCL16$^+$ glyCAF retain CD8$^+$ T cells at the tumor margin

To functionally characterize the role of glyCAF in T-cell migration, we utilized a transwell migration assay that recapitulates the spatial organization of the tumor mass in vitro. We FACS-sorted glyCAF (CD90$^+$CD73$^+$) and non-glyCAF (CD90$^+$CD73$^-$) from Ccne1$^+$ tumor bearing mice (Supplementary Fig. 5a) and briefly expanded them in vitro. The resulting CAF lines did not exhibit tumorigenic potential when transplanted in vivo (Supplementary Fig. 5b.), further supporting their non-malignant CAF phenotype. First, we validated that the two isolated populations maintain their differential glycolytic rate (Fig. 5a and Supplementary Fig. 5c), uptake of 2-NBDG (Supplementary Fig. 5d) and differential expression of glycolytic pathways (Supplementary Fig. 5e). With these cells, we set up a transwell migration assay whereby activated CD8$^+$ T cells were seeded in the top chamber, on top of a monolayer of either glyCAF or non-glyCAF and allowed to migrate towards a gradient of CXCL10 and Ccne1$^+$ tumor cells in the bottom chamber (Fig. 5b). In this regard, we observed that glyCAF exhibit the strongest blocking of CD8$^+$ T cell migration toward the tumor cells compared to the non-glyCAF, Ccne1$^+$ cancer cells, and macrophages (Fig. 5c), Then, we investigated whether blocking GLUT1 in the glyCAF impacts their functions. Accordingly, we observed that the glyCAFs lose their capability to block T-cell migration both in the presence of GLUT1i or when the expression of GLUT1 is silenced by a short hairpin RNA (shRNA) targeting *Slc2a1* (Glut1-KD) (Fig. 5d and Supplementary Fig. 5f), reinforcing the idea that glyCAF and their glycolytic properties are the main determinants for the trafficking of T cells in the tumor mass.

After defining the critical role of the glyCAF in preventing T-cell infiltration of the tumor parenchyma, we next investigated the underlying molecular mechanisms. Accordingly, we defined possible CAF-CD8$^+$ T cell interactions that would promote defective T-cell migration by analyzing differentially expressed receptor and ligand pairs between the CAF subsets and CD8$^+$ T cells. We isolated immune

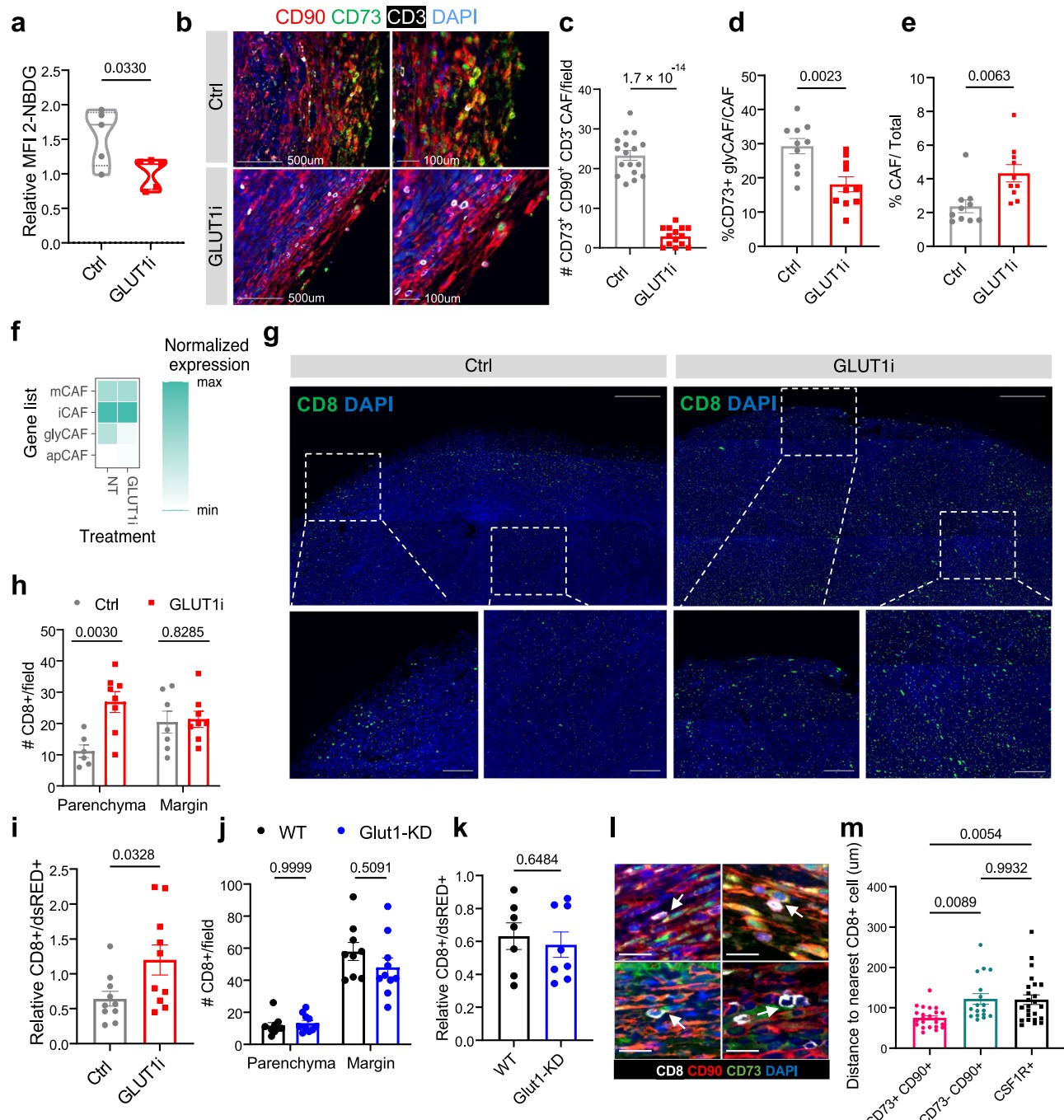

**Fig. 4 | GLUT1 inhibition targets glycolytic CAF and promotes intratumoral T cell infiltration. a** Glucose uptake of glyCAF quantified by flow cytometry of 2-NBDG (FITC) MFI relative to non-glyCAF (*n* = 5 mice). **b** Multiplex immunofluorescence staining of CAF marker CD90 (red) glyCAF marker CD73 (green), and CD3 (white) at the tumor margin. Images are representative of two independent experiments with *n* = 3 mice. **c** Quantifications of the number of CD73⁺ CD90⁺ cells (glyCAF) in ~1 mm² ROIs encompassing the tumor margin (Ctrl: *n* = 17 ROI, GLUT1i: *n* = 14 ROI from *n* = 3 mice). **d** Proportion of CD73⁺ glyCAF (dsRED⁻, CD45⁻, CD31⁻, CD90⁺, CD73⁺) by flow cytometry (*n* = 10 mice). **e** Proportion of CD90⁺ CAF (dsRED⁻, CD45⁻, CD31⁻, CD90⁺) by flow cytometry (*n* = 10 mice). **f** Averaged expression of mCAF, iCAF, glyCAF, and apCAF signature genes (top 10 DEG per cluster) in CAFs from mouse tumors treated with GLUT1i or control (NT) (*n* = 4 mice). **g** Immunofluorescence staining of CD8⁺ cells (green) in Ctrl and GLUT1i treated Ccne1⁺ tumors (Scale bars 500 μm (top) and 100 μm (bottom)). Images are representative of two independent experiments with *n* = 3 mice. **h** Quantifications of CD8⁺ cells/field in ROIs encompassing the tumor margin or the tumor parenchyma in Ctrl and GLUT1i treated Ccne1⁺ tumors (*n* = 3 mice) and **i** relative

proportions of tumor infiltrating CD8⁺ T cells (CD45⁺CD8⁺) cells determined by flow cytometry (*n* = 10 mice). **j** Quantifications of CD8⁺ cells/field in ROIs encompassing the tumor margin or the tumor parenchyma in WT and Glut1-KD tumors (Ctrl, Parenchyma: *n* = 6 ROI, Margin: *n* = 7 ROI; GLUT1i, Parenchyma: *n* = 8, Margin *n* = 8 ROIs from *n* = 3 mice) and **k** relative proportions of tumor infiltrating CD8⁺ T cells (CD45⁺CD8⁺) cells determined by flow cytometry (*n* = 8 mice). **l** Multiplex immunofluorescence staining of CAF marker CD90 (red), glyCAF marker CD73 (green), and CD8 (white) at the tumor margin. White arrows illustrate association between glyCAF and CD8⁺ T cells. Scale bars 100 μm. Images representative of three independent experiments with *n* = 3 mice. **m** Quantifications of the distance to the nearest CD8⁺ T cell using multiplex immunohistochemistry images. Dots represent individual cell-cell interactions (*n* = 24 CD73⁺ CD90⁺, *n* = 17 CD73⁻ CD90⁺, *n* = 24 CSF1R⁺) acquired from *n* = 3 mice each group. *P*-values determined by one-way ANOVA with Tukey's multiple comparisons test. Unless otherwise indicated, results are presented as mean ± SEM and *p*-values are derived by a two-tailed unpaired Student *t* test. Source data are provided as a Source Data file.

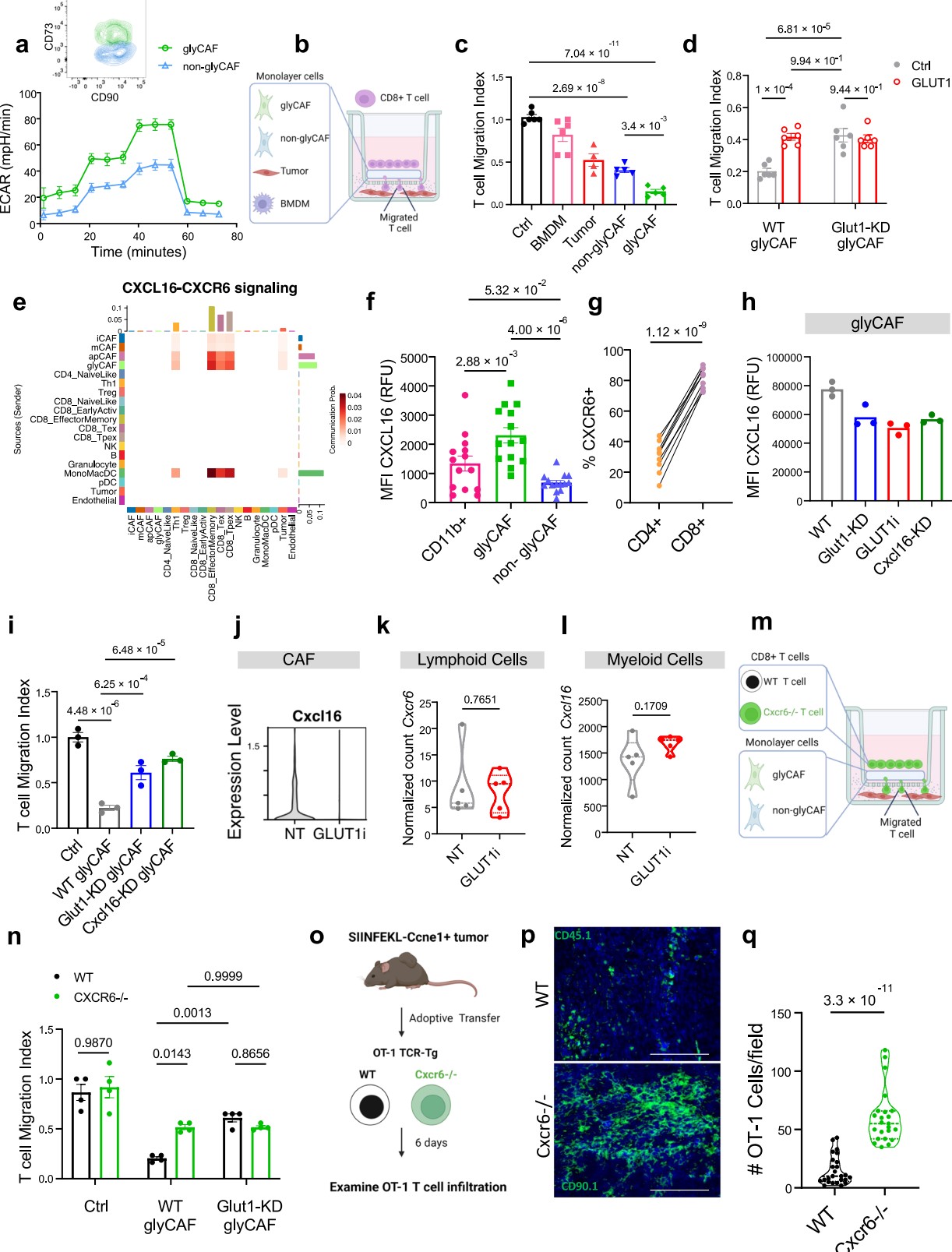

cells from Ccne1[+] tumors and subjected them to scRNA-seq; following initial unsupervised clustering of immune cells and identification of T cells based on the expression of *Cd3e*, subtypes of CD4[+] and CD8[+] T cells were identified using the T-cell annotation R package ProjecTILs[52]. Immune cells and the four CAF subclusters (Fig. 3d) were analyzed by CellChat[53] for potential ligand-receptor interactions. Among the strongest interactions was *CXCL16-CXCR6* (Fig. 5e). *Cxcl16*

encodes a T-cell retention chemokine that has been implicated in the spatial positioning of CXCR6[+] T cells in the tumor stroma near CXCL16-producing dendritic cells[54] and preventing egress of tumor-resident memory precursor T cells leading to enhanced metastasis[55]. Notably, the transmembrane domain of CXCL16 allows it to function as both an adhesion molecule as well as a secreted chemoattractant in fibroblasts and myeloid cells[56,57]. Indeed, in our model we observe that glyCAF,

**Fig. 5 | CXCL16⁺ glyCAF retain CD8⁺ T cells at the tumor margin. a** Seahorse metabolic profiling of in vitro CAF. Data shown (*n* = 3 technical replicates) is representative of three independent experiments. **b** Schematic of T cell transwell migration assay: bottom chamber containing tumor cells, top chamber containing one of the following: BMDM, tumor, glyCAF or non-glyCAF, seeded with activated CD8⁺ T cells. **c, d** Quantification of migrated CD8⁺ T cells. The x-axis refers to the cell-type in the upper chamber of the transwell system, certain cultures treated with GLUT1i (BAY-876, 75 uM). Representative data from one experiment shown out of three total experiments (*n* = 6 Ctrl, *n* = 6 BMDM, *n* = 4 Tumor, *n* = 5 glyCAF, *n* = 6 non-glyCAF) and (*n* = 6 WT glyCAF, and *n* = 6 non-glyCAF). *p*-values determined by two-way ANOVA with Tukey's multiple comparison. **e** Predicted ligand-receptor interaction scores of CXCL16-CXCR6 signaling pathway (averaged expression from *n* = 4 mice). Data shown is representative of three total experiments. **f** Median fluorescence intensity (MFI) of CXCL16 (*n* = 14 mice). **g** Proportion of T cells expressing CXCR6 (*n* = 9 mice). **h** Median fluorescence intensity (MFI) of CXCL16 in glyCAF (in vitro). Dots correspond to *n* = 3 technical replicates from one experiment are shown out of three with a similar trend. **i** Migration index of CD8⁺ T cells cultured with glyCAF. *p*-values are presented as determined by one-way ANOVA with Fisher's LSD. Representative data from one experiment is shown out of three (*n* = 3 biological replicates). **j** *Cxcl16* expression in CD90⁺ CAF. **k** Expression of *Cxcr6* mRNA in lymphoid cells from GLUT1i and control (NT) tumors (*n* = 5 mice). **l** Expression of *Cxcl16* mRNA in myeloid cells from GLUT1i and control (NT) tumors (*n* = 5 mice). **m** Schematic of T cell transwell migration assay with WT or Cxcr6⁻/⁻ CD8⁺ T cells. **n** Migration index of WT or Cxcr6⁻/⁻ T cells cultured in transwell in the presence of glyCAF. Dots correspond to *n* = 4 biological replicates from one experiment out of two with a similar trend. *p*-values are presented as determined by two-way ANOVA with Tukey's multiple comparison test. **o** Schematic of adoptive transfer experiment. **p** Multiplex IHC staining of transferred WT OT-1 (CD45.1) and Cxcr6⁻/⁻ (CD90.1). Scale bars 500 μm. Image representative of *n* = 4 mice. **q** Quantification of transferred OT-1 T cells by IHC. *n* = 25 ROIs (WT) and *n* = 23 s ROI (Cxcr6⁻/⁻) from *n* = 4 mice. Unless otherwise indicated, results are presented as mean ± SEM and *p*-values derived by two-tailed unpaired Student's *t* test. Source data are provided as a Source Data file.

apCAF, and myeloid cells (MonoMacDC cluster) exhibit communication along the CXCL16-CXCR6 axis with CD8⁺ T cells. However, while the expression of *Cxcl16* mRNA is similar in between glyCAF, apCAF, and myeloid cells, post-transcriptional regulations of the RNA are differentially occurring in these cells, with the highest protein expression observed in the glyCAF (Fig. 5f), suggesting that glyCAF are the primary source of CXCL16 at the tumor margin. In the same line, flow cytometry of T cells confirmed higher expression of CXCR6 on CD8⁺ T cells relative to CD4⁺ T cells (Fig. 5g), in agreement with the observation that CD8⁺ T cells are especially segregated at the tumor margin in the immune-excluded model.

We next asked if *Cxcl16* expression in glyCAF is regulated by GLUT1 and whether CXCL16 plays a functional role in T-cell exclusion. First, we observed a significant reduction of CXCL16 at both the transcript (Supplementary Fig. 5g) and protein level (Fig. 5h), upon GLUT1 inhibition (BAY-876, 75uM) or silencing of GLUT1 in the glyCAF using a short hairpin RNA (shRNA) targeting *Slc2a1* (Glut1-KD). Functionally, silencing *Cxcl16* (Cxcl16-KD) in glyCAF rescued CD8⁺ T cell migration, phenocopying the inhibition and silencing of Glut1 (Fig. 5i), further corroborating the hypothesis that CXCL16 is a critical regulator of the CD8⁺ T cells migratory abilities, and that GLUT1 controls *Cxcl16* expression in glyCAF.

Because we observed that myeloid cells may also communicate with CD8⁺ T cells through the CXCL16-CXCR6 axis, we assessed functionally whether this impacts T-cell migration. In contrast to what we observed in glyCAF, Glut1-KD in macrophages did not reduce *Cxcl16* expression, and GLUT1i or Glut1-KD macrophages did not alter the migration of T cells (Supplementary Fig. 5h, i). In vivo, GLUT1i diminishes *Cxcl16* expression in the CAF (Fig. 5j), but not in myeloid cells, and lymphoid cells expression of *Cxcr6* is unaffected (Fig. 5k, l), further confirming that *Cxcl16* expression is regulated by GLUT1 primarily in glyCAF but no other TME cell types. These results suggested that GLUT1i may function in repressing *Cxcl16* expression, which repolarizes the glyCAF towards a T-cell permissive non-glyCAF phenotype, which support CD8⁺ T cell trafficking into the inner tumor mass.

To further corroborate the role of CXCL16-CXCR6 signaling in restricting T-cell migration, we assessed the migratory capability of CXCR6-deficient (*Cxcr6⁻/⁻*) CD8⁺ T cells in vitro and in vivo[57,58]. First, we observed that *Cxcr6⁻/⁻* CD8⁺ T cells exhibited improved migration relative to WT when cultured in transwell with glyCAF (Fig. 5m, n). Then, we assessed if *Cxcr6⁻/⁻* T cells exhibit increased infiltration into the tumor mass in vivo. Congenically marked WT (CD45.1) or *Cxcr6⁻/⁻* (CD90.1) OT-1 T cells (Supplementary Fig. 5j) were adoptively transferred into Ccne1⁺ tumors expressing the SIINFEKL antigen (Ccne1-SIIN) and the T-cell infiltration was assessed 6 days later (Fig. 5o). Histological analysis of Ccne1-SIIN tumors showed an increased

accumulation of transferred CXCR6-deficient OT-1 T cells in the tumor parenchyma relative to wildtype T cells (Fig. 5p, q), suggesting that releasing CD8⁺ T cells from CXCL16-CXCR6 mediated retention at the tumor margin can reverse T-cell exclusion.

## GLUT1 inhibition facilitates chemotherapy response

Doxorubicin is often used in the treatment of STS patients[24]; it induces immunogenic cell death leading to increased antigen presentation by dendritic cells to CD8⁺ T cells in the tumor-draining lymph node, thereby triggering a cytotoxic T cell response within the tumor mass[59]. Accordingly, high T cell content in the TME favors chemotherapy response (Fig. 1h). Building upon our findings that GLUT1 inhibition promotes the accumulation of CD8⁺ T cells in the tumor mass, we hypothesized that combining GLUT1 inhibition with doxorubicin could impede tumor growth by enhancing the antitumor function of cytotoxic T cells. Thus, we first treated Ccne1+ tumor-bearing immunocompetent mice (immune-excluded, chemo-resistant model) with a standard regimen of doxorubicin (DOX) (6 mg/kg), alone or in combination with the GLUT1 inhibitor (GLUT1i, BAY-876, 5 mg/kg) (Fig. 6a). While neither doxorubicin (D) (Fig. 1g) or GLUT1i (G) (Fig. 6b, c) as single agents had a significant effect on tumor growth or survival, the combination of DOX +GLUT1i (D + G) resulted in significantly impaired tumor growth (Fig. 6d, e), prolonged survival (Fig. 6f), and reduction in the proportion of dsRED⁺ cancerous cells in the tumor mass (Fig. 6d). Accordingly, the combinatorial treatment resulted in increased proportions of both CD4⁺ and CD8⁺ T cells in the tumor mass (Fig. 6g), where CD8⁺ T cells also expressed higher levels of the activation marker PD-1 (Fig. 6h) and increased numbers of CD8⁺ T cells spatially localized in the inner tumor parenchyma (Fig. 6i, j). Furthermore, we observed an increase in Granzyme-B⁺CD8⁺ T cells in the tumor parenchyma in DOX+GLUT1i treated mice relative to the single treatments (Fig. 6k and Supplementary Fig. 6a), suggesting that the combinatorial treatment promotes T-cell activation and cytotoxicity. Among CD4⁺ T cells, we noted no changes to the number of tumor-infiltrating FOXP3⁺ Tregs (Supplementary Fig. 6b), and CD4⁺ FOXP3- effectors in the tumor parenchyma following the combination treatment (Supplementary Fig. 6c), suggesting that only CD8⁺ T cells are critically implicated in the observed phenotype. To further assess whether the reduction of tumor growth upon DOX+GLUT1i is, at least partially, CD8⁺ T cell-dependent, we performed an analogous experiment in which we depleted CD8⁺ T cells from Ccne1⁺ tumor bearing mice before DOX +GLUT1i treatment (Supplementary Fig. 6d). Depletion of the CD8⁺ T cells resulted in significantly increased tumor growth (Fig. 6l) suggesting that CD8⁺ T cells are critical drivers of the anti-tumor effect observed upon DOX+GLUT1i (Fig. 6m).

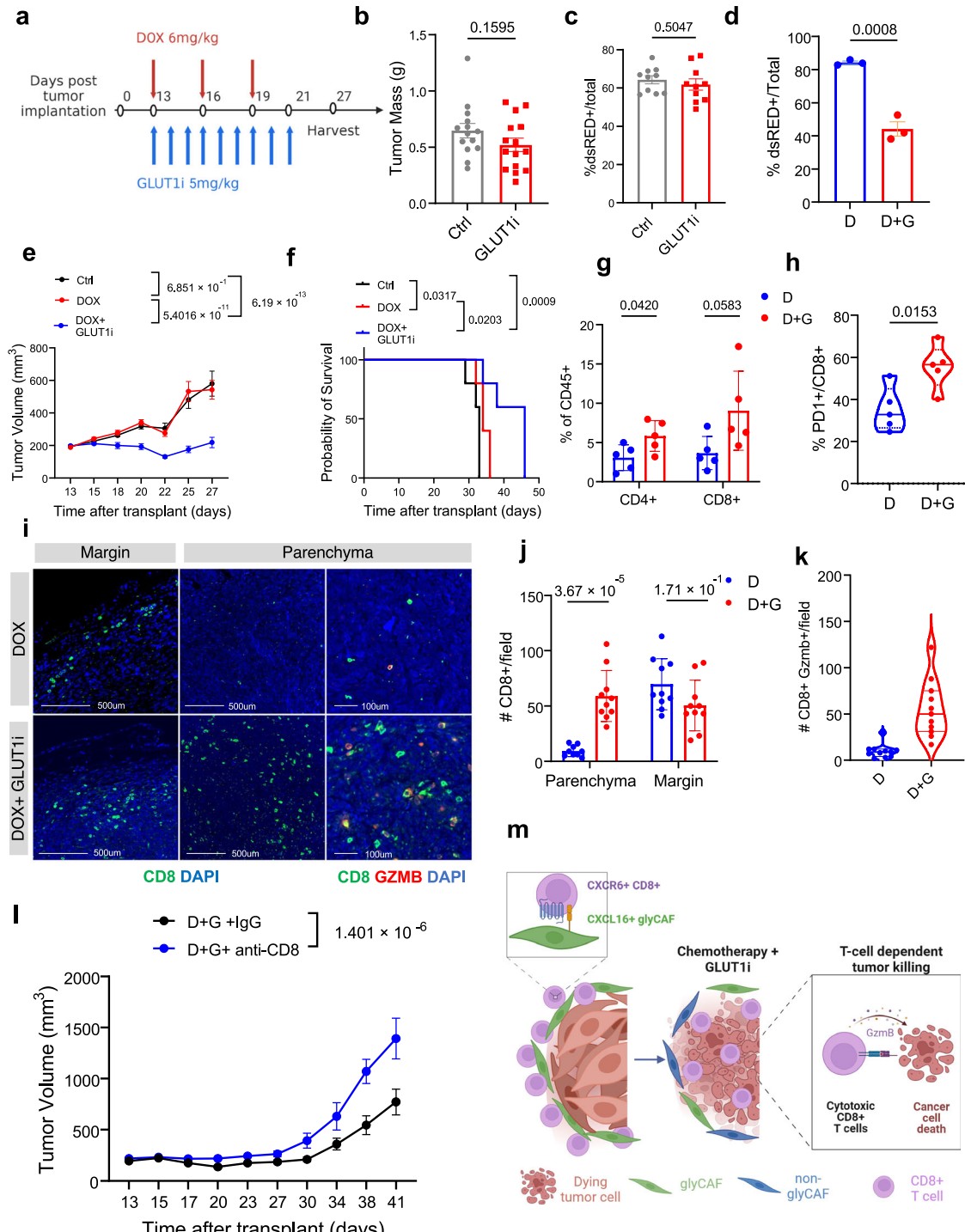

**Fig. 6 | GLUT1 inhibition facilitates chemotherapy response. a** Treatment timeline for Doxorubicin (DOX, D) and GLUT1 inhibition (GLUT1i, G). **b** Tumor mass measured in grams of GLUT1i treated tumors ($n = 10$ mice). **c** Flow cytometry proportions of dsRED[+] tumor cells in GLUT1i treated tumors ($n = 10$ mice). **d** Flow cytometry proportions of dsRED[+] tumor cells in DOX (D) or DOX+GLUT1i (D + G) treated tumors ($n = 3$ mice). **e** Tumor bearing mice were treated with DOX (6 mg/ kg) with or without GLUT1i (5 mg/kg) for 9 days. Tumor volume was measured every 2–3 days. Two-way ANOVA with Tukey's multiple comparison was used to determine differences in tumor growth between groups ($n = 8$ mice). **f** Kaplan-Meier survival curve for DOX+GLUT1i treated mice ($n = 5$ mice). Log-rank test was used to determine differences in survival between groups. **g** Proportions of CD4[+] and CD8[+] T cells assessed by flow cytometry ($n = 5$ mice). **h** Proportion of PD1[+] CD8[+] T cells by flow cytometry ($n = 5$ mice). **i, j** Immunofluorescence staining of CD8[+]

cells (green) and Granzyme B (red) infiltrating the tumor margin and parenchyma and quantifications of the number of CD8[+] cells in ~1 mm$^2$ ROIs encompassing the tumor margin and parenchyma of DOX or DOX+GLUT1i treated Ccne1[+] tumor bearing mice (D, Parenchyma: $n = 9$ ROI, Margin: $n = 10$ ROI; D + G, Parenchyma: $n = 10$ ROI, Margin, $n = 10$ ROI from $n = 3$ mice). **k** Quantifications of CD8[+] GZMB[+] cells in the tumor parenchyma ($n = 11$ ROIs from $n = 3$ mice). **l** Ccne1[+] tumor-bearing mice, with or without anti-CD8a neutralizing antibody treatment, received 9 days of combination DOX+GLUT1i treatment. Tumor volume was monitored every 2–3 days. Two-way ANOVA with Tukey's multiple comparison was used to determine differences between groups ($n = 5$ mice). **m** Schematic illustrating synergistic T-cell dependent effect of GLUT1i and DOX promotes anti-tumor immunity. Unless otherwise indicated, results are presented as mean ± SEM and *p*-values are derived by a two-tailed Student's *t* test. Source data are provided as a Source Data file.

## Discussion

The presence of T cells in the inner tumor parenchyma significantly impacts the prognosis and treatment outcomes of tumors with varied histology, including soft-tissue sarcomas (STS)[26,27]. However, only a subset of tumors exhibits high levels of infiltrating lymphocytes[3,60], emphasizing the need to boost T-cell infiltration for improved responses to T-cell-based immunotherapies like immune checkpoint blockade, CAR-T cell therapy, and chemotherapy. Despite this, our understanding of the mechanisms behind T-cell exclusion from the tumor parenchyma, especially in STS, remains limited. In our study, we utilized mouse models that replicate immune-infiltrated and immune-excluded microenvironments observed in STS patients. Our goal was to investigate the cellular and molecular factors contributing to immune exclusion, with a specific focus on CAFs. CAFs have been recognized for obstructing T-cell trafficking in carcinomas[5,6] but have been understudied in STS due to challenges in distinguishing CAFs from malignant cells. Through scRNA-seq and a tumor-tracking system, we profiled murine and human STS tumors, enabling the identification of specific markers to differentiate CAFs from sarcoma cells. Interestingly, we found distinct subpopulations of CAFs, and we focused especially on glycolytic CAF (glyCAF) for functional studies, which had not been extensively studied before. We observed that CXCL16[+] glyCAF were enriched at the tumor margin of immune-excluded STS tumors, but their frequency was lower in highly immune-infiltrated tumors. Functionally, we discovered that CXCL16[+] glyCAF act as a barrier between CD8[+] T cells and malignant cells, thereby preventing T-cell contact with cancer cells and infiltration into the inner tumor parenchyma. Our findings underscore the significance of CAF subtypes at the tumor margin, rather than the CAF overall abundance, in influencing Tcell exclusion or infiltration. GlyCAF share similar transcriptional profiles with previously identified myCAF[6,45,61], which are driven by TGFβ signaling and implicated in CD8[+] T cell exclusion. TGFβ also drives glycolytic metabolism in fibroblasts[62–64], uncovering a potential role for TGFβ in glyCAF biogenesis, although this warrants further investigation. This work highlights the importance of considering the functional features of CAFs when developing therapeutic strategies targeting the tumor microenvironment[65]. Accordingly, reprogramming the glycolytic properties of CAFs by inhibiting the GLUT1 glucose transporter led to decreased *Cxcl16* expression in the glyCAF and increased infiltration of cytotoxic CD8[+] T cells into the tumor, independent of cancer cell-intrinsic metabolism and other immune cells in the TME. Notably, disrupting glucose uptake solely in Ccne1[+] tumor cells did not replicate the increased T-cell trafficking observed with global GLUT1 inhibition. Combining GLUT1 inhibition with doxorubicin, which is often used clinically for STS, resulted in reduced tumor growth in a T cell-dependent manner. While clinical use of GLUT1 inhibitors is yet to be established, preclinical investigations using small molecules targeting GLUT1, such as STF-31, WZB-117, and BAY-876, have shown promising results in various cancers[66]. Additionally, combining metformin, a common therapy for type 2 diabetes that reduces blood glucose levels, with anti-PD-1 treatment has shown promising outcomes in promoting T cell-dependent tumor regression and is currently being evaluated in clinical trials[67,68].

While the current study offers mechanistic insights with the potential to impact the clinical practice for immune-excluded sarcoma, some limitations have been noted. Firstly, further investigations are needed to comprehensively profile and functionally characterize changes in the TME cells following GLUT1 inhibition. Specifically, we observed that GLUT1 inhibition impacted the glycolytic metabolism, and thereby *Cxcl16* levels, of the glyCAF but not of myeloid and lymphoid cells. While compensatory mechanisms through the up-regulation of other glucose transporters in immune cells may occur, additional investigations are required to clarify this matter. Moreover, while utilizing GLUT1 inhibitors (GLUT1i) to repolarize T-cell-restrictive

glyCAF at the tumor margin shows promising results, lineage tracing experiments are necessary to unravel the nature of the glyCAF reprogramming induced by GLUT1 inhibition. Additionally, functional investigations into the role of other CAF subtypes (iCAF, mCAF, apCAF) in sarcoma are warranted. Another aspect requiring further exploration is the functional contribution of tumor-associated macrophages (TAMs) upon GLUT1 inhibition. Our analysis revealed an increase in *Spp1*[+] TAMs upon GLUT1i treatment, with previous studies indicating their highly immunosuppressive nature[51,69,70]. We may speculate that *Spp1*[+] TAMs are not direct targets of GLUT1i; rather, their expansion may occur indirectly as a compensatory mechanism of immunosuppression due to the increased presence of intratumoral cytotoxic CD8[+] T cells.

The investigation into how glyCAF influence T-cell migration and tumor trafficking merits further exploration. Although we focused on the CXCL16/CXCR6 axis as a potential mechanism for T cell blockade at tumor margins, proposing the disruption of this axis by blocking glycolysis in CXCL16[+] glyCAF[54,55], future studies may delve into more targeted interventions. These could involve inhibiting small molecules or blocking antibodies specific to the CXCL16/CXCR6 axis. Considering the spatial positioning of CXCL16-expressing cells and their impact on the CXCR6[+] T cells within the tumor microenvironment is crucial. Previous studies have implicated tumor-associated dendritic cells[54] and tumor cells[55] in positioning lymphocytes in the tumor, where the spatial niche (peritumoral or intratumoral) of CXCL16 expressing cells defines the location of the CXCR6[+] T cells. The expression of CXCL16 by tumor cells traps CXCR6[+] effector T cells in the tumor mass, limiting the generation of memory T cells by sequestering their precursors in the tumor. In the context of CAR-T cells, where trafficking is a major challenge, overexpressing CXCR6 in CAR-T cells improves their trafficking when CXCL16 is highly expressed on tumor cells[71]. Therefore, it becomes critical to consider the cellular sources of CXCL16 and their spatial positioning within the tumor mass. Accordingly, while CXCL16[+] tumor cells may trap T cells in the tumor mass, a tumor border composed by CXCL16[+] glyCAF may, on the contrary, block T-cell migration and sequester the T cells in the tumor periphery. In this respect, it remains critical to decipher the contribution of myeloid-derived CXCL16 in positioning lymphocytes in the tumor microenvironment. Hence, employing conditional knockout mouse models of *Glut1* and *Cxcl16* in myeloid cells or glyCAF will be essential to unravel these complexities.

Lastly, we conducted the profiling of the tumor microenvironment in our murine STS models using a syngeneic, sex-matched approach, exclusively involving female mice in this study. Recognizing the significance of sex differences in the immune response to cancer, we initially conducted pilot experiments with male mice bearing Ccne1[+] and Vgll3[+] tumors, yielding comparable results in terms of immune composition and immune exclusion. However, further investigations in this regard are warranted to comprehensively assess and validate these findings.

In summary, our study highlights the potential of personalized therapies to reprogram CAFs, overcoming T-cell exclusion and enhancing the efficacy of chemotherapy and immunotherapies in solid tumors. Further research is essential to unravel the complexities of the interactions between CAFs, T cells, and tumor cells, paving the way for more effective treatments in clinical practice.

## Methods
### Mice
C57Bl/6J wild type (#000664), and p53[KO] (#002101) were purchased from The Jackson Laboratory and used to generate syngeneic tumor cell lines p53[KO] Ccne1[+] and p53[KO] Vgll3[+] from female mice. Sex-matched eight-week-old female mice were used as tumor recipients for all experiments. Cxcr6[−/−] OT-1 and Cxcr6[wt/wt] OT-1 mice were a gift from Amanda Lund in agreement with NYU Grossman School of Medicine.

Briefly, these mice were generated by crossing B6.SJL-PtprcaPepcb/BoyJ (CD45.1), C57BL/6-Tg(TcraTcrb)1100Mjb/J (OT-1), B6.129P2-Cxcr6tm1Litt/J (CXCR6[−/−])[72], B6.PL-Thy1a/CyJ (CD90.1), which were purchased from Jackson Laboratory. Maximal tumor burden allowed was 1.5 cm³. In some cases, this limit has been exceeded the last day of measurement and the mice were immediately euthanized. Maximal tumor burden and all other aspects of animal experiments were performed in accordance with the guidelines of Cedars-Sinai Medical Center Institutional Animal Care and Use Committee.

## GLUT1 inhibition and chemotherapy
Ccne1[+] tumor bearing mice were treated with 5 mg/kg BAY-876 (MedChemExpress) by oral gavage daily for 9 days beginning when the tumor volume reached 200 mm³. Doxorubicin (Cayman Chemical) was administered i.p. at 6 mg/kg every 3 days for 9 days beginning when the tumor volume reached 200 mm³.

## In vivo T-cell depletion
Anti-CD8a (BioXcell, BE0061) neutralizing antibody (100 μg) or IgG (BioXcell, BE0090) was administered i.p. beginning 4 days before tumor scaffold implantation and continuing every 4 days during the course of chemotherapy treatment. Following treatment, neutralizing antibodies were given once per week until the experimental endpoint.

## Tumor growth
Tumor volume was monitored every 2–3 days by caliper measurement. Tumor volume was calculated using the standard formula (width × length × length/2).

## Human cell lines
The human 293T cell line for viral preparation was purchased from ATCC (Cat # CRL-3216). Cells were grown in DMEM supplemented with glutamine, 10% fetal bovine serum (Gibco), 100 IU/ml penicillin, and 100 μg/ml streptomycin (Gibco). Cells were cultured in an incubator at 37 °C and 5% $CO_2$.

## Mouse mesenchymal stromal cells isolation, maintenance, and in vivo tumorigenesis
Subcutaneous sarcomas were generated according to protocols developed in our lab[31]. Briefly, long bones were collected from p53[KO] mice, crushed and digested with collagenase II (1 mg/ml) for 1 h at 37 °C on a shaker. Recovered cells were stained and FACS-sorted (CD45[−]CD31[−]Ter119[−]Sca1[+]PDGFRα[+]) to obtain mesenchymal stem cells and cultured in complete MesenCult medium (STEMCELL Technologies). MSCs were maintained in a humidified chamber with 5% $CO_2$ and 1% $O_2$, with half of medium changed every 3 days. After 7 days in culture at 1% $O_2$, cells formed visible CFU-F colonies; after this point cells were periodically split at 80% of confluency. To generate sarcoma cells, the p53[KO] mesenchymal cells were transduced for the stable expression of Ccne1 or Vgll3 and red fluorescent protein (see below for plasmid generation). The stable cells were assessed by RT-qPCR and/or western blot, expanded in vitro and then used for in vivo tumorigenesis assays. Experiments aimed at measuring in vivo tumorigenesis (subcutaneous tumors) were carried out following the protocol previously described in ref. 31. Briefly, 3D scaffolds (5 mm × 2 mm disks) made with reticulated polycarbonate polyurethane urea matrix (CS1-0502-25, Biomerix Corp/DSM Biomedical) were seeded with MSCs at a concentration of 1 × 10⁵ cells/scaffold. Cells were allowed to adhere to the scaffolds for a minimum of 6 h. Scaffolds were then implanted subcutaneously into mouse flanks, and tumors were harvested 3 weeks after implantation. After isolation from primary recipient mice, sarcoma cells were expanded in culture, carried in DMEM supplemented with glutamine, 10% fetal bovine serum, 100 IU/ml penicillin and 100 μg/ml streptomycin (Gibco), and maintained at 37 °C and 5% $CO_2$. Cells were then implanted into secondary recipients for sarcoma

generation and the experiments presented in the manuscript. Sarcomas were grown in mice until ~600 mm³. Tumors were resected and enzymatically digested.

## In vivo adoptive transfer
Ccne1[+] tumor cells were infected with lentiviral vectors encoding the SIINFEKL antigen (Addgene plasmid #185662). Scaffolds bearing Ccne1-SIIN cells were implanted into C57BL/6 mice and grown until the tumor volume reached 500m³. CD8[+] cells were isolated from the spleens of congenically marked Cxcr6[wt/wt] or Cxcr6[−/−] OT-1 mice (StemCell Easy Sep Mouse CD8[+] Isolation Kit) and 1 × 10⁶ cells were transferred intravenously into sex and age-matched tumor bearing hosts. Tumors were collected for histological analysis 6 days post adoptive transfer.

## Cell proliferation assay
Cell proliferation was measured by crystal violet staining[31]. Briefly, cells were plated at low confluency in 12-well plates (2000 cells per well) and allowed to proliferate for 5 days. Cell viability was measured by crystal violet staining (Sigma Aldrich, 0.1% in 20% methanol) of adherent cells after 10 min fixation with 10% formalin. After washing twice and air-drying, stained cells were washed with 10% acetic acid to solubilize the crystal violet, and OD595 values were measured with a spectrophotometer.

## Generation of retrovirus, lentivirus, knockdown, knockout and overexpressing cells
The retroviral vector pCMMP-MCS-IRES-mRFP (Addgene #36972) was used for the overexpression of Ccne1 and Vgll3 genes. The genes were amplified from the cDNA of mouse mesenchymal cells and cloned into the retroviral vector by using the Gibson Assembly kit (New England Biolabs). The expression of the transgene was assessed by RTqPCR. ShRNAs were cloned into the pLKO.1 lentiviral vector (Addgene #10879), following Addgene instructions. The shRNA sequences were designed according to the following program provided by the Broad Institute GPP Web Portal:

https://portals.broadinstitute.org/gpp/public/seq/search
Oligo Sequences:
shSlc2a1: CCGGATCACTGCAGTTCGGCTATAACTCGAGTTATAGCCGAACTGCAGTGATTTTTTG
shCxcl16: CCGGGCAGGGTACTTTGGATCACATCTCGAGATGTGATCCAAAGTACCCTGCTTTTTG

All the viral particles were produced in 293T cells, which were co-transfected with the specific viral vector and packaging-expressing plasmids: pECO for the retroviral vectors, and VSV-G, REV, and d8.74 for the lentiviral vectors. Transfection of the cells was performed by using Lipofectamine 3000 diluted in Opti-MEM, according to manufacturer instructions. Transfection medium was changed 8 h after transfection, and the lentiviral particles were collected 24 and 48 h after transfection. Viral supernatant was used with 10 μg/ml polybrene (TR-1003-G, Sigma Aldrich) to infect the cells, which were seeded at a confluence of 50% the day prior to transduction. Cells were incubated overnight with the viral supernatant, washed with PBS and then supplemented with complete medium. Antibiotic selection (puromycin 2 μg/ml) was performed at least 72 h post-infection.

## Tissue dissociation
Tumors were harvested, minced, and then enzymatically and mechanically digested using the 37C_m_TDK_2 protocol on the Miltenyi gentleMACS with Tumor Dissociation Kit for Mouse (Miltenyi Biotec, Auburn, CA). For experiments where CAF enrichment for scRNA-seq was performed, tumors were minced and digested in DMEM containing 0.8 mg/mL Dispase (Roche), 0.2 mg/mL Collagenase P (Gibco), and 0.1 mg/mL DNAse (Roche) for 1 h at 37 °C. Cell suspensions were washed in Cell Staining Buffer (BioLegend) and filtered through 70 μm

strainers (Bioland Scientific LLC, Paramount, CA). Red blood cells were lysed with ACK buffer.

## FFPE tissue dissociation

Formalin-fixed, paraffin-embedded tumor samples were prepared for single-cell Fixed-RNA Profiling (10x Genomics) using manufacturer protocols. In brief, two 25 µm sections were taken from each paraffin block. Tissue sections were deparaffinized in xylene, rehydrated through an ethanol gradient, dissociated on the gentleMACS Octo Dissociator (Miltenyi Biotec) in Liberase TH (Sigma, 1 mg/ml in RPMI) using the *37C_FFPE_1* program, and washed in quenching buffer (10x Genomics, p/n 2000516).

## Single-cell RNA sequencing and analysis

Single cell suspensions from mouse and human tumors were generated as above. In certain mouse experiments where indicated, cell suspensions from individual mice were each tagged with a unique TotalSeq-A Hashtag reagent (BioLegend, San Diego, CA)—an antibody-conjugated oligo barcode. Final cell suspensions were washed 3 times in PBS, filtered through 40 µm Bel-Art FlowMi strainers (Bel-Art/SP Scienceware, Wayne, NJ), viability verified by trypan staining or AO/PI staining, counted, and pooled into a single sample at a concentration of 1000 cells/uL. The cell suspension was loaded into the Chromium Controller or Chromium X (10x Genomics) for either poly-A primed mRNA capture using the Next GEM 3' v3.1 kit (fresh samples) or the Fixed RNA Profiling kit (FFPE samples). Reverse transcription and library preparation steps were performed in accordance with manufacturer protocols. Completed libraries were sequenced at the Cedars-Sinai Applied Genomics Core Facility.

## Processing of scRNA-seq datasets

To produce gene expression counts matrices, FASTQ reads were aligned and counted with the Cell Ranger Count pipeline (10x Genomics). In the case of mouse tumors, reads were aligned against a custom reference genome that included the sequence for the *dsRed* transgene expressed by the malignant cells; otherwise, default Cell Ranger references were used. For all further downstream processing, Seurat v4 was used[73].

## Counts normalization, quality control, and dimensionality reduction

Counts were normalized and variance-stabilized by *SCTransform* with method parameter 'glmGamPoi'. Dimensionality reduction, visualization, and initial clustering were performed by *RunPCA, RunTSNE, FindNeighbors*, and *FindClusters*. Differentially expressed genes were calculated using *FindAllMarkers*. For mouse only: sample identity was demultiplexed using *HTODemux*. Cells were labeled as dead, ambient, or otherwise failing QC if: percent mitochondrial reads greater than 15%, belonging to a cell cluster whose top 10 DE genes were mostly mitochondrial transcripts, number of genes detected <200 or >7500, or marked as "Doublet" by *HTODemux* (for hashtagged datasets only). Dimensionality reduction, clustering, and visualization were repeated on the filtered dataset.

## Labeling of CAF, tumor, and immune cell populations

Annotation of high-level cell types was performed based on key markers: Cells negative for *PTPRC, THY1*, and any positive markers below were assigned as malignant tumor cells; for mouse samples, cells positive for *dsRED* and negative for other markers were assigned as malignant tumor cells. Cells positive for *THY1* and *PDPN* and negative for *PTPRC* (or the mouse homologs) were labeled as CAFs. Genes that were significantly upregulated in the glyCAF cluster were used as input to enrichment analysis in g:profiler, and significantly enriched KEGG terms were ranked by adjusted p-value. All other cells were classified as one of the following cell types: endothelial (*PECAM1, CDH5*), pericyte

(*HIGD1B, COX4I2*), T (*CD3E*), NK (*NKG7*), B (*MHC-II* and *CD79A*) and plasma (*Ig, CD79A*), red blood cell (*HBA1*), DC/myeloid (*MHC-II, BATF3, ITGAM, ITGAX, CSF1R, CSF3R*). Further sub-clustering of DC/myeloid cell clusters was done manually following previously published signatures[51].

## Ligand-receptor prediction

*Ptprc*+ immune cells were identified from scRNA-seq of mouse UPS (p53^KO^Ccne1+) tumors, and subclustered using the *Seurat* R package (v4)[73]. Among immune cells, T cells were identified as clusters expressing *Cd3e*. The R package *ProjecTILs*[52] was used to infer CD4 or CD8 status of each T cell, aside from a cluster of cells corresponding to double-negative gamma-delta T cells (*n* = 12). CD8 T cells and the 4 CAF subclusters were analyzed for potential ligand-receptor interactions with the R package *CellChat*[53].

## Clinical samples and datasets

Sarcomas for single-cell RNA sequencing were collected at Cedars-Sinai Medical Center. The two frozen tissues were provided by the Cedars Sinai BioBank and Research Pathology Resource after receiving patient consent. No population data are available except in the case of uterine leiomyosarcoma (LMS), which was collected from a female subject. The two FFPE tissues were provided by the Department of Pathology at Cedar-Sinai as de-identified samples and do not qualify as human subject research. Bulk RNA-seq data for 251 patients derived from The Cancer Genome Atlas (TCGA) sarcoma dataset were queried in cBioPortal[74–76]; sarcoma patient samples were stratified either on the basis of *CCNE1*-high vs. unaltered expression (*n* = 16; 235) or *VGLL3*-high vs. unaltered expression (*n* = 8; 243), where RSEM z-score > 1.5 constituted high expression. Genes that were significantly upregulated in the expression-high vs unaltered groups were used as input to enrichment analysis in g:profiler, and significantly enriched GO:BP (Gene Ontology−Biological Process) terms were ranked by adjusted *p*-value.

## RNA extraction, PCR, and RT-qPCR

Total RNA was extracted from in vitro and ex vivo FACS-sorted cells by using TRIzol (Invitrogen) according to the manufacturer's instructions. RNA was directly reverse-transcribed using the High-Capacity cDNA Reverse Transcription Kit (Applied Biosystems) according to manufacturer instructions. 5−10 ng of RNA were used for each PCR reaction. Quantitative PCRs were carried out using Power SYBR Green PCR Master Mix (Applied Biosystems) and QuantStudio 3 real-time PCR system (Applied Biosystems).

Mouse Oligonucleotide sequences (qPCR):

*Ccne1*-Fw:TGAATACCCCAGGACTGCAT. *Ccne1*-Rv:AGGATGACGC TGCAGAAAGT. *Vgll3*-Fw: TGGGTCAGTAGTGGATGAACA. *Vgll3*-Rv:GCTGGTCCAAAAGGAAGTTG. *Slc2a1*-Fw: AGCAGAGGCTTGCTTG TAGAG. *Slc2a1*-Rv:GCCCGTCACCTTCTTGCT. *Nt5e*-Fw: GCAGCATTC CTGAAGATGCG. *Nt5e*-Rv:CTCCCGAGTTCCTGGGTAGA. *Pkm*-Fw: GCA GCGACTCGTCTTCACTT. *Pkm*-Rv:ATGGTTCCTGAAGTCCTCGG. *Pgk1*-Fw: CCACAGAAGGCTGGTGGATT. *Pgk1*-Rv:GTCTGCAACTTTAGCG CCTC. *Ldha*-Fw: CGTGCACTAGCGGTCTCAA. *Ldha*-Rv:TCCATGACGT CAACAAGGGC. *Cxcl16*-Fw:TCCTTTTCTTGTTGGCGCTG. *Cxcl16*-Rv: GG ACTGCAACTGGAACCTGATA

## Flow cytometry

Tumors were digested to single cell suspension enzymatically and filtered twice through 70µm filters. Red blood cells were lysed with ACK solution (Gibco), washed twice with Cell Staining Buffer (BioLegend), and then stained with the fluorophore-conjugated antibodies for 30 min at 4 °C. The excess of unbound antibodies was washed out before acquisition in flow cytometry. Cells were analyzed using ID700 Spectral Cell Analyzer (SONY) and sorted using FACS-ARIA III (BD, Pharmingen). The following antibodies were for flow cytometry: anti-

CD45 PAC (30-F11), anti-CD8b APC-Cy7 (YTS156.7.7), anti-CD4 PE(H129.19), anti-NK1.1 APC-Cy7 (PK136), anti-CD11b APC (M1/70), anti-Ly6c PE (HK1.4), anti F4/80 (BM8), anti-CXCR6 APC (SA051D1), anti-GranzymeB, anti-CD90 FITC (30-H12), anti-CD73 APC (TY/11.8), anti-PD-1 APC (RMP1-30) all purchased from BioLegend and used at 1:100 dilution. Anti-CD31-PE (eBioscience, 390, 1:100), anti-GLUT1 (Novus, Fgi.72, 1:100), anti-Cxcl16 (Bioss, BS-1441R, 1:50) and anti-Rabbit IgG AF488 (Invitrogen, A21245, 1:100) were used to identify CAF populations. Ghost Dye Red 710 (Tonbo, Cat #13-0871-T100) was used to exclude dead cells from the analysis.

## Western Blot

Protein lysates were prepared with ice-cold RIPA buffer (Boston Bio Products) supplemented with protease and phosphatase inhibitors (Roche). Protein lysates were separated using SDS-polyacrylamide gel electrophoresis (pre-cast gels, Thermo Fisher) and transferred to a nitrocellulose membrane. After blocking the membrane with 5% milk in PBST (PBS with 0.1% Tween-20, Sigma), the membranes were incubated overnight at 4 °C with anti-GLUT1(Novus, Fgi.72) and anti-Beta Actin (Bethyl, Cat #A300-485A) antibodies, diluted in PBST with 5% BSA to 1:1000. The blots were washed 3 times and then incubated with secondary antibodies (anti-Rabbit HRP, Thermo Fisher), diluted in PBST with 5% milk. Finally, the membranes were incubated with ECL substrate (Pierce) for 1 min and exposed for signal detection with the iBright Imager (Thermo Fisher).

## Tissue processing and multiplex immunohistochemistry

Immunofluorescence staining utilized tyramide signal amplification staining performed using OPAL Reagents (Akoya Biosciences, Inc, Marlborough, MA). Briefly, tumor tissues were formalin fixed and embedded in paraffin blocks and cut into 10 μm sections at the Cedars-Sinai Biobank and Research Pathology Core. Tissue sections were deparaffinized and rehydrated, then antigen retrieval was performed in Tris-EDTA or citrate buffer. Primary antibodies anti-CD8 (ebioscience, 4SM15), anti-CD4 (R&D, GK1.5), anti-CD90 (Sino Biological, Cat #50461-T44), anti-CD73 (Sino Biological, Cat # 50231-T56), anti-GranzymeB (ebioscience, 16G6), anti-CD90.1 (BioLegend, OX-7), anti-CD45.1 (Invitrogen, A20) diluted to 1:200 were incubated overnight at 4°C. Secondary antibodies conjugated to HRP-polymers (Abcam, Cambridge, UK) were incubated for 15 min and then washed before slides were incubated with OPAL fluorophores (OPAL 520, OPAL, 650, OPAL 570) diluted 1:200 for 10 min at RT. Slides were washed in PBST before performing subsequent rounds of antigen retrieval and staining. Tissues were mounted with ProLong Gold Antifade Mountant with DNA Stain DAPI (Invitrogen). Images were acquired on ECHO Revolution upright microscope equipped with sCMOS Mono camera.

## Image quantification

Images were acquired at 10x or 20x magnification and analyzed using HALO (Indica Labs Inc., Albuquerque, NM). Regions of interest (ROIs) were manually determined which encompass the tumor margin or the inner parenchyma. 3–4 ROIs per tumor selection were quantified. CD4$^+$ and CD8$^+$ T cells were defined as CD4 or CD8 positive with DAPI positive co-staining and quantified within each ROI. For CAF quantifications, single cells were segmented on DAPI$^+$ cells, and CD73$^+$ CAF were assigned as CD3$^-$CD90$^+$CD73$^+$ cells, while CD73$^-$ CAF were defined by CD3$^-$CD90$^+$CD73$^-$ cells using the HALO HighPlexFL module. ROIs encompassing the tumor margin, -1.5 mm × 1.5 mm were then used to quantify the CD73$^+$ and CD73$^-$ CAF.

## 2-NBDG glucose uptake assay

2-NBDG (2-(N-(7-Nitrobenz-2-oxa-1,3-diazol-4-yl)Amino)−2-Deoxyglucose) (Invitrogen), was diluted in sterile PBS to 5 μg/mL. For in vivo glucose uptake assays, 100 μg of 2-NBDG was injected intravenously via tail vein 30 min prior to sacrificing tumor-bearing mice. For in vitro experiments, cells were incubated with 15 uM of 2-NBDG for 30 min, washed with 1× PBS to remove free 2-NBDG, and then analyzed by flow cytometry.

## Seahorse metabolic profiling

5000 CAF were seeded in an Agilent XFe/XF96 well plate 24 h prior to the assay. The Seahorse Glycolysis Stress Test (Agilent Technologies, #103017-100) was performed in assay media according to the manufacturers protocol. Data was acquired using Seahorse XFe/XF96 Analyzer and analysis was performed by Seahorse XF Stress Test Report Generator and Wave (Agilent Technologies).

## Generation of CAF cell-lines and Bone Marrow Derived Macrophages (BMDM)

Scaffolds containing p53$^{KO}$Ccne1$^+$ tumor cells were transplanted subcutaneously into in p53$^{KO}$ mice to generate sarcoma tumors. Tumors were dissociated and immortalized p53$^{KO}$ CAF were FACS sorted as follows: glyCAF (Live, dsRED$^-$, CD31$^-$, CD45$^-$ CD90$^+$ CD73$^+$ cells) and non-glyCAF (Live, dsRED$^-$, CD31$^-$, CD45$^-$ CD90$^+$ CD73$^-$ cells). Individual cells were isolated by serial dilution, expanded, and then validated for CAF markers by positive expression of CD90 and lack of tumor dsRED by flow cytometry. The sorted cells were maintained in DMEM supplemented with glutamine, 10% fetal bovine serum, 100IU/ml penicillin and 100 μg/ml streptomycin (Gibco) and maintained at 37 °C and 5% CO$_2$. Bone marrow-derived macrophages were generated by isolating total hematopoietic cells from mouse bone marrow and stimulated in vitro with recombinant mouse M-CSF (BioLegend) for 6 days. New M-CSF was added every other day with fresh medium.

## Transwell T-cell migration assay

$1.5 × 10^5$ Ccne1$^+$ tumor cells were seeded in the bottom chamber containing complete T cell media (RPMI 1640, 10% FBS, 1% Pen-Strep, 1% MEM-NEAA, 1 mM Sodium Pyruvate, 50 μM β-Mercaptoethanol, 50IU/mL recombinant mouse IL-2 (BioLegend) with 10 ng/mL of recombinant CXCL10 (BioLegend) in a 24-well plate containing polystyrene transwell membranes with 3 μm pores. $0.75 × 10^5$ CAF, BMDM, or Ccne1$^+$ tumor cells were seeded in the upper chamber and allowed to adhere overnight. CD8$^+$ splenocytes were isolated from sex-matched naïve (non-tumor bearing) C57BL/6, Cxcr6$^{wt/wt}$ OT-1, or Cxcr6$^{-/-}$ OT-1 mice using Easy Sep Mouse CD8$^+$ T-Cell Isolation Kit (STEMCELL Technologies Inc., Cambridge, MA) and activated using plate bound anti-CD3/anti-CD28 (BioLegend) for 24 h. $2 × 10^5$ activated CD8$^+$ T cells were seeded on top of the CAF monolayer in the top chamber. In certain cases, GLUT1 inhibitor (BAY-876, 75 μM) was added to the culture system. After 24 h, the number of migrated CD8$^+$ T cells in the bottom chamber was counted and the migration index was calculated by dividing the number of migrated cells in a sample divided by the number of cells migrated in control conditions.

## Quantification and statistical analysis

All statistical analyses were performed using GraphPad Prism 9 software (GraphPad, San Diego, USA). A two-tailed unpaired Student's *t* test was used for comparisons between two groups. When comparing multiple groups, one-way ANOVA with Tukey's multiple comparison test was utilized. Differences in tumor growth were evaluated by two-way ANOVA with Tukey's multiple comparison test. Survival curves were generated with an endpoint when a tumor reached 1000 mm$^3$. Survival comparisons were performed using a Log Rank test. In all graphs, each symbol represents an individual sample, and the error bars represent the mean ± standard error of mean. Exact *p*-values are shown for all analyses.

## Reporting summary

Further information on research design is available in the Nature Portfolio Reporting Summary linked to this article.

## Data availability

The human bulk RNA-sequencing data was queried using cBioPortal (https://www.cbioportal.org/, cohort TCGA SARC) and can be downloaded from the TCGA data portal (https://portal.gdc.cancer.gov/).The single-cell RNA-sequencing data generated in this study have been deposited in the NCBI Gene Expression Omnibus database under accession code GSE237638. The remaining data are available within the Article, Supplementary Information or Source Data file. Source data are provided with this paper.

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

## Acknowledgements

J.G. was supported by the K99/R00 grant (CA212200) and R01 (CA258265) for the execution of this work. Additionally, J.G. was supported by grants from the Sarcoma Foundation of America (Grant 2019 SFA 15–19), and the Cedars-Sinai Cancer Accelerator Award. M.B. was supported by a National Cancer Institute F31 fellowship (1F31CA284888). K.S.C was supported by R01CA175397, R01CA175397-Supplement,

U54CA274375. X.P.H received funding from U54CA274375. A.W.L. received support from the Cancer Research Institute (Lloyd J. Old STAR Award). A.K. was supported by the National Heart, Lung and Blood Institute (R00 HL141702) and the Leukemia Research Foundation New Investigator Award (Grant No. 941997). We acknowledge the Cedars Sinai Flow Cytometry Core, Cedars Sinai Biobank and Research Pathology Resource, and the Cedars Sinai Applied Genomics, Computation and Translation Core for assisting with this work. Schematic representations were Created with BioRender.com.

## Author contributions

J.G. and M.B. conceptualized the study and designed and performed in vivo experiments. J.G. supervised the study. J.G., M.B., and E.K. wrote and edited the manuscript. M.B., K.I., J.X., R.P., B.G., and F.H.G.T., J.X., and J.G. performed in vitro assays and cloning, and designed and produced original reagents. E.K. analyzed transcriptomic data and conducted scRNA-seq. M.B., K.I., F.H.G.T., and X.P.H. conducted flow cytometry and imaging experiments. A.W.L. provided transgenic mice for adoptive transfer experiments. A.K. conducted metabolomic studies. M.D.S. assisted with single-cell sequencing experiments. K.S.C. and S.O. provided valuable commentary, discussion, and reviewed the manuscript.

## Competing interests

The authors declare no competing interests.
