## [Peer Review File · Nature Communications]

Metabolic targeting of cancer associated fibroblasts
overcomes T-cell exclusion and chemoresistance in soft-tissue
sarcomasREVIEWER COMMENTS

Reviewer #1 (Remarks to the Author): with expertise in CAFs, scRNAseq

In this study the authors present a comprehensive analysis of stromal compartment in soft tissue sarcomas that has a direct bearing on response to immunotherapy. Specifically, using mouse models of immune excluded and infiltrated STS phenotypes along with distinguishing CAFs from tumor cells, the authors show that immune exclusion is not linked to differences in CAF proportions. Instead, they identify a specific CAF subtype that switches on a glycolytic program, preventing T cell migration into tumor beds. This is an important finding and would merit publication after the authors have addressed the following comments:

Major Comments:

1. Apart from traditional differential gene expression methods, can the authors use topic modeling approaches such as cNMF to demonstrate that glycolysis program is upregulated specifically in glyCAF. In Fig 2H mCAF and apCAF also upregulate genes implicated in glycolysis, albeit to a lesser extent. Can the authors speculate on why glyCAF upregulate this program as opposed to a traditional myCAF (that is also found in the periphery of tumor nests in desmoplastic cancers such as PDAC)? Do they expect a continuum of CAF states that lie between myCAF and glyCAF? And what are the putative extrinsic signals that drive this transition in STS?
2. Can you please show whether the mCAF and glyCAF are driven by TGF β signature in STS (ref. Mariathasan et al., Nature 2018; Dominguez & Muller et al., Can Discov 2020)? In other desmoplastic indications mCAF are TGF β driven and it would be worthwhile comparing these signatures in STS.
3. While this study is restricted to STS, it would be interesting to check whether glyCAF are found in other indications. Recent papers in solid cancers (Xue et al Front Pharmacol. 2022, Chen et al., Front Immunol. 2022) describe NT5E or CD73 expressing CAFs to be implicated

in creating an immunosuppressive environment but do not allude to upregulation of glycolysis. Can the authors comment on why this specific subset emerges in the context of STS?

4. Do mCAFs in STS secrete collagen and deposit ECM? If so, how does GLUT1 inhibition help loosen up the stroma barrier for T cell entry? Can the authors demonstrate the spatial distribution of glyCAFs, mCAFs and T cells in an immune excluded setting?

Minor comments:

Fig 2a: please mark CAFs and tumor clusters clearly (similar to the annotations in ED Fig 2a) for ease in reading.

Reviewer #2 (Remarks to the Author): with expertise in sarcomas

Thank you for the opportunity to review this tour-de-force of sarcoma immunology. The finding that specific sarcoma CAFs may be manipulated to change the TME and consequently leverage the efficacy of doxorubicin is very exciting and clinically relevant.

There are certainly compelling aspects to these data, but many relevant criticisms are listed below in the order presented. Minor weaknesses are a consistent lack of detail. The evaluation of only two human soft tissue sarcomas for comparison is a moderate weakness, as is the use of only female animals and the subcutaneous model. More significant weaknesses are the essentially neglecting the importance of metastatic sarcoma metastases and the lack of addressing any limitations of the work. Specific comments are below.

Review: Cancer-associated fibroblasts metabolic targeting overcomes T-cell exclusion and chemoresistance

Introduction

- Bottom of page 3/top of page 4. Please expand a little more on the limitations of current

treatment and prognoses. Local recurrence at 2 years post-op is around 20% globally, and the incidence of metastatic disease is basically a coin-toss (~50%). Highlighting these realities will help to illustrate the importance of novel treatments.

Results

- Please consider mentioning on page 5 that the p53KO mice are indeed immune competent, and explain why only female mice were used or list it as a limitation of the study.
- Bottom of page 5. What was the metastatic phenotype of the p53KOCcne1+ and p53KOVgll3+ cells? This is no small point, as metastatic disease is ultimately responsible for the most sarcoma mortality.
- Please be more precise than “slight increased tumor growth” on the bottom of page 5. This is pretty subjective. Please give a % difference or a non-significant p value or ideally both.
- What was the doxorubicin response of the p53KOCcne1+ animals? The robust response of the other animals is mentioned by nor these.
- The confirmatory human tumors for the CD 90 analysis on page 7: MFS, LMS, UPS. These were single sarcomas or microarrays of these histologic subtypes? I have terrific sympathy for the challenge of finding solid numbers of such rare histologies, but 2 or 3 cases is not very compelling even in STS.
- Page 7-Similar to the above comment, please be more subjective and granular than “negligible” (...negligible differences across the two models...).

Discussion

- Page 13. The absence of a discussion of limitations is conspicuous.

Methods

- Page 15. Were other orthotopic sites (intramuscular, for example) considered? Certainly sarcomas superficial to the fascia occur, but disease deep to the fascia is much more common. Ostensibly the changes in oxygen tension between subcutaneous and subfascial locations have implications for this study.

Reviewer #3 (Remarks to the Author): with expertise in CAFs

In this manuscript, Broz et al. identify a subset of CAFs in soft-tissue sarcoma (STS) that express glycolysis-related genes and display increased glycolytic activity, termed glyCAFs. Increased glyCAFs was associated with reduced CD8+ T cell infiltration, which could be reversed with administration of a GLUT1 inhibitor (GLUT1i). Increased glycolysis in CAFs has been previously reported in other cancer types (Yang et al. IJOS 2021, Zhang et al. Cell Reports 2015, Becker et al. Cell Reports 2020) and this manuscript links such metabolic reprogramming with the immune microenvironment. Several points should be addressed, as outlined below.

1. In Fig. 2, immune cells and other non-CAFs and cancer cells are analyzed ('Immune/other'). It would be more informative to have more in depth separation of this subset of cells, as the overlap in expression with CAFs varies depending on the type of immune/stromal cell, i.e. Thy1 is expressed by T cells but not myeloid cells.
2. The authors refer to Thy1/CD90 as a pan-CAF marker; however, the data as currently presented only shows enrichment of Thy1 in CAFs. The percentage of CAFs that express Thy1 should also be included to further strengthen its use as a pan-CAF marker.
3. Doxorubicin in combination with GLUTi treatment is associated with increased CD8+ Granzyme B+ cells, but whether GLUTi alone induces T cell activation should be evaluated. Similarly, the abundance of CD4+ T cell subsets, e.g., FoxP3+ Tregs vs. FoxP3 Teff, should be determined in response to doxorubicin and GLUT1i alone and in combination.
4. GLUT1i alters the abundance of CD73+CD90+ cells, which the authors term as 'glyCAFs'. Both CD73 and CD90 are also expressed by T cells; thus, the stains in Fig. 3b need to include an exclusion marker (CD45, CD3, etc.) to specifically identify CAFs. In addition, the authors should clarify if doxorubicin alone or in combination with GLUT1i also impacts glyCAF abundance.
5. While depletion of CD8+ T cells promotes tumor growth in the context of doxorubicin and GLUT1i, it does not completely reverse growth inhibition. The authors should clarify if CD4+

T cells are also important, or if there is an additional unknown mechanism by which GLUT1i acts.

6. In Fig 5, authors are recommended to explain which cell type the GLUT1i target. The authors have shown us reprogramming the glycolytic properties of CAFs by inhibiting GLUT1 led to increased infiltration of cytotoxic CD8+ T cells into the tumor, independent of cancer cell-intrinsic metabolism, but the metabolic markers used for glyCAF like Slc2a1, Pgk1, Pkm, Pgam1, Hk2, Nt5e, and the upregulated glycolysis pathways may not only exist in glyCAF and cancer cells and such cells could be impacted by GLUT1i. A scRNA-seq feature or violin plot showing specificity of the glyCAF marker genes would further support that the phenotypes observed are specific to CAFs and not present in other cell subsets.

7. In the abstract, the authors state, "Targeting glycolysis in glyCAF enhances T-cell infiltration and augments the efficacy of chemotherapy." However, the results do not specifically support the assertion that glyCAF, and not other non-cancer cell types with glycolytic features, that enhance T-cell infiltration and improve chemotherapy efficacy. The abstract should be revised accordingly.

8. In Fig 2h, authors are recommended to perform enrichment or quantitative metabolic prediction analysis based on scRNA-seq results to substantiate the metabolic signature of glyCAFs.

9. In Fig 4d and Extended Data Fig 5, it would be helpful to compare glyCAFs to other cell types in terms of communication strength with CD8/CD4 T cells and differences in ligand-receptor pairs. The current results don't exclude the potential effects on T cells by cell types other than glyCAFs. These other cells, also present in tumor tissues and characterized by high metabolic features, could be inhibited by GLUT1i.

Minor comments

1. Introduction, second paragraph: citations 8 and 9 do not include targeting of pan-CAFs, aSMA+ CAFs (marking a subset of CAFs) and type I collagen produced by aSMA CAFs are targeted. The text should be revised accordingly.

2. 'Tumor' and 'tumor cells' should presumably be 'cancer cells', as tumor is a more general term that includes all cells that would be present in a tumor.

3. Page 7: the text states that dsRed, CD45, and CD31 are used as exclusion markers; however, CD31 is not included in the schematic in Fig. 2 or the gating strategy in Extended Data Fig. 3a. Please clarify. In addition, it is not clear if the analysis in Fig. 2i also includes exclusion of dsRed and CD45 expressing cells.

Reviewer #4 (Remarks to the Author): with expertise in sarcomas

In this manuscript by Broz et al., the authors investigate the role of cancer-associated fibroblast (CAFs) in dictating immune response and chemosensitivity for CCNE+ soft tissue sarcomas, which have low immune infiltration compared to Vgll+ STS. Through comprehensive scRNA-seq of fibroblasts associated with mouse and human STS, they elucidated the expression programs of CAFs. Specifically, they have identified a subpopulation of CAFs with altered glycolytic activity, which are known as glyCAFs. The authors provide evidence that glyCAFs are associated with immune exclusion for STS and reside with the CD8 T cells at the periphery/margin of the tumor. Through the use of genetic and pharmacological approaches, they demonstrate that the glyCAFs significantly contribute to immune exclusion and that when used in combination with chemotherapy, such as doxorubicin, targeting the glycolytic aspects of the tumor (supposedly glyCAFs) can lead to enhanced immune infiltration and better anti-tumor activity. While the authors provide some insightful studies and findings, there remain some deficiencies with regards to the models and mechanisms. A couple of the more significant issues center around (1) why are CCNE+ tumors associated with more glyCAFs (is the tumor secreting factors that are recruiting these glyCAFs) and (2) the mechanisms by which the glyCAFs (the cell-cell communication data is intriguing, but underdeveloped and not conclusive) are excluding CD8 T cells in this genetically defined population of STS. Other concerns and critiques are noted below.

Minor comments:

1. Abrupt transition from mention of fibrotic cell numbers to scRNA seq re: CAF subtypes.

Page 7, paragraph 2.

2. It is not clear to this reviewer: Is the identification of the glyCAF novel or has it been previously identified? This reviewer would conclude that this is a novel description since there does not seem to be a reference for these glyCAFs provided as was provided for the other CAF subpopulations mentioned? Just want to make sure this is correct.

3. Fig. 5F. Difficult to believe that DOX alone provided a statistically significant advantage? 2-3 days? Had enough animals/statistical power to make that conclusion?

More significant comments:

1. Cross-comparison analysis of the models/tumors: How do the molecular signatures of the MSC-transformed cells compare to human tumors with respective genetic traits. It was not presented early on how truly representative the genetically engineered MSC models compare to human tumors with these genetic amplifications. Are there potentially other genetic features associated with CCNE+ tumors that are contributing to the immune profiles in human patient tumors?

2. Besides T cells, other immune profiles altered in the CCNE+ tumors?

3. Fig 1: For pathway analysis, what are the negative normalized enrichment scores. For (A), how did the authors determine low immune infiltration for CCNE+ based on this pathway analysis...no indication. Unless a negative NES? Fig 1b: the influence of the 3D scaffold on immune infiltration? Has that been controlled for? Fig 1h: Vgl3+ tumors still grow and become resistant to Dox. How do the T cell populations change over time in response to Dox?

4. Page 7, related to Fig 2: The authors mention that the distance of fibrotic stroma to invasive margin and then provide rationale that number of fibrotic cells does not explain the CD8 distribution. The description for the "criteria" is confusing and the rationale/justification should be mentioned for using this criteria as the determination for

immune exclusion deserves further explanation.

5. Fig 3b, appears that Glut1 inh eliminates the glyCAFs. Total CAFs increase with Glut1 inh...how do the other CAF populations change?

6. Additional characterization of the glyCAFs would be beneficial: For example, in vitro comparison of glycolysis properties for glyCAF vs other CAFs through the use of Seahorse metabolic profiling? Quantification of lactate levels? Any changes in mitochondrial properties/functions for these glyCAF cells?

7. What is the effect of GLUT1 inhibitor on the T cells functions/properties? Further characterization of the effect of the Glut1 inhibitor on the other tumor cell constituents/properties would be beneficial.

8. Any plasticity for the CAF signatures/properties? Can a glyCAF change to "non-glyCAF" in culture?--Since they do expand these cells in vitro (page 9, for example in fig 4). Also vice versa, can non-glyCAF gain properties of glyCAF?

9. Besides preventing migration of the T cells to the tumor cells in the transwell, are the glyCAFs altering other phenotypic functions of the T cells? Viability, enzymatic release, etc...?

10. Bottom of page 10: When describing the predicted cell-cell communications, it is unclear what the authors are concluding re: T cell activation and properties. They discuss several possibilities, including CXCL16-CXCR6 interactions between the glyCAFs and T cells, but nothing is definitively proven. Need to further define/prove their statements. Also, for Fig 4D: Seems to be redundancy in signaling cell-cell communications as both glyCAF and apCAF have similar intensity signaling networks, while mCAF and iCAF also do but at lower intensity. Since these cell-cell networks are not unique to glyCAF why would they explain the noted changes in CD8 cell infiltration? Also, upon inhibition of the glyCAF with Glut1, how do the signaling networks change for the other CAFs?

11. Fig 5i: Does Glut1 inhibition alone increase T cell infiltration into tumor? The authors show CD8 cell staining for DOX and then DOX+Glut1 inh, but not Glut1 alone. Please show.

12. Fig 5i: When were these tumors taken? At end of study? As the tumor is growing/becoming resistant to the combo therapy (day 34 or so), do the T cells revert to being at the margin again?

13. Prove the role of Doxorubicin in their models. Does Doxo only treatment increase antigen presentation in vivo model employed?

14. In discussion mention acidification of TME due to glyCAFs. Is there any evidence for this in vivo? Can you stain for CA9/12 on the tumors? Also why do the authors think acidification only affects migration. What about viability, other properties for T cells? Please discuss.

Reviewer #5 (Remarks to the Author): with expertise in sarcomas

Broz et al have submitted a compelling manuscript that argues the existence of sarcoma associated fibroblasts akin to those that exist in epithelial tumors known as cancer associated fibroblasts. This concept is novel and potentially very interesting. The authors have also identified a subset of fibroblasts that govern T cell recruitment to the tumor parenchyma and subsequent responsiveness to chemotherapy. However there are some textual and experimental/mechanistic clarifications necessary before the manuscript would be ready for publication. The following is a figure by figure analysis of the primary manuscript:

Figure 1: The models. The authors have generated a new system for syngeneic transplantation of transformed and fluorescently labeled mesenchymal stem cells into host mice. Aside from one H&E imaging the extended figures there is no extensive validation that the models do indeed recapitulate either existing murine models of undifferentiated pleomorphic sarcoma (UPS) or human UPS. Ideally the authors would have certified veterinary and human clinical pathology specialists evaluate the histological images and

provide classification. This requirement is typical of any new model system. Moreover, the authors should provide comparative transcriptomics of their models relative to human and mouse UPS as can be found in this reference (Cross species genomic analysis identifies a mouse model as undifferentiated pleomorphic sarcoma/malignant fibrous histiocytoma by Mito et al 2009). It's critical that the authors demonstrate that their models are in fact representative of UPS before drawing additional conclusions.

Figure 2. Validation of CAFs in sarcoma. The authors provide extensive scRNA-seq gene expression data, as well as limited flow and imaging data and conclude that a small population of CAFs exist in at least some sarcomas. As this claim is quite substantial and important functional validation is necessary. Using the FACS sorted CAFs vs other tumor cell populations the authors should perform collagen contraction assays. These well established assays confirm functional fibroblast activity.

Figure 3: Glut1 inhibition recruits T cells to tumor parenchyma. The authors show that Glut1 knockdown in tumor cells has no effect on T cell recruitment whereas Glut1 inhibition in tumors leads to increased T cell recruitment in the parenchyma. However, UPS tumors are comprised of many immune cells including macrophages etc as well as tumor cells. The authors do not rule out the effect of Glut1 on these cells when concluding that CAFs are responsible for T cell recruitment. This is not sufficient. Large macrophage/dendritic populations should be ruled out in addition to tumor cells before drawing this conclusion.

Figure 5: Chemotherapy in UPS. The authors should correct text that indicates chemotherapy is standard first line treatment in UPS. This is often not the case. First line Radiation and surgery are primarily used to treat UPS with adjuvant chemotherapy sometimes being added in advanced disease with mixed results. As the authors have effectively demonstrated chemotherapy including doxorubicin is ineffective as first line therapy against UPS. The authors conclusions are important in spite of this error in rationale. The ability to sensitize UPS cells to chemotherapy using Glut1 is an important discovery but should be framed in the text as such.

REVIEWER COMMENTS

Reviewer #1 (Remarks to the Author): with expertise in CAFs, scRNAseq

In this study the authors present a comprehensive analysis of stromal compartment in soft tissue sarcomas that has a direct bearing on response to immunotherapy. Specifically, using mouse models of immune excluded and infiltrated STS phenotypes along with distinguishing CAFs from tumor cells, the authors show that immune exclusion is not linked to differences in CAF proportions. Instead, they identify a specific CAF subtype that switches on a glycolytic program, preventing T cell migration into tumor beds. This is an important finding and would merit publication after the authors have addressed the following comments:

We express our gratitude to the reviewer for providing valuable and constructive suggestions that significantly enhanced the robustness of our study's findings. In response to the insightful recommendations made by the reviewer, we performed additional experiments, which have played a crucial role in improving our comprehension of glyCAF's features and elucidating the molecular mechanisms through which glyCAFs mediate the exclusion of T cells from the inner tumor parenchyma.

Major Comments:

1. Apart from traditional differential gene expression methods, can the authors use topic modeling approaches such as cNMF to demonstrate that glycolysis program is upregulated specifically in glyCAFs. In Fig 2H mCAFs and apCAFs also upregulate genes implicated in glycolysis, albeit to a lesser extent. Can the authors speculate on why glyCAFs upregulate this program as opposed to a traditional myCAF (that is also found in the periphery of tumor nests in desmoplastic cancers such as PDAC)? Do they expect a continuum of CAF states that lie between myCAFs and glyCAFs? And what are the putative extrinsic signals that drive this transition in STS?

We thank the reviewer for suggesting the use of topic modeling approaches. Accordingly, topic modeling with the fastTopics package^{1,2} similarly identified topics that approximated the 4 CAF programs, with topic 1 marked by genes related to glycolysis, topic 2 related to antigen presentation, topic 4 related to Hypoxia (*Car3*, *Hp*), topic 5 to matrix remodeling (*Tnn*, *Tnmd*, *Postn*), topic 6 to genes associated with iCAFs (*Pi16*, *Clec3b*). Topic 3 included several noncoding genes associated with tissue dissociation-related stress (*Gm42418*, *Malat1*), but it was not the dominant topic in any one CAF subtype.

Proportionally, each CAF subtype demonstrated strongest use of one of the topics (**Figure R1A**). The glycolysis-associated topic 1 exhibited nearly exclusive use by glyCAFs but was present in a very small number of cells in the iCAF and mCAF populations. Interestingly, glyCAFs exhibited trivial use of topic 6, which was found predominantly in iCAF and to lesser degree in mCAF and apCAF. Notably, performing PCA and re-clustering of the CAFs using topic strengths only, four CAF clusters emerged which largely recapitulated the existing clustering (**Figure R1B**)

Figure R1. A, Topic usage plot across the 4 CAF programs. **B,** t-SNE of CAF re-clustered using topic strengths (left) and annotated CAF clusters (right)

Moreover, based on existing literature data, we speculate that glyCAFs might partially overlap with what have been described as myCAF in PDAC and other tumors, because the glyCAFs express the highest levels of myCAF genes such as *Acta2*, *Tagln*, and *Myf9*—although not exclusively, with the mCAFs in this study also expressing the myCAF genes at lower levels (**Extended Data Figure 3c**).

Interestingly, the pathway analysis of Elyada 2019 of CAF subtypes demonstrated that myCAFs in PDAC are not especially glycolytic compared to other CAF subtypes. This was a remarkable difference from what we observe in sarcoma, given the similarities between sarcoma and PDAC such as a fibrotic phenotype as well as similar overall levels of glycolytic activity compared to other tumors³.

We invite the reviewer to continue on to our response to Comment 3, where we extend the comparison between the glyCAFs in sarcoma and CAF subtypes in other cancers including PDAC. We believe that further study is required to determine whether myCAFs indeed contribute to the glyCAF population, and if so, why they might be especially primed to do so. Perhaps *in vivo* lineage-tracing based experiments could be better suited for finding conclusive answers to these questions.

2. Can you please show whether the mCAFs and glyCAFs are driven by TGF β signature in STS (ref. Mariathasan et al., Nature 2018; Dominguez & Muller et al., Can Discov 2020)? In other desmoplastic indications mCAFs are TGF β driven and it would be worthwhile comparing these signatures in STS.

This is an interesting suggestion. In this respect, we observed that glyCAF and to a lesser extent, mCAF, are driven by the TGF β signature outlined in Mariathasan et al., Nature 2018. TGF β signaling in multiple cell types has been shown to upregulate glycolysis machinery such as the glucose transporter (GLUT1) and Hexokinase 2^{4,5}. Thus, we may speculate that glyCAF receive higher TGF β stimulation compared to other CAF subtypes including mCAF, which may trigger the glycolysis preferentially in glyCAF. Expression of the TGF β response signature genes is plotted individually and in aggregate below (**Figure R2**).

Figure R2: A, Dotplot showing the genes identified in the Pan-Fibroblast TGF β Response signature (F-TBRS, Marthasian et al.) (left) and TGF β pathway genes (right) projected across 4 CAF subsets. **B,** Violin plot of the aggregated TGF β pathway and F-TBRS signature.

These results prompted us to assess the impact of TGF β on CAFs *in vitro*. Interestingly, we observed that treating non-glyCAFs with recombinant TGF β induces the expression of *Nt5e* (CD73) and *Glut1* (**Figure R3A**). On the contrary, treating glyCAFs with anti-TGF β reduces the expression of *Nt5e*, *Glut1*, and *Pgk1* (**Figure R3B**), suggesting that TGF β signaling might be responsible for the induction of CAFs glycolytic properties.

Figure R3: A, RT-qPCR of the glycolytic signature in non-glyCAF treated with recombinant TGFβ (5ng/mL) for 48h. **B**, RT-qPCR of the glycolytic signature in glyCAF treated with anti-IgG or TGFβ neutralizing antibody (30ng/mL) for 48h.

While these findings are intriguing and potentially relevant to understanding how glyCAFs accumulate in Ccne1+ tumors compared to Vgll3+ tumors, we consider them preliminary at this stage, and we think that further investigations are warranted. Consequently, we have not included these data in the revised manuscript but are willing to do so upon the reviewer's request.

3. While this study is restricted to STS, it would be interesting to check whether glyCAFs are found in other indications. Recent papers in solid cancers (Xue et al Front Pharmacol. 2022, Chen et al., Front Immunol. 2022) describe NT5E or CD73 expressing CAFs to be implicated in creating an immunosuppressive environment but do not allude to upregulation of glycolysis. Can the authors comment on why this specific subset emerges in the context of STS?

We thank the reviewer for this question. It would indeed be important to know whether glyCAFs emerge in any cancers other than sarcomas. Since the submission of our paper, a new study by Cords et al.⁶ published in 2023 put forward a transcriptomic and proteomic characterization of CAF subtypes, first in breast cancer and then validated in multiple cancer datasets (colon, lung, HNSCC, PDAC). Not surprisingly, different cancers were found to contain unique proportions of CAF subtypes, with iCAFs and mCAFs predominating in PDAC, for example.

The average expression in the sarcoma CAF subtypes of the top 40 marker genes of the relevant Cords et al. subtypes is shown below. Accordingly, we found that the tCAF gene signature reported in Cords et al. was highest in our glyCAFs (**Figure R4**).

Figure R4. Signatures of human iCAF, mCAF, apCAF, and tCAF as identified by Cords et al., 2023 projected onto the murine CAF clusters identified in this manuscript.

Interestingly, CD73 (NT5E) was used as a marker for tCAFs in the breast cancer study, and pathway analysis demonstrated that the glycolysis was significantly enriched in tCAFs but no other CAFs. Additionally, the breast cancer study also found upregulation of glycolysis in CD73+ CAFs. The work was descriptive, however, and did not functionally demonstrate any function for these CAFs in terms of immune exclusion or suppression. Thus, to our knowledge, our manuscript would be the first to functionally describe immune exclusion driven by a highly glycolytic, CD73+ CAF population.

Despite this, we caution that until further study, CD73/NT5E is currently a marker and not necessarily an integral part of the glycolytic phenotype. Indeed, while the glycolytic “tCAF” population was found in multiple cancers by Cords et al.’s validation scRNA-seq dataset, NT5E was expressed lowly and non-differentially by tCAFs in cancers other than breast (**Figure R5**) suggesting that CD73 might serve as a marker of glycolytic CAFs in breast cancer and sarcoma, but not all cancers.

Figure R5. NT5E (encoding CD73) expression across the CAF clusters in the Cords et al. validation cohort.

4. Do mCAFs in STS secrete collagen and deposit ECM? If so, how does GLUT1 inhibition help loosen up the stroma barrier for T cell entry? Can the authors demonstrate the spatial distribution of glyCAFs, mCAFs and T cells in an immune excluded setting?

We value the insightful comment on the role of mCAFs in our observed phenotype. At the transcriptional level, we demonstrate distinct expression patterns of collagens (Col1a1, Col3a1, Col12a1), extracellular matrix proteins, and glycoproteins (Postn, Tnc) by mCAFs (**Fig. 3e,f**). However, all the components that are enriched in the mCAFs are intracellular molecules, necessitating experimental assessment at the protein level through cell fixation and intracellular staining—methods incompatible with FACS-sorting and functional assays. In the attempt to isolate mCAFs (CD34⁻CD73⁻CD74⁺), we isolated cells as the negative fraction

upon exclusion of iCAFs (CD34⁺), glyCAFs (CD73⁺), and apCAFs (CD74⁺), however we were not able to identify any CD74⁺ apCAF from the *in vitro* cultured cells. Subsequently, we investigated the expression of selected collagens and ECM proteins informed by the scRNAseq analysis (COL1A1, FN1, and POSTN) via western blotting in the enriched glyCAFs, iCAFs and mCAFs. Unfortunately, the used antibodies (Col1a1 (CST Cat #72026), FN1 (CST Cat #63779), and Postn (Abcam Cat #(ab152099)) produced no signals or multiple bands of varying sizes, hindering conclusive results.

Figure R6. Western blot of **A**, COL1A1 **B**, FN1 (Fibronectin1), and **C**, POSTN (Periostin) in glyCAF (CD73⁺) iCAF (CD34⁺) and mCAF (CD34⁻CD73⁻ CD74⁻).

Identifying mCAF by IHC/IF has also proven challenging since many mCAF enriched markers observed at the transcriptional level are also expressed in other cells, albeit to a lower level (**Fig. 3f**). Accordingly, semi-quantitative approaches such as IHC/IF are challenging in this context and likely non-reliable. However, we found that CD73 is a reliable marker for the glyCAF; indeed, we proved that FACS sorted CD73⁺ CD90⁺ glyCAF showed a consistent and significant glycolytic phenotype, if compared to the CD73⁻ CD90⁺ non-glyCAF cells (**Extended Data Fig. 3d,e**). Therefore, we argue that assessing the spatial positioning of the CD73⁻ CD90⁺ non-glyCAF can give us insight about the proximity of the iCAF, mCAF, and apCAF to CD8⁺ T cells in comparison to the CD73⁺ glyCAFs, although we cannot individually discriminate between the non-glyCAF populations. Following this line, we measured the distance from CD73⁺ CD90⁺ glyCAF to the nearest CD8⁺ T cell and observed that it is shorter than the distance from CD73⁻ CD90⁺ non-glyCAF to the nearest CD8⁺ T cell, suggesting that there is a preferential location of glyCAF near CD8⁺ T cells (**Fig. 4m**). While we acknowledge the need of further exploration to comprehend the spatial and functional distinction between the different CAF populations, especially mCAFs and glyCAFs, the available reagents do not currently favor this analysis. Thus, generating ad hoc reagents and assays will be critical and pursued in future research plans. We are addressing this important point in the discussion (**page 16**).

To assess if the stromal barrier changes under GLUT1 inhibition, we performed Masson's Trichrome staining in which collagen is stained in blue and nuclei are stained dark brown (**Figure R7**). The stromal barrier identified by blue-stained collagen did not exhibit any structural changes following GLUT1 inhibition, suggesting that a loosening of the extracellular matrix barrier surrounding the tumor mass is likely not contributing to the increased presence of intra-parenchymal CD8⁺ T cells. To this end, we show there is a net decrease in the proportion of CD73⁺ glyCAF at the tumor border, following GLUT1i, which may be responsible for the observed phenotype (**Fig. 4b,c**).

Figure R7. 20x images taken from Ctrl and GLUT1i treated tumors stained with Masson's Trichrome. Collagen at the tumor margin is shown in blue and nuclei are shown in brown.

Beside the functional differences between the CAFs subtypes, we also want to point out that since our initial submission, we have further investigated the molecular mechanisms used by the glyCAF to block T cells migration into the tumor parenchyma, and further studied the role of the CXCL16- CXCR6 signaling axis between glyCAF and CD8⁺ T cells. We now show additional evidence that CXCL16 expressing glyCAF serves as a mechanism to retain CD8⁺ T cells in the tumor stroma in Figure 5 of the revised manuscript. Notably, glyCAF are particularly enriched for expression of CXCL16 (**Fig. 5f**). Knockdown of either *Cxcl16* or *Glut1*, or GLUT1 inhibition is successful in abrogating CXCL16 expression in glyCAF (**Fig. 5h,j**), and as a result can rescue the defects in CD8⁺ T cell migration in transwell (**Fig. 5i**). In the same line, *Cxcr6*^{-/-} T cells exhibit increased migration toward tumor cells relative to *Cxcr6*^{+/+} wildtype CD8⁺ T cells in the presence of glyCAF (**Fig. 5m,n**). We further corroborated this result *in vivo* with adoptive transfer experiments, in which we observed increased infiltration of *Cxcr6* deficient T cells in the tumor parenchyma relative to wildtype T cells (**Fig. 5o-q**).

Minor comments:

Fig 2a: please mark CAFs and tumor clusters clearly (similar to the annotations in ED Fig 2a) for ease in reading.

We have updated **Fig. 2a** to include cluster labeling.

Reviewer #2 (Remarks to the Author): with expertise in sarcomas

Thank you for the opportunity to review this tour-de-force of sarcoma immunology. The finding that specific sarcoma CAFs may be manipulated to change the TME and consequently leverage the efficacy of doxorubicin is very exciting and clinically relevant.

There are certainly compelling aspects to these data, but many relevant criticisms are listed below in the order presented. Minor weaknesses are a consistent lack of detail. The evaluation of only two human soft tissue sarcomas for comparison is a moderate weakness, as is the use of only female animals and the subcutaneous model. More significant weaknesses are the essentially neglecting the importance of metastatic sarcoma metastases and the lack of addressing any limitations of the work. Specific comments are below.

Review: Cancer-associated fibroblasts metabolic targeting overcomes T-cell exclusion and chemoresistance.

We express our gratitude to the reviewer for providing valuable and constructive suggestions that significantly enhanced the robustness of our study's findings. In response to the insightful recommendations made by the reviewers, we performed additional experiments, which have played a crucial role in elucidating the molecular mechanisms through which glyCAFs mediate the exclusion of T cells from the inner tumor parenchyma.

Introduction

- Bottom of page 3/top of page 4. Please expand a little more on the limitations of current treatment and prognoses. Local recurrence at 2 years post-op is around 20% globally, and the incidence of metastatic disease is basically a coin-toss (~50%). Highlighting these realities will help to illustrate the importance of novel treatments.

We have incorporated these suggestions into the manuscript (**Page 3-4**).

Results

- Please consider mentioning on page 5 that the p53KO mice are indeed immune competent and explain why only female mice were used or list it as a limitation of the study.

We have updated the text to highlight that the p53^{KO}Ccne1⁺ and p53^{KO}Vgll3⁺ mouse models are indeed immunocompetent STS models (**Page 5**). In our study, we chose to use sex-matched female mice as tumor recipients for our syngeneic transplantation because the parental p53^{KO}Ccne1⁺ and p53^{KO}Vgll3⁺ cells were derived from female p53^{KO} mice. However, implanting p53^{KO}Ccne1⁺ and p53^{KO}Vgll3⁺ cell lines in male mice recapitulates the differences in immune infiltration we originally observed in the female mice (**Figure R8**) in which Vgll3⁺ tumors are infiltrated by more CD45⁺ immune cells including CD4⁺ and CD8⁺ T cells, compared to the Ccne1⁺ ones. Nevertheless, because all the experiments shown in the manuscript have been performed with female recipient mice, we agreed with the reviewer that this should be addressed as a limitation of the work (see **page 17** of the revised manuscript).

Figure R8. Differences of immune infiltration in Ccne1+ and Vgll3+ sarcoma growing in male recipient mice.

- Bottom of page 5. What was the metastatic phenotype of the p53KOCcne1+ and p53KOVgll3+ cells? This is no small point, as metastatic disease is ultimately responsible for the most sarcoma mortality.

Our focus in this study revolves around the understanding that while metastasis is a significant contributor to sarcoma-related mortality, studying and addressing the primary lesion is crucial for preventing these metastatic events. Therefore, the primary aim of this study was to explore new mechanisms that enhance the control of primary lesions, and more specifically, on identifying the means employed by tumor cells to evade immune responses.

Recent advancements, such as the use of anti-PD1/PDL1 antibodies (immune-checkpoint inhibitors) and genetically engineered CAR-T cells, have shown potential in regressing tumor growth. However, immune-checkpoint inhibitors exhibited limited efficacy for STS, and the main hurdle for CAR-T cells is their inability to penetrate the inner tumor parenchyma. Notably, a significant majority of STS are characterized as "T cell-excluded" tumors, with minimal lymphocyte infiltration (82.2% of n=213 analyzed patients from TCGA-SARC), in contrast to tumors exhibiting high cytotoxic T cell infiltration (17.8%)⁷.

To implement available immunotherapies effectively, our focus has been on understanding how to increase T cell infiltration within the inner mass. Consequently, experiments evaluating the metastatic potential of the generated cells were not conducted. Such assessments would require specialized experiments involving tumor surgical resection and the maintenance of animals to observe relapsing disease, which will be executed later for projects focused on mechanisms behind metastasis formation. The time frame of our *in vivo* experiments spanned between 3 and 5 weeks, at which point we sacrificed the animals with a tumor size of ~1 cm³. However, this duration is likely insufficient for the metastatic process to manifest visible macro-metastasis in distal tissues. As expected, no visible macro-metastasis was detected in the lung and liver, which are the primary sites of metastasis in patients.

Nevertheless, we concur with the reviewer that undertaking targeted studies to investigate the differential contributions of tumor genetics and distinct immune tumor microenvironments (TMEs) to the cells' metastatic potential is crucial, and this aspect will be further explored in future projects.

- Please be more precise than "slight increased tumor growth" on the bottom of page 5. This is pretty subjective. Please give a % difference or a non-significant p value or ideally both.

We have updated the corresponding text (see **page 6** of the revised manuscript) to reflect that the tumors generated from p53^{KO}Ccne1+ cells are 34% larger than those generated from p53^{KO}Vgll3+ cells (-0.2080g ± 0.09167g, p= 0.0530).

- What was the doxorubicin response of the p53KOCcne1+ animals? The robust response of the other animals is mentioned by nor these.

Figure 1g shows that *Ccne1*+ tumor bearing mice do not present a significant reduction in tumor growth in response to doxorubicin ($p=0.5525$), while the *Vgll3*+ tumor bearing mice show a significant reduction in tumor growth in response to doxorubicin ($p<0.0001$).

- The confirmatory human tumors for the CD90 analysis on page 7: MFS, LMS, UPS. These were single sarcomas or microarrays of these histologic subtypes? I have terrific sympathy for the challenge of finding solid numbers of such rare histologies, but 2 or 3 cases is not very compelling even in STS.

We appreciate the reviewer understanding of the challenges in acquiring primary human STS samples, and we agreed that adding a few samples would have strengthened the results of our study. Per reviewer's request, we have included two additional independent LMS primary tumors in our analysis, which both show a population of CAF consistent with the markers we had previously presented, thus fully corroborating our initial data.

Our study now includes 4 independent human samples (**Fig 2 and Extended Figure 2** of the revised manuscript); however, we would like to further highlight the challenges that do not currently favor the inclusion of more human samples in this study:

Firstly, we would like to emphasize that the single-cell RNA-seq (scRNA-seq) data previously presented in Figure 2 of the manuscript is from single sarcomas that were processed from fresh surgical specimens collected from our institution. The need for fresh samples combined with the rare nature of these tumors does not currently allow for high-throughput analysis of these specimens. Further, while microarrays are ideal for slide-based histological applications, tissue microarray samples are not suitable for the currently available single cell transcriptomic pipelines due to the need for a single cell suspension from fresh tissue with relatively high cell numbers (>10,000 cells).

Recent technological advances now allow for the digestion of FFPE blocks to single cell suspensions that are suitable for scRNA-seq processing and analysis, thus bypassing the need for fresh tissue. Still this technology is not applicable to the use of tissue microarrays.

While the ability to analyze FFPE embedded tumors has advanced the ability to capture transcriptomic data from biobanked tissues, it requires relatively new (<1 year old) fixed specimens with high RNA integrity to generate biologically relevant data. Additionally, the analysis of FFPE blocks is restricted to small tissue specimens that fit in standard paraffin molds, often which are taken from surgical specimens which may not include the tumor invasive front where many CAF may be prevalent. The resulting tumor specimen lacking the CAF-rich tumor margin may only contain small proportions of CAF which infiltrate the parenchyma. Nevertheless, by employing this newer technology based on FFPE single-cell RNA-seq, we were able to provide an additional two samples to our analysis, which corroborated and strengthened our initial results.

- Page 7-Similar to the above comment, please be more subjective and granular than "negligible" (...negligible differences across the two models...).

We have updated the text to incorporate these suggestions (see **page 8** of the revised manuscript).

Discussion

- Page 13. The absence of a discussion of limitations is conspicuous.

We agreed with the reviewer on this point and accordingly have updated the discussion to incorporate a paragraph listing the limitations of the current study (see **page 17** of the revised manuscript).

Methods

- Page 15. Were other orthotopic sites (intramuscular, for example) considered? Certainly sarcomas

superficial to the fascia occur, but disease deep to the fascia is much more common. Ostensibly the changes in oxygen tension between subcutaneous and subfascial locations have implications for this study.

We agree with the reviewer that it is important to consider if glyCAF are present in additional orthotopic sites which may vary in oxygen tension because a large fraction of STS occur in the retroperitoneum and in muscular tissue⁸. As such, we modeled our Ccne1+ immune-excluded tumors in these sites by injecting tumor cells intramuscular (i.m) and intraperitoneal (i.p). In these distinct anatomical sites, the proportion of total CAFs (**Fig. R9A**) and among them the glyCAF (Tumor⁻, CD45⁻, CD31⁻, CD90⁺ CD73⁺ cells) are comparable, with glyCAF comprise roughly 40-50% of the total CAF at each distinct site (**Fig. R9B**), suggesting that the presence of glyCAF is not an artifact of the subcutaneous location of these tumors but is a commonality of STS at the three tissue microenvironments.

Figure R9. A, Proportions of CAF (Tumor⁻, CD45⁻, CD31⁻, CD90⁺ cells) and **B**, glyCAF (Tumor⁻, CD45⁻, CD31⁻, CD90⁺ CD73⁺ cells) in Ccne1+ tumors at distinct anatomical sites.

Reviewer #3 (Remarks to the Author): with expertise in CAFs

In this manuscript, Broz et al. identify a subset of CAFs in soft-tissue sarcoma (STS) that express glycolysis-related genes and display increased glycolytic activity, termed glyCAF. Increased glyCAF was associated with reduced CD8+ T cell infiltration, which could be reversed with administration of a GLUT1 inhibitor (GLUT1i). Increased glycolysis in CAFs has been previously reported in other cancer types (Yang et al. IJOS 2021, Zhang et al. Cell Reports 2015, Becker et al. Cell Reports 2020) and this manuscript links such metabolic reprogramming with the immune microenvironment. Several points should be addressed, as outlined below.

We express our gratitude to the reviewer for providing valuable and constructive suggestions that significantly enhanced the robustness of our study's findings. In response to the insightful recommendations made by the reviewers, we performed additional experiments, which have played a crucial role in improving our comprehension of glyCAF features and elucidating the molecular mechanisms through which glyCAF mediate the exclusion of T cells from the inner tumor parenchyma.

1. In Fig. 2, immune cells and other non-CAF and cancer cells are analyzed ('Immune/other'). It would be more informative to have more in depth separation of this subset of cells, as the overlap in expression with CAFs varies depending on the type of immune/stromal cell, i.e. *Thy1* is expressed by T cells but not myeloid cells.

We guide the reviewer to **Extended Data Fig. 2a-e** in which we show a more in-depth separation of cell subsets in the immune/other category (i.e T cells, B cells, NK cells, DC and Myeloid Cells) and marker *Thy1* expression across these clusters. We show that in addition to CAF, *Thy1* is also expressed in T cells and NK cells, but not by myeloid cells at the transcript level.

2. The authors refer to *Thy1/CD90* as a pan-CAF marker; however, the data as currently presented only shows enrichment of *Thy1* in CAFs. The percentage of CAFs that express *Thy1* should also be included to further strengthen its use as a pan-CAF marker.

We thank the reviewer for raising this important point regarding *Thy1* as a CAF marker. We agree with the reviewer that "pan"-CAF is a stronger claim than is warranted, and we have changed the language accordingly to highlight that *Thy1/CD90* serves as an enrichment marker for total CAF in our STS mouse models. Also, in our single cell RNA-Seq dataset of Mouse UPS (**Extended Data Fig. 2a**), we show *Thy1* expression in the CAF cluster. We now include that the statistic that 87.6% of cells in the cluster annotated as CAF express *Thy1* (see **page 7** of the revised manuscript). Please note that the actual percentage is likely higher, due to dropout common to scRNA-seq.

3. Doxorubicin in combination with GLUTi treatment is associated with increased CD8+ Granzyme B+ cells, but whether GLUTi alone induces T cell activation should be evaluated. Similarly, the abundance of CD4+ T cell subsets, e.g., FoxP3+ Tregs vs. FoxP3⁻ Teff, should be determined in response to doxorubicin and GLUT1i alone and in combination.

We thank the reviewer for drawing our attention to the importance of acknowledging the abundance of FoxP3+ Tregs, FoxP3⁻ Teff and Granzyme B+ CD8+ cells under GLUT1i. We initially showed that GLUT1i enhances the abundance of CD8+ T cells in the tumor parenchyma (**Fig. 4h,i** of the revised manuscript), although we observed no changes in tumor growth with GLUT1i alone. In this line, we now show that the abundance of CD8+ GranzymeB+ expressing cells in the tumor parenchyma is unchanged in GLUT1i treated tumors (**Extended Data Fig. 6a**), suggesting that the single treatment, while recruiting more CD8+ T cells in the tumor mass, does not alter the cytotoxic ability of the recruited CD8+ T cells. Interestingly, the

single treatment with Doxorubicin also showed no effects in recruiting T cells in the tumor mass or altering the abundance of CD8⁺ GranzymeB⁺ in the tumor. Importantly, however, we found that the combinatorial treatment (GLUT1i with DOX) trigger activation of the tumor infiltrating CD8⁺ T cells, with clear increased proportions of cytotoxic Granzyme B⁺ CD8⁺ T cells. We speculate that the increasing cytotoxicity of the T cells is due to enhanced antigen presentation following immunogenic cell death of tumor cells, which is induced by anthracycline based chemotherapy⁹⁻¹¹, and which is maximized in the Glut1i situation where a significant number of T cells is newly recruited to the tumor parenchyma (**Extended Data Fig. 6a**).

Additionally, following the reviewer suggestion, we quantified the numbers of FOXP3⁺ Tregs and CD4⁺ Teff in at the tumor margin and parenchyma in each treatment arm. We did not observe any significant changes to the abundance of FOXP3⁺ Tregs at the tumor margin or parenchyma in either the single or combination treatment arms. Interestingly, however, we observed an increase of CD4⁺ Teff in the parenchyma in GLUT1i treated tumors, although in combination treatment the effect on CD4⁺ Teff was not significantly maintained (p= 0.0844) (see **new figures in Extended Data Fig.6b-c** of the revised manuscript).

4. GLUT1i alters the abundance of CD73⁺CD90⁺ cells, which the authors term as 'glyCAFs'. Both CD73 and CD90 are also expressed by T cells; thus, the stains in Fig. 3b need to include an exclusion marker (CD45, CD3, etc.) to specifically identify CAFs. In addition, the authors should clarify if doxorubicin alone or in combination with GLUT1i also impacts glyCAF abundance.

Following the reviewer suggestion, we have updated the image in Fig. 3b (**Fig. 4b** of the revised manuscript) and the corresponding quantification (**Fig. 4c**) to include CD3 as an exclusion marker. Importantly, addition of CD3 confirms that the identified CD73⁺CD90⁺ cells are not T cells but CAFs, and it does not change the initial findings in which we conclude that GLUT1i reduces the number of glyCAF (CD3⁻CD90⁺CD73⁺ cells) at the tumor margin. These findings are also corroborated by flow cytometry (**Fig. 4d**) in which the CAFs are identified upon exclusion of tumor cells, immune cells, and endothelial cells (gating strategy for CAFs: Tumor- CD45⁻CD31⁻CD90⁺CD73⁺ cells)

While we recognize the importance of investigating glyCAF abundance upon DOX/GLUT1i treatment, we encountered technical challenges that impeded this analysis and hindered our ability to draw conclusions. One significant challenge was observed immediately following the combination treatment on Day 27 (**Fig. 6e**), where the tumor mass was extremely small, partly due to the high levels of T-cell-induced tumor killing (**Fig. 6d,i,k**). This resulted in the collection of only a few live cells from these tumors during the peak efficacy of the treatment, making it impossible to conduct a comprehensive analysis and quantification of CAF subtypes. After concluding the treatment and allowing the tumors to regrow, approximately 14 days following the last dose of DOX/GLUT1i, we analyzed the CAF compartment. However, no differences were observed in terms of CAF subtypes in DOX/GLUT1i samples compared to the control untreated tumors (data not shown). We suspect that at this later timepoint, although the tumor mass was sufficient for analysis, the glycolytic features of the CAFs were reactivated in the absence of GLUT1i, leading to the repopulation of the tumor mass by glyCAFs. To overcome these challenges, future experiments will be crucial, involving lower doses of DOX/GLUT1i and additional time points.

5. While depletion of CD8⁺ T cells promotes tumor growth in the context of doxorubicin and GLUT1i, it does not completely reverse growth inhibition. The authors should clarify if CD4⁺ T cells are also important, or if there is an additional unknown mechanism by which GLUT1i acts.

In response to the reviewer's suggestion, we conducted a replicated experiment that included a CD4⁺ T cell depletion arm alongside the previously studied CD8⁺ depleted group. The updated experiment, presented in Fig. R11, underscores the crucial role of both CD4⁺ and CD8⁺ T cells in sustaining the anti-tumor effect induced by the combination of doxorubicin and Glut1 inhibition. While CD4⁺ T cells likely contribute to the anti-tumor effect by supporting CD8 T cell activation, we posit that the infiltration of CD4 T cells into the tumor is less reliant on the glyCAFs through the Cxcl16-Cxcr6 mechanism identified for CD8⁺ T cells.

Supporting this hypothesis, CD4 T cells express lower levels of the CXCR6 receptor compared to the CD8 T cells (**Fig. 5g**). Additionally, the interaction of CD4 Th1 with the Cxcl16 glyCAFs at the tumor border is less intense than that of CD8 cells (**Fig. 5e**). Consequently, CD4 cells are relatively less blocked at the tumor margin (**Fig. 1e**). Nevertheless, both CD4 and CD8 cells benefit from GLUT1 inhibition, facilitating their infiltration into the tumor parenchyma.

Given that we recognize the need for further investigations into the underlying mechanisms regarding the role of CD4+ T cells in facilitating an anti-tumor response in the DOX/GLUT1i setting, we have opted not to include the CD4 depletion experiment in the manuscript. Nevertheless, we remain open to incorporating it should the reviewers request it.

Figure R11: Tumor growth curve of DOX/GLUT1 tumors depleted of CD4 or CD8 T cells (n=5).

We invite the reviewer to continue on to our discussion of comment #6 below where we further explore whether other TME cells are affected by GLUT1i and propose convincing evidence in support of the glyCAF CXCL16-CXCR6 axis in mediating immune exclusion.

6. In Fig 5, authors are recommended to explain which cell type the GLUT1i target. The authors have shown us reprogramming the glycolytic properties of CAFs by inhibiting GLUT1 led to increased infiltration of cytotoxic CD8+ T cells into the tumor, independent of cancer cell-intrinsic metabolism, but the metabolic markers used for glyCAF like *Slc2a1*, *Pgk1*, *Pkm*, *Pgam1*, *Hk2*, *Nt5e*, and the upregulated glycolysis pathways may not only exist in glyCAF and cancer cells and such cells could be impacted by GLUT1i. A scRNA-seq feature or violin plot showing specificity of the glyCAF marker genes would further support that the phenotypes observed are specific to CAFs and not present in other cell subsets.

We acknowledge that while we observe the primary effect of the GLUT1 inhibitor on reprogramming the glyCAF, other cells of the tumor microenvironment may be impacted. Firstly, we showed that although tumor cells glycolysis is impaired under GLUT1i (**Extended Data Fig. 4g**), tumor cells short-hairpin RNA targeting of *Glut1* (shGLUT1) does not recapitulate the increase in T cell infiltration we observe under GLUT1i (**Fig. 5j,k and Extended Data Fig. 4h**), suggesting that other cellular components of the tumor microenvironment are responsible. Certain immune cells, namely T-lymphocytes and certain subsets of tumor associated macrophages (TAMs) cells utilize glycolysis for their metabolism^{12,13} and express the glycolytic signature, which we acknowledge is not specific to glyCAF. As such, we now have characterized the immune microenvironment (CD45+ fraction) following GLUT1i to confirm that glyCAF are the primary cellular target of GLUT1i.

We now include a projection of the glycolysis signature (*Slc2a1*, *Pkm*, *Pgk1*, *Pgk1*, *Eno1*, *Ldha*, *Ldhb*, *Hk1*, *Aldoa*) in the myeloid cells (monocytes/macrophages) and lymphocytes in the tumor microenvironment under GLUT1i (**Extended Data Fig. 4b,c** of the revised manuscript). Interestingly, while we observe a decrease in the glycolysis signature in glyCAF (**Fig. 5f and Extended Data Fig. 4a**), we do not observe a

reduction in glycolysis pathway transcripts in other myeloid (**Extended Data Fig. 4b**) or lymphoid cells (**Extended Data Fig. 4c**), suggesting that glyCAF are the primary target of GLUT1i. It is unclear why immune cells are not particularly affected by GLUT1i while glyCAF are. We suspect that glycolysis may be particularly essential to certain myeloid and lymphoid lineages, and thus upon GLUT1i these cells utilize compensatory mechanisms, such as the increase of other glucose transporters (GLUT3, GLUT4) to sustain glucose import without impairing glycolysis, although this warrants further exploration.

We next focused our attention on investigating the effects of GLUT1i on tumor associated macrophages (TAMs), which are the dominant immune cell in *Ccne1*⁺ tumors. In a previous study, we have shown that *Ccne1*⁺ tumors are infiltrated by TAMs that exist in several states within the tumor mass, namely MΦ *Thbs1*, MΦ *C1qa*, MΦ *Spp1*, and Mono/MΦ *Irf14* (**Extended Data Fig. 4d,e**). Therefore, we investigated if GLUT1i alters the proportions of these subtypes within the tumor mass. In the revised manuscript, we show that despite a trending increase of *Spp1*⁺ MO in GLUT1i treated tumors, which was largely driven by two outliers, the TAM heterogeneity is comparable between the two conditions (**Extended Data Fig. 4f**), further suggesting that TME cells other than the glyCAF are not particularly affected by the GLUT1i.

Importantly, in addition to characterizing the effect of GLUT1 inhibitor on the TME cells, we also want to highlight that since our initial submission, we have further investigated the molecular mechanisms used by the glyCAFs to block T cells migration into the tumor parenchyma, and further studied the role of the CXCL16- CXCR6 signaling axis between glyCAF and CD8⁺ T cells. We now show additional evidence that CXCL16 expressing glyCAF serves as a mechanism to retain CD8⁺ T cells in the tumor stroma in Figure 5 of the revised manuscript. Notably, glyCAF are particularly enriched for expression of CXCL16 (**Fig. 5f**). Knockdown of either *Cxcl16* or *Glut1* is successful in abrogating CXCL16 expression in glyCAF (**Fig. 5h and Extended Data Fig. 5g**), and as a result can rescue the defects in CD8⁺ T cell migration in transwell (**Fig. 5i**). In the same line, *Cxcr6*^{-/-} T cells exhibit increased migration toward tumor cells relative to *Cxcr6*^{+/+} wildtype CD8⁺ T cells in the presence of glyCAF (**Fig. 5m,n**). We further corroborated this result in vivo with adoptive transfer experiments, in which we observed increased infiltration of *Cxcr6* deficient T cells in the tumor parenchyma relative to wildtype T cells (**Fig. 5o-q**).

7. In the abstract, the authors state, "Targeting glycolysis in glyCAFs enhances T-cell infiltration and augments the efficacy of chemotherapy." However, the results do not specifically support the assertion that glyCAFs, and not other non-cancer cell types with glycolytic features, that enhance T-cell infiltration and improve chemotherapy efficacy. The abstract should be revised accordingly.

We have updated the abstract to reflect this distinction (See **page 2** of the revised manuscript), however in the revised manuscript we show additional convincing evidence illustrating that glyCAF are selectively targeted by GLUT1i, while other TME cells are spared from this effect (see **Fig. 4, Extended Data Fig. 4, and Fig. 5** of this revised manuscript).

8. In Fig 2h, authors are recommended to perform enrichment or quantitative metabolic prediction analysis based on scRNA-seq results to substantiate the metabolic signature of glyCAFs.

Following the reviewer suggestion, we now include in **Fig. 3g** of the revised manuscript an over-representation analysis based on the CAF scRNA-seq and show that KEGG pathways related to glycolysis/gluconeogenesis and HIF-1 signaling are significantly enriched in glyCAF. Furthermore, pathways related to carbon metabolism and central carbon metabolism—which include the overarching pathways related to the breakdown of glucose via glycolysis, TCA Cycle, and the Pentose Phosphate Pathway (PPP)—are enriched in glyCAF.

9. In Fig 4d and Extended Data Fig 5, it would be helpful to compare glyCAFs to other cell types in terms of communication strength with CD8/CD4 T cells and differences in ligand-receptor pairs. The current results

don't exclude the potential effects on T cells by cell types other than glyCAFs. These other cells, also present in tumor tissues and characterized by high metabolic features, could be inhibited by GLUT1i.

We thank the reviewer for bringing our attention to this critical component of our receptor-ligand communication analysis. We now include other TME cellular components, such as CD4⁺ T cells (Treg, Th1, Naïve), B cells, NK-cells, and myeloid lineages (Granulocyte, MonoMacDC, and pDC) as well as Tumor and endothelial cells in our receptor-ligand analysis (**Fig. 5e**).

First, among the CAF types, apCAFs and glyCAFs showed the highest probability of interaction with the CD8⁺ T cells via the Cxcl16/Cxcr6 axis. However, while the expression of Cxcl16 mRNA is similar between glyCAFs and apCAFs, post-transcriptional regulations of the RNA are differentially occurring in these cells, with Cxcl16 protein being higher expressed in the glyCAFs (**Fig. 5f**). These data suggest that the glyCAFs are, among the CAFs, more likely to interact with the CD8⁺ T cells through the Cxcl16/Cxcr6 axis. It is possible that the glycolytic profile of the CAFs is somehow responsible for the Cxcl16 protein expression, and accordingly, we found changes in the Cxcl16 expression upon GLUT1 modulation in the glyCAFs (**Fig. 5h,j and Extended Data Fig. 5g**), but not in other cells (**Fig. 5k,l**).

When we included other TME cells in the receptor-ligand communication analysis, we observe that the MonoMacDC cluster, in addition to CAFs, is also likely to interact with effector and exhausted CD8⁺ T cells through the CXCL16-CXCR6 signaling axis, suggesting the possibility that myeloid cells, likewise the CAFs, could block CD8 T cells infiltration in the tumor parenchyma. Nevertheless, the following observations minimize the possibility that myeloid cells are involved in the observed phenotype:

- 1- Increased T cells infiltration in the tumor parenchyma upon Glut1 inhibition correlated to decreased *Cxcl16* expression in the CAFs (**Fig. 5j**) but not in the myeloid cells (**Fig. 5l**). Proportions of CXCL16 expressing glyCAF are also significantly diminished following GLUT1i *in vivo* (**Fig. 4b-d**). Unlike the CAF, knockdown of Glut1 in macrophages does not regulate the expression of Cxcl16 (**Extended Data Fig. 5h**) and does not alter the T cells migration index *in vitro* (**Extended Data Fig. 5i**). Additionally, we also did not note changes of the overall proportions of TAM subsets upon GLUT1i (**Extended Data Fig. 4f**).
- 2- We show that CD90⁺ CD73⁺ glyCAF are closer in proximity to CD8⁺ T cells than CSF1R⁺ myeloid cells and CD90⁺ CD73⁻ non-glyCAF (which encompasses the populations of iCAF, mCAF, and apCAF). (**Fig. 4m**), suggesting that glyCAF have a higher likelihood of communicating via Cxcl16 to CD8⁺ T cells. This new evidence favors the hypothesis that the chemoattractant and adhesion properties of Cxcl16⁺ glyCAF retain CD8⁺ T cells at the glyCAF rich tumor margin.

Minor comments

1. Introduction, second paragraph: citations 8 and 9 do not include targeting of pan-CAFs, aSMA⁺ CAFs (marking a subset of CAFs) and type I collagen produced by aSMA CAFs are targeted. The text should be revised accordingly.

We thank the reviewer for bringing this error to our attention. The introduction text has been revised to reflect the correct information within these citations (See **Page 3** of the revised manuscript).

2. 'Tumor' and 'tumor cells' should presumably be 'cancer cells', as tumor is a more general term that includes all cells that would be present in a tumor.

We have changed the term "tumor cells" to "malignant cells" or "cancer cells" throughout the manuscript. In certain cases, we have used the term "dsRed⁺ tumor cells" or "Ccne1⁺ tumor cells", which correspond to our malignant cell lines used to initiate tumorigenesis, to clarify our reference to the malignant tumor cells within the tumor mass.

3. Page 7: the text states that dsRed, CD45, and CD31 are used as exclusion markers; however, CD31 is not included in the schematic in Fig. 2 or the gating strategy in Extended Data Fig. 3a. Please clarify. In

addition, it is not clear if the analysis in Fig. 2i also includes exclusion of dsRed and CD45 expressing cells.

We thank the reviewer for bringing this error to our attention, we had initially left out CD31 from the gating strategy schematic in **Extended Data Fig. 3a**. We have updated the figure to include CD31 in the representative gating schematic.

Regarding Figure 2i (now **Fig. 3h** in the revised manuscript), the plot shown is generated from gating on the “CAF” identified as in **Extended Data Fig. 3a** which is consistent with the exclusion of dsRED+, CD45+, and CD31+ cells and inclusion of CD90+ cells. CD90 is shown again in the subsequent gate on the x-axis of the plot in to show the separation of CD73 expression within the CD90+ CAF subset.

Reviewer #4 (Remarks to the Author): with expertise in sarcomas

In this manuscript by Broz et al., the authors investigate the role of cancer-associated fibroblast (CAFs) in dictating immune response and chemosensitivity for CCNE+ soft tissue sarcomas, which have low immune infiltration compared to Vgll+ STS. Through comprehensive scRNA-seq of fibroblasts associated with mouse and human STS, they elucidated the expression programs of CAFs. Specifically, they have identified a subpopulation of CAFs with altered glycolytic activity, which are known as glyCAFs. The authors provide evidence that glyCAFs are associated with immune exclusion for STS and reside with the CD8 T cells at the periphery/margin of the tumor. Through the use of genetic and pharmacological approaches, they demonstrate that the glyCAFs significantly contribute to immune exclusion and that when used in combination with chemotherapy, such as doxorubicin, targeting the glycolytic aspects of the tumor (supposedly glyCAFs) can lead to enhanced immune infiltration and better anti-tumor activity. While the authors provide some insightful studies and findings, there remain some deficiencies with regards to the models and mechanisms. A couple of the more significant issues center around (1) why are CCNE+ tumors associated with more glyCAFs (is the tumor secreting factors that are recruiting these glyCAFs) and (2) the mechanisms by which the glyCAFs (the cell-cell communication data is intriguing, but underdeveloped and not conclusive) are excluding CD8 T cells in this genetically defined population of STS. Other concerns and critiques are noted below.

We express our gratitude to the reviewer for providing valuable and constructive suggestions that significantly enhanced the robustness of our study's findings. In response to the insightful recommendations made by the reviewers, we performed additional experiments, which have played a crucial role in elucidating the molecular mechanisms through which glyCAFs mediate the exclusion of T cells from the inner tumor parenchyma.

Minor comments:

1. Abrupt transition from mention of fibrotic cell numbers to scRNA seq re: CAF subtypes. Page 7, paragraph2.

We have added a transition to bridge the gap in logic between the number of fibrotic cells to the scRNA-seq of CAF sub-types in regard to explaining differences in immune infiltrate between the two models (See **Page 8** of the revised manuscript).

2. It is not clear to this reviewer: Is the identification of the glyCAF novel or has it been previously identified? This reviewer would conclude that this is a novel description since there does not seem to be a reference for these glyCAFs provided as was provided for the other CAF subpopulations mentioned? Just want to make sure this is correct.

We thank the reviewer for this question. To our knowledge, a unique population of CAFs with glycolytic properties has not been reported yet, although glycolytic metabolism has been suggested to be upregulated in CAF vs normal fibroblasts¹⁵⁻¹⁷. This would be the first observation to link the glycolytic features of CAF with immune exclusion.

However, it would indeed be important to know whether glyCAFs emerge in any cancers other than sarcomas. Since the submission of our paper, a new study by Cords et al.,⁶ published in 2023 put forward a transcriptomic and proteomic characterization of CAF subtypes, first in breast cancer and then validated in multiple cancer datasets (colon, lung, HNSCC, PDAC). Not surprisingly, different cancers were found to contain unique proportions of CAF subtypes, with iCAFs and mCAFs predominating in PDAC, for example.

The average expression in the CAF subtypes of the top 40 marker genes of the relevant Cords et al subtypes is shown below. Accordingly, we found that the tCAF gene signature reported in Cords et al was highest in our glyCAFs (**Fig. R12 below**).

Figure R12: Signatures of human iCAF, mCAF, apCAF, and tCAF as identified by Cords et al., 2023 projected onto the murine CAF clusters identified in this manuscript.

Interestingly, CD73 (NT5E) was used as a marker for tCAFs in the breast cancer study, and pathway analysis demonstrated that the glycolysis was significantly enriched in tCAFs but no other CAFs. Additionally, the breast cancer study also found upregulation of glycolysis in CD73+ CAFs. The work was descriptive, however, and did not functionally demonstrate any function for these CAFs in terms of immune exclusion or suppression. Thus, to our knowledge, our manuscript would be the first to functionally describe immune exclusion driven by a highly glycolytic, CD73+ CAF population.

Despite this, we caution that until further study, CD73/NT5E is currently a marker and not necessarily an integral part of the glycolytic phenotype. Indeed, while the glycolytic “tCAF” population was found in multiple cancers by Cords et al.’s validation scRNA-seq dataset, *NT5E* was expressed lowly and non-differentially by tCAFs in cancers other than breast, suggesting that CD73 might serve as a marker of glycolytic CAFs in breast cancer and sarcoma, but not all cancers.

3. Fig. 5F. Difficult to believe that DOX alone provided a statistically significant advantage? 2-3 days? Had enough animals/statistical power to make that conclusion?

We thank the reviewer for bringing this to our attention. For this experiment, we harvested half of the mice (n=4) at the cessation of treatment to understand how the treatments are impacting the tumor immune infiltrate, and half of the mice (n=4 per group) were left in the study for the survival analysis. In this case, we performed the Log-rank (Mantel-Cox) test which returned the p-value 0.0317 using n=4 mice per group. It is possible that the p-value may have become inflated due to the small sample size, suggesting that DOX provided a statistically significant survival advantage. However, this does not change the proposed mechanism in which the combination treatment (DOX+GLUT1i) provided a significant advantage relative to either Control or DOX alone groups, suggesting that GLUT1i is critical contributor to this effect.

More significant comments:

1. Cross-comparison analysis of the models/tumors: How do the molecular signatures of the MSC-transformed cells compare to human tumors with respective genetic traits. It was not presented early on how truly representative the genetically engineered MSC models compare to human tumors with these genetic amplifications. Are there potentially other genetic features associated with CCNE+ tumors that are contributing to the immune profiles in human patient tumors?

Ccne1-high STS tumors (TCGA-SARC, n=213) exhibit a positive correlation with the expression of other cell-cycle regulators (e.g., CDK1, MCM7, and CDC25A) (**Fig.R13**), a pattern also recapitulated in murine MSC-derived Ccne1+ sarcoma relative to non-transformed mesenchymal stem cells (MSC) (**Fig. R14**). Accordingly, **Extended Data Fig. 1e** demonstrates that Ccne1+ cells display a higher proliferation rate compared to non-transformed MSC and Vgll3+ sarcoma cells. Recent investigations across various tumor types, including synovial sarcomas¹⁸ have identified specific expression programs in tumor cells linked to T-cell exclusion and the development of cold environments. Notably, these studies have associated elevated expression of cell-cycle-related genes with reduced T cell infiltration in the tumor mass^{19–21}. In this respect, another study for example correlated Ccne1 amplification with poor immune infiltration in the context of ovarian cancer²².

Figure R13. Correlations between Ccne1 mRNA expression and other cell-cycle regulators in TCGA-SARC (n=213).

Figure R14. Expression of cell cycle regulator genes in murine MSC and Ccne1+ tumor cells.

Despite these correlations, it remains unclear how the increased expression of cell-cycle-related genes may impact T cell infiltration, whether directly (through tumor cells) or indirectly (potentially through the generation of specific subtypes of cancer-associated fibroblasts, CAFs). Further studies are needed to elucidate this aspect. Additionally, it is crucial to evaluate the direct involvement of tumor cells with different genetics in the biogenesis of distinct CAF types. These aspects will be the focus of future publications, aiming to provide a clearer understanding of the interplay between cell-cycle regulation, tumor cells, and the tumor microenvironment in influencing T cell infiltration.

2. Besides T cells, other immune profiles altered in the CCNE+ tumors?

We would like to guide the reviewer to **Fig. 1c** which in addition to the CD4+ and CD8+ T cells, we show differences in proportions of overall tumor associated macrophages (F480+ CD11b+ TAMs), and NK cells are significantly reduced in Ccne1+ tumors relative to Vgll3+ tumors, while the proportions of monocytes (F480- CD11b+) are unchanged.

3. Fig 1: For pathway analysis, what are the negative normalized enrichment scores. For (A), how did the authors determine low immune infiltration for CCNE+ based on this pathway analysis...no indication. Unless a negative NES?

We thank the reviewer for directing our attention to this figure, whose corresponding text contained an error. The analysis performed in Fig. 1a was an Over-Representation Analysis method that did not return a negative enrichment score—and not GSEA as was stated in the manuscript. We have replaced the figure with GSEA conducted on the same samples, which indeed demonstrates negative NES of GO:BP immune-response gene sets for the CCNE1 group (see revised **Fig. 1a**).

Fig 1b: the influence of the 3D scaffold on immune infiltration? Has that been controlled for?

We appreciate the reviewer's comment and would like to emphasize that all experiments presented in this manuscript, including those comparing immune infiltration between the p53^{KO}Ccne1+ and p53^{KO}Vgll3+ models as depicted in Fig. 1, were conducted using scaffolds. Therefore, it is implausible that the scaffold itself is responsible for the observed differences in immune infiltrate. Our use of a scaffold-based approach offers the advantage of generating tumors from a minimal number of tumor cells (150,000 cells per scaffold). This contrasts with traditional transplantation models, such as those involving subcutaneous injection of tumor cell suspension in Matrigel, which typically requires a minimum of 1 million sarcoma cells for tumor growth. Importantly, initiating tumors from a smaller number of cells results in slower tumor formation, providing ample time for the host immune response to develop and for immune tolerance to establish over an extended period.

To address concerns about the impact of the scaffold on immune infiltration, we compared the immune profiles of p53koCcne1+ tumors (Cold/immune excluded model) growing within scaffolds versus those injected with Matrigel. Tumors resected at a similar size (~1 cm³) revealed no significant impact of the scaffold on the overall CD45⁺ immune infiltrate when compared to tumors generated in Matrigel (**Figure R15**).

Figure R15. Expression of cell cycle regulator genes in murine MSC and Ccne1+ tumor cells.

Fig 1h: Vgll3+ tumors still grow and become resistant to Dox. How do the T cell populations change over time in response to Dox?

To address this question, we treated Vgll3+ tumors with doxorubicin (6mg/kg) every 3 days for 9 days and then assessed the T cells immediately following treatment (Early timepoint, Day 29) and one week following the treatment when the tumors begin to become resistant to DOX (Late timepoint, Day 37) (**Fig. 1h**). We FACS-sorted CD4 and CD8 T cells and performed gene expression analysis.

Among CD4+ T cells, we noticed a trend in the reduction of regulatory T cell transcripts (*Foxp3*, *Rora*, *Ctla4*) at an early time-point after doxorubicin treatment (**Fig. R16**). Conversely, we noted an increase in these T-reg associated transcripts at a later time-point after doxorubicin. We did not note any differences in Th1 or Th2 cells, identified for the expression of Tbet (*Tbx21* gene)/Irfng and Gata3 respectively.

Figure R16. Gene expression analysis of Treg, Th1, and Th2 markers in sorted CD4+ T cells isolated from DOX treated tumors at early (D29) and late (D37) timepoints.

Among CD8+ T cells, we noted a reduction in transcripts encoding exhaustion and checkpoint molecules (*Pdcd1/Pd1*, *Tim3*) at an early time-point after doxorubicin treatment, which was resolved at later timepoints. No differences were observed for cytotoxic markers (*Prf1*, *Gzmb*, *Ifng*) at either early or late timepoints.

Figure R17. Gene expression analysis of cytotoxic and exhaustion markers in sorted CD8+ T cells isolated from DOX treated tumors at early (D29) and late (D37) timepoints.

While our current work is centered on understanding the mechanisms of immune exclusion in the *Ccne1* model, we believe that it will be critical to conduct further studies aimed at understanding how highly immune-infiltrated models, such as the *Vgll3* model, evade immune responses.

4. Page 7, related to Fig 2: The authors mention that the distance of fibrotic stroma to invasive margin and then provide rationale that number of fibrotic cells does not explain the CD8 distribution. The description for the “criteria” is confusing and the rationale/justification should be mentioned for using this criteria as the determination for immune exclusion deserves further explanation.

We agree with the reviewer that this justification was not clearly stated, and we have revised the corresponding text on **page 8** to clarify our rationale.

5. Fig 3b, appears that *Glut1* inh eliminates the glyCAFs. Total CAFs increase with *Glut1* inh...how do the other CAF populations change?

To properly address how the remaining CAF populations change with GLUT1 inhibition would involve generating glyCAF-tracing mouse models to follow the fate of glyCAF under GLUT1i. Indeed, we attempted to generate a glyCAF reporter mouse by crossing *aSMA-Cre^{ERT2}* mice (*aSMA* is indeed mainly expressed in the glyCAFs) to the *Rosa26-EGFP^{LSL}* mice to generate GFP expressing glyCAF, useful to understand how these cells change upon GLUT1i. Following tamoxifen administration, we were unable to identify any GFP+ cells in the tumor bearing reporter mice and noticed technical issues with the Cre expression in this model, therefore we were unable to perform the lineage tracing experiments.

In lieu of lineage tracing, we analyzed single-cell transcriptomic data of the CAFs from GLUT1i treated tumors. In the revised manuscript, we show that among the CAFs only the glyCAF signature, but not iCAF, mCAF, or apCAF signatures are altered by GLUT1i (**Fig. 4f**), suggesting that the glyCAF lose their glycolytic metabolism and re-direct towards becoming one of the other non-glyCAF populations. Again, lineage-tracing *in vivo* experiments would be needed to properly assess the changes in the specific glyCAF population; accordingly, we will re-generate the appropriate mouse models and we will perform these analyses in future investigations.

6. Additional characterization of the glyCAFs would be beneficial: For example, *in vitro* comparison of glycolysis properties for glyCAF vs other CAFs through the use of Seahorse metabolic profiling? Quantification of lactate levels? Any changes in mitochondrial properties/functions for these glyCAF cells?

Following the reviewer suggestion, we have updated the manuscript to include Seahorse metabolic profiling of glyCAF and non-glyCAF using the Glycolytic Stress Test. By measuring the extracellular acidification rate (ECAR) throughout the assay (**Fig. 5a** and **Extended Data Fig. 5c**) we observe that glyCAF have a significantly increased glycolytic rate compared to non-glyCAF, functionally confirming the enhanced glycolysis of glyCAF and strengthening the conclusions of this manuscript.

7. What is the effect of GLUT1 inhibitor on the T cells functions/properties? Further characterization of the effect of the *Glut1* inhibitor on the other tumor cell constituents/properties would be beneficial.

Per reviewer suggestion, we first explored the possibility that GLUT1i directly alters CD8+ effector function in addition to the increased migratory capabilities we have defined in Fig. 4 and Fig. 5 of the manuscript. In **Extended Data Fig. 6a**, we show that GLUT1i by itself, while recruiting more CD8+ T cells in the tumor mass, fails to increase Granzyme B+ cytotoxic cells. However, we found that the combinatorial treatment (GLUT1i with DOX) trigger activation of the tumor infiltrating CD8+ T cells, with clear increased proportions of cytotoxic Granzyme B+ CD8+ T cells. We speculate that the increasing cytotoxicity of the T cells is due to enhanced antigen presentation following immunogenic cell death of tumor cells, which is induced by anthracycline based chemotherapy⁹⁻¹¹, and which is maximized in the GLUT1i situation where a significant number of T cells is newly recruited to the tumor parenchyma (**Extended Data Fig. 6a**).

We acknowledge that while we observe the primary effect of the GLUT1 inhibitor on reprogramming the glyCAF, other cells of the tumor microenvironment may be impacted. Firstly, we showed that although tumor cells glycolysis is impaired under GLUT1i (**Extended Data Fig. 4g**), tumor cells short-hairpin RNA targeting of Glut1 (shGLUT1) does not recapitulate the increase in T cell infiltration we observe under GLUT1i (**Fig. 4k-l and Extended Data Fig. 4h**), suggesting that other cellular components of the tumor microenvironment are responsible. Certain immune cells, namely T-lymphocytes and certain subsets of tumor associated macrophages (TAMs) cells utilize glycolysis for their metabolism^{12,13} and express the glycolytic signature, which we acknowledge is not specific to glyCAF. As such, we now have characterized the immune microenvironment (CD45+ fraction) following GLUT1i to confirm that glyCAF are the primary cellular target of GLUT1i.

We now include a projection of the glycolysis signature (*Slc2a1, Pkm, Pgk1, Pfkfb3, Eno1, Ldha, Ldhd, Hk1, Aldoa*) in the myeloid cells (monocytes/macrophages) and lymphocytes in the tumor microenvironment under GLUT1i (**Extended Data Fig. 4b,c** of the revised manuscript). Interestingly, while we observe a decrease in the glycolysis signature in glyCAF (**Fig. 4f,g and Extended Data Fig. 4a**), we do not observe a reduction in glycolysis pathway transcripts in other myeloid (**Extended Data Fig. 4b**) or lymphoid cells (**Extended Data Fig. 4c**), suggesting that glyCAF are the primary target of GLUT1i. It is unclear why immune cells are not particularly affected by GLUT1i while glyCAF are. We suspect that glycolysis may be particularly essential to certain myeloid and lymphoid lineages, and thus upon GLUT1i these cells utilize compensatory mechanisms, such as the increase of other glucose transporters (GLUT3, GLUT4) to sustain glucose import without impairing glycolysis, although this warrants further exploration.

We next focused our attention on investigating the effects of GLUT1i on tumor associated macrophages (TAMs), which are the dominant immune cell in *Ccne1*+ tumors. In a previous study, we have shown that *Ccne1*+ tumors are infiltrated by TAMs exist in several states within the tumor mass, namely MΦ *Thbs1*, MΦ *C1qa*, MΦ *Spp1*, and Mono/MΦ *Irf1*¹⁴ (**Extended Data Fig. 4d,e**). Therefore, we investigated if GLUT1i alters the proportions of these subtypes within the tumor mass. In the revised manuscript, we show that despite a trending increase of *Spp1*+ MΦ in GLUT1i treated tumors, which was largely driven by two outliers, the TAM heterogeneity is comparable between the two conditions (**Extended Data Fig. 4f**), further suggesting that cells other than the glyCAF are not particularly affected by the GLUT1i.

Importantly, in addition to characterizing the effect of GLUT1 inhibitor on the TME cells, we also want to highlight that since our initial submission, we have further investigated the molecular mechanisms used by the glyCAFs to block T cells migration into the tumor parenchyma, and further studied the role of the CXCL16- CXCR6 signaling axis between glyCAF and CD8+ T cells. We now show additional evidence that CXCL16 expressing glyCAF serves as a mechanism to retain CD8+ T cells in the tumor stroma in Figure 5 of the revised manuscript. Notably, glyCAF are particularly enriched for expression of CXCL16 (**Fig. 5f**). Knockdown of either *Cxcl16* or *Glut1* or GLUT1 inhibition is successful in abrogating CXCL16 expression in glyCAF (**Fig. 5h and Extended Data 5f**), and as a result can rescue the defects in CD8+ T cell migration in transwell (**Fig. 5i**). In the same line, *Cxcr6*^{-/-} T cells exhibit increased migration toward tumor cells relative to *Cxcr6*^{+/+} wildtype CD8+ T cells in the presence of glyCAF (**Fig. 5m,n**). We further corroborated this result in vivo with adoptive transfer experiments, in which we observed increased infiltration of *Cxcr6* deficient T cells in the tumor parenchyma relative to wildtype T cells (**Fig. 5o-q**).

8. Any plasticity for the CAF signatures/properties? Can a glyCAF change to “non-glyCAF” in culture?-- Since they do expand these cells in vitro (page 9, for example in fig 4). Also vice versa, can non-glyCAF gain properties of glyCAF?

We did not observe any spontaneous conversion from glyCAFs to non-glyCAFs (and vice versa) during a short period of cell culture (~3 passages). However, we did note that glyCAFs tend to lose their glycolytic signature over an extended period in culture at 12 passages (Fig. R18). While we observe that the differential expression of *Glut1* in glyCAF is lost in later passages, expression of glycolytic enzyme *Pgk1* is maintained, and expression of *Nt5e* (encoding CD73) was enriched in the non-glyCAF after 12 passages. Consequently, all transwell migration and *in vitro* experiments were conducted using CAFs that were only briefly expanded in culture.

Figure R18. Gene expression analysis of the glyCAF signature in early (P3) and late passage (P12) glyCAF and non-glyCAF

Additionally, in response to a query from Reviewer #1, we investigated whether the accumulation of glyCAFs is influenced by TGFβ, a signaling pathway known to upregulate glycolytic machinery such as glucose transporter (GLUT1) and Hexokinase 2 in various cell types^{4,5}. Bioinformatic analysis of sarcoma-associated CAFs indicated that glyCAFs may experience higher TGFβ stimulation compared to other CAF subtypes (Fig. R19A,B), which may promote a preferential glycolysis in these cells.

Figure R19. A, Dotplot showing the genes identified in the Pan-Fibroblast TGFβ Response signature (F-TBRS, Marthasian et al.) (left) and TGFβ pathway genes (right) projected across 4 CAF subsets. **B,** Violin plot of the aggregated TGFβ pathway and F-TBRS signature.

These results prompted us to assess the impact of TGFβ on CAFs *in vitro*. Interestingly, we observed that treating non-glyCAFs with recombinant TGFβ induces the expression of *Nt5e* (CD73) and *Glut1* (Figure R20A). On the contrary, in speculative experiments, treating glyCAFs with anti-TGFβ reduces the expression of *Nt5e*, *Glut1*, and *Pgk1* (Figure R20B), suggesting that TGFβ signaling might be responsible for the induction of CAFs glycolytic properties.

While these findings are intriguing and potentially relevant to understanding how glyCAFs accumulate in Ccne1+ tumors compared to Vgll3+ tumors, we consider them preliminary at this stage, and we think that further investigations are warranted. Consequently, we have not included these data in the revised manuscript but are willing to do so upon the reviewer's request.

Figure R20: A, RT-qPCR of the glycolytic signature in non-glyCAF treated with recombinant TGFβ (5ng/mL) for 48h. **B**, RT-qPCR of the glycolytic signature in glyCAF treated with anti-IgG or TGFβ neutralizing antibody (30ng/mL) for 48h.

9. Besides preventing migration of the T cells to the tumor cells in the transwell, are the glyCAFs altering other phenotypic functions of the T cells? Viability, enzymatic release, etc...?

We thank the reviewer for bringing up this important point. We now include an analysis of these key phenotypic functions of the CD8+ T cells following co-culture with glyCAF and non-glyCAF. We found no significant difference in the viability and proportion of Granzyme B+ CD8+ T cells in co-cultures compared to monoculture (Fig. R21). Interestingly, we observe an increase of Perforin+ CD8+ cells when cultured with glyCAF or non-glyCAF. These data indicate that at least *in vitro*, glyCAF primarily impair the migratory abilities of CD8+ T cells (Fig. 5c,i), while viability and cytolytic capacity of the CD8+ T cells are unaffected and, in some cases, even enhanced.

Figure R21: Analysis of viability, Granzyme B and Perforin expression in activated CD8+ T cells co-cultured with CAF for 24h.

10. Bottom of page 10: When describing the predicted cell-cell communications, it is unclear what the authors are concluding re: T cell activation and properties. They discuss several possibilities, including CXCL16-CXCR6 interactions between the glyCAFs and T cells, but nothing is definitively proven. Need to further define/prove their statements. Also, for Fig 4D: Seems to be redundancy in signaling cell-cell communications as both glyCAF and apCAF have similar intensity signaling networks, while mCAF and iCAF also do but at lower intensity. Since these cell-cell networks are not unique to glyCAF why would they explain the noted changes in CD8 cell infiltration? Also, upon inhibition of the glyCAF with Glut1, how do the signaling networks change for the other CAFs?

We thank the reviewer for bringing our attention to this critical component of our receptor-ligand communication analysis. In this respect, we also want to highlight that since our initial submission, we have further investigated the molecular mechanisms used by the glyCAFs to block T cells migration into the tumor parenchyma and focused specifically on the CXCL16- CXCR6 signaling axis between glyCAF and CD8+ T

cells. We now show additional evidence that CXCL16 expressing glyCAF serves as a mechanism to retain CD8⁺ T cells in the tumor stroma in Figure 5 of the revised manuscript. Notably, glyCAF are particularly enriched for expression of CXCL16 (**Fig. 5f**). Knockdown of either *Cxcl16* or *Glut1* or GLUT1 inhibition is successful in abrogating CXCL16 expression in glyCAF (**Fig. 5h and Extended Data 5f**), and as a result can rescue the defects in CD8⁺ T cell migration in transwell (**Fig. 5i**). In the same line, *Cxcr6*^{-/-} T cells exhibit increased migration toward tumor cells relative to *Cxcr6*^{+/+} wildtype CD8⁺ T cells in the presence of glyCAF (**Fig. 5m,n**). We further corroborated this result *in vivo* with adoptive transfer experiments, in which we observed increased infiltration of *Cxcr6* deficient T cells in the tumor parenchyma relative to wildtype T cells (**Fig. 5o-q**).

Additionally, in respect to the cell-cell communication analysis, we now include other TME cellular components, such as CD4⁺ T cells (Treg, Th1, Naïve), B cells, NK-cells, and myeloid lineages (Granulocyte, MonoMacDC, and pDC) as well as Tumor and endothelial cells in our receptor-ligand analysis (**Fig. 5e**).

First, among the CAF types, apCAFs and glyCAFs showed the highest probability of interaction with the CD8⁺ T cells via the *Cxcl16*/*Cxcr6* axis. However, while the expression of *Cxcl16* mRNA is similar between glyCAFs and apCAFs, post-transcriptional regulations of the RNA are differentially occurring in these cells, with *Cxcl16* protein being higher expressed in the glyCAFs (**Fig. 5f**). These data suggest that the glyCAFs are, among the CAFs, more likely to interact with the CD8⁺ T cells through the *Cxcl16*/*Cxcr6* axis. It is possible that the glycolytic profile of the CAFs is somehow responsible for the *Cxcl16* protein expression, and accordingly, we found changes in the *Cxcl16* expression upon GLUT1 modulation in the glyCAFs (**Fig. 5h,j and Extended Data Fig. 5g**), but not in other cells such as myeloid and lymphoid cells (**Fig. 5k,l**).

When we included other TME cells in the receptor-ligand communication analysis, we observe that the MonoMacDC cluster, in addition to CAFs, is also likely to interact with effector and exhausted CD8⁺ T cells through the CXCL16-CXCR6 signaling axis, suggesting the possibility that myeloid cells, likewise the CAFs, could block CD8 T cells infiltration in the tumor parenchyma. Nevertheless, the following observations minimize the possibility that myeloid cells are involved in the observed phenotype:

- 1- Increased T cells infiltration in the tumor parenchyma upon *Glut1* inhibition correlated to decreased *Cxcl16* expression in the CAFs (**Fig. 5j**) but not in the myeloid cells (**Fig. 5l**). Proportions of CXCL16 expressing glyCAF are also significantly diminished following GLUT1i *in vivo* (**Fig. 4b-d**). Unlike the CAF, knockdown of *Glut1* in macrophages does not regulate the expression of *Cxcl16* (**Extended Data Fig. 5h**) and does not alter the T cells migration index *in vitro* (**Extended Data Fig. 5i**). Additionally, we also did not note changes of the overall proportions of TAM subsets upon GLUT1i (**Extended Data Fig. 4f**).
- 2- We show that CD90⁺ CD73⁺ glyCAF are closer in proximity to CD8⁺ T cells than CSF1R⁺ myeloid cells and CD90⁺ CD73⁻ non-glyCAF (which encompasses the populations of iCAF, mCAF, and apCAF). (**Fig. 4m**), suggesting that glyCAF have a higher likelihood of communicating via *Cxcl16* to CD8⁺ T cells. This new evidence favors the hypothesis that the chemoattractant and adhesion properties of *Cxcl16*⁺ glyCAF retain CD8⁺ T cells at the glyCAF rich tumor margin.

11. Fig 5i: Does *Glut1* inhibition alone increase T cell infiltration into tumor? The authors show CD8 cell staining for DOX and then DOX+*Glut1* inh, but not *Glut1* alone. Please show.

We guide the reviewer to revisit **Fig. 4g-i** in which we show GLUT1i alone increases the CD8⁺ T cell infiltration into the tumor mass.

12. Fig 5i: When were these tumors taken? At end of study? As the tumor is growing/becoming resistant to the combo therapy (day 34 or so), do the T cells revert to being at the margin again?

We thank the reviewer for pointing our attention to addressing durability of the combination of Doxorubicin and GLUT1i. The immunofluorescence images shown in Fig. 5i (**Fig. 6i** of the revised manuscript) were taken from tumors at Day 26 when the combination therapy (DOX+GLUT1i) was maximally effective. We performed immunofluorescence analysis of CD8+ T cells from Day 34 at which the combination treated tumors start to regrow. At Day 34, we observe a decreased presence of CD8+ cells at the tumor margin in combo treated tumors (**Fig. R22A**). Importantly, at Day 34 we do not observe a significant reduction in the number of CD8+ cells infiltrating the tumor parenchyma, suggesting that that the T cells do not completely revert to the tumor margin at D34 (**Fig. R22B**). However, we noticed a slight downward trend at D34 compared to D27 ($p= 0.1330$), which may suggest that at later timepoints the T cells would further diminish from the parenchyma.

Even if the CD8+ T cell numbers remain consistent during the regrowth period, it is possible that in the absence of sustained chemotherapy and GLUT1 inhibition, cytotoxic T cells lose their effector function or become exhausted, although this remains to be tested.

Figure R22: Quantification of CD8+ cells by immunofluorescence in the **A**, Tumor Margin and **B**, Parenchyma from DOX or D+G (DOX+GLUT1i) treated tumors at Day 26 or Day 34.

13. Prove the role of Doxorubicin in their models. Does Doxo only treatment increase antigen presentation in vivo model employed?

Anthracycline based chemotherapies including doxorubicin are well known inducers of immunogenic cell death^{9,23}. Dendritic cells and CD8+ T cells are the critical immune cellular components that promote chemotherapy responses. The critical antigen presenting cells are CD11b⁺ CD11c⁺ Ly6c^{hi} cells that accumulate intratumorally following anthracycline-based chemotherapy and promote chemotherapy responses through increased antigen presentation to CD8+ T cells^{10,11}. In our models, we assessed the presence of the antigen presenting CD11b⁺ CD11c⁺ Ly6c^{hi} cells and show that this subset indeed increases in the tumor following DOX, in both Ccne1⁺ and Vgll3⁺ models, thus proving that DOX is functional in our sarcoma models (**Fig. R23**).

Figure R23: Proportions of CD11b⁺ CD11c⁺ Ly6c^{hi} antigen presenting cells in Ccne1⁺ or Vgll3⁺ tumors following DOX treatment.

14. In discussion mention acidification of TME due to glyCAFs. Is there any evidence for this in vivo? Can you stain for CA9/12 on the tumors? Also why do the authors think acidification only affects migration. What about viability, other properties for T cells? Please discuss.

In the initial submission of this manuscript, we had speculated on mechanisms of glyCAF role in the exclusion of T cells and had proposed acidification as a one potential mechanism which was primarily based on published data^{24,25}. Since our initial submission, we have performed more thorough investigation and have pinpointed the CXCL16-CXCR6 axis as the mechanism of glyCAF mediated immune exclusion which we now corroborate in **Fig.5** of the revised manuscript. Although we have not directly ruled out that acidification of the TME from CAF glycolytic byproducts may contribute to this effect, we provide convincing evidence to support the hypothesis that glyCAF expression of CXCL16+ glyCAF positions CXCR6+CD8+ T cells at the tumor margin.

Reviewer #5 (Remarks to the Author): with expertise in sarcomas

Broz et al have submitted a compelling manuscript that argues the existence of sarcoma associated fibroblasts akin to those that exist in epithelial tumors known as cancer associated fibroblasts. This concept is novel and potentially very interesting. The authors have also identified a subset of fibroblasts that govern T cell recruitment to the tumor parenchyma and subsequent responsiveness to chemotherapy. However there are some textual and experimental/mechanistic clarifications necessary before the manuscript would be ready for publication. The following is a figure by figure analysis of the primary manuscript:

We express our gratitude to the reviewer for providing valuable and constructive suggestions that significantly enhanced the robustness of our study's findings. In response to the insightful recommendations made by the reviewers, we performed additional experiments, which have played a crucial role in elucidating the molecular mechanisms through which glyCAFs mediate the exclusion of T cells from the inner tumor parenchyma.

Figure 1: The models. The authors have generated a new system for syngeneic transplantation of transformed and fluorescently labeled mesenchymal stem cells into host mice. Aside from one H&E imaging the extended figures there is no extensive validation that the models do indeed recapitulate either existing murine models of undifferentiated pleomorphic sarcoma (UPS) or human UPS. Ideally the authors would have certified veterinary and human clinical pathology specialists evaluate the histological images and provide classification. This requirement is typical of any new model system.

Moreover, the authors should provide comparative transcriptomics of their models relative to human and mouse UPS as can be found in this reference (Cross species genomic analysis identifies a mouse model as undifferentiated pleomorphic sarcoma/malignant fibrous histiocytoma by Mito et al 2009). It's critical that the authors demonstrate that their models are in fact representative of UPS before drawing additional conclusions.

We thank the reviewer for point out the need of further clarification about the sarcoma mouse model. The methodology used for generating sarcoma models from mesenchymal stem cells is not new to this manuscript, but previously published by our group^{14,26}. To supplement this, we now include pathologist grading of the p53^{KO}Ccne1⁺ and p53^{KO}Vgll3⁺ models. Pathologist classification of Ccne1⁺ and Vgll3⁺ tumors are consistent with human High Grade Undifferentiated Sarcoma with Spindle Cell morphology which are usually diagnosed by exclusion of other differentiated sarcoma sub-types.

We thank the reviewer for directing us to the publication by Mito et al.²⁷, which in Table S5 finds a set of 10 genes that were enriched in the LSL-Kras(G12D); Trp53(Flox/Flox) mouse model of soft tissue sarcoma relative to normal tissue and also enriched in patient with UPS/MFH but no other STS subtypes.

As a parallel to that experiment, we analyzed the expression of this gene signature in malignant (dsRed+) vs non-malignant cells from our mouse single-cell dataset of **Fig. 2a**. The complete table of significant differential expression is presented below.

p_val	avg_log2_FC	pct.1	pct.2	p_val_adj	cluster	gene
0	0.3865229	0.384	0.080	0.00e+00	Tumor	Bcat1
0	0.7700363	0.267	0.054	0.00e+00	Tumor	Cenpe
0	0.4111160	0.248	0.062	0.00e+00	Tumor	Ccnb2
0	0.3863646	0.183	0.033	0.00e+00	Tumor	Ccnb1
0	0.1988423	0.146	0.022	0.00e+00	Tumor	Foxm1
0	0.1912663	0.141	0.021	0.00e+00	Tumor	Melk
0	0.2705522	0.367	0.533	1.68e-05	Tumor	Cenpa
0	0.7814572	0.239	0.020	0.00e+00	Nonmalignant	Lpxn

Eight of 10 genes were upregulated in the tumor cells of our mouse models vs non-malignant cells, thus validating our sarcoma models. These genes included known oncogenes and cell cycle regulators; importantly, it also included *Foxm1*, a component of the Ras pathway that Mito et al. found to be enriched in human UPS/MFH compared to other types of STS.

The genes not upregulated by cancer cells were *Fcgr1* (not significantly DE) and *Lpxn* (higher in non-malignant vs dsRed+ UPS), which indeed should not be enriched in our pure tumor-cell population since these genes are mostly expressed in immune and hematopoietic cells. In Mito et al, the up-regulation of *Fcgr1* and *Lpxn* was found in whole (unpurified) tumor vs normal tissue and would suggest greater immune-cell content in UPS vs normal tissue; this last implication was not tested in our data.

Figure 2. Validation of CAFs in sarcoma. The authors provide extensive scRNA-seq gene expression data, as well as limited flow and imaging data and conclude that a small population of CAFs exist in at least some sarcomas. As this claim is quite substantial and important functional validation is necessary. Using the FACS sorted CAFs vs other tumor cell populations the authors should perform collagen contraction assays. These well established assays confirm functional fibroblast activity.

We acknowledge the reviewer's suggestion regarding the functional validation of fibroblast activity to further confirm the identity of cancer-associated fibroblasts (CAFs). However, executing such assays for human CAFs presents challenges due to the limited availability of fresh human sarcoma tissue, which is essential for these experiments. In our study, we have collected a range of human samples (flash-frozen or FFPE) and conducted extensive gene expression analyses, establishing that the Thy1-expressing CAF population expresses well-established CAF markers such as RARRES2^{28,29} and LUM^{30,31} in addition to other fibroblast markers (see **Fig. 2d,e** of the revised manuscript).

To enhance the robustness of our findings, we have incorporated immunohistochemical (IHC) staining of CD90 in primary leiomyosarcoma (LMS) (**Fig. 2f**), confirming that CD90 is restricted to a subset of cells with a spindle-shaped morphology that we identify as CAFs. Acknowledging the limitations of fixed samples for sorting CD90+ CAFs and CD90- tumor fractions for functional assays, we recognize this as a potential constraint of our study. However, we believe our work can serve as a foundation for future investigations and the identification of CAFs in sarcomas from fresh tissue, enabling the proposed functional fibroblast experiments.

To affirm that isolated CAFs represent a distinct cellular population from tumor cells and do not possess tumorigenic potential, we implanted mice with scaffolds bearing *Ccne1*⁺ tumor cells or CD90+ CAFs (**Extended Data Fig. 5b**) Tumor growth occurred only in scaffolds bearing *Ccne1*⁺ tumor cells, not in those with CAFs, underscoring that the CD90+ CAF population is indeed a distinct fibroblast population separate from tumor cells.

Figure 3: Glut1 inhibition recruits T cells to tumor parenchyma. The authors show that Glut1 knockdown in tumor cells has no effect on T cell recruitment whereas Glut1 inhibition in tumors leads to increased T cell recruitment in the parenchyma. However, UPS tumors are comprised of many immune cells including macrophages etc as well as tumor cells. The authors do not rule out the effect of Gluti on these cells when concluding that CAFs are responsible for T cell recruitment. This is not sufficient. Large macrophage/dendritic populations should be ruled out in addition to tumor cells before drawing this conclusion.

We acknowledge that while we observe the primary effect of the GLUT1 inhibitor on reprogramming the glyCAF, other cells of the tumor microenvironment may be impacted. Firstly, we showed that although tumor cells glycolysis is impaired under GLUT1i (**Extended Data Fig. 4g**), tumor cells short-hairpin RNA targeting of Glut1 (shGLUT1) does not recapitulate the increase in T cell infiltration we observe under GLUT1i (**Fig. 4j,k and Extended Data Fig. 4h**), suggesting that other cellular components of the tumor microenvironment are responsible. Certain immune cells, namely T-lymphocytes and certain subsets of tumor associated

macrophages (TAMs) cells utilize glycolysis for their metabolism^{12,13} and express the glycolytic signature, which we acknowledge is not specific to glyCAF. As such, we now have characterized the immune microenvironment (CD45+ fraction) following GLUT1i to confirm that glyCAF are the primary cellular target of GLUT1i.

We now include a projection of the glycolysis signature (*Slc2a1, Pkm, Pgk1, Pgk1, Eno1, Ldha, Ldha, Hk1, Aldoa*) in the myeloid cells (monocytes/macrophages) and lymphocytes in the tumor microenvironment under GLUT1i (**Extended Data Fig. 4b,c** of the revised manuscript). Interestingly, while we observe a decrease in the glycolysis signature in glyCAF (**Fig. 4d,f and Extended Data Fig. 4a**), we do not observe a reduction in glycolysis pathway transcripts in other myeloid (**Extended Data Fig. 4b**) or lymphoid cells (**Extended Data Fig. 4c**), suggesting that glyCAF are the primary target of GLUT1i. It is unclear why immune cells are not particularly affected by GLUT1i while glyCAF are. We suspect that glycolysis may be particularly essential to certain myeloid and lymphoid lineages, and thus upon GLUT1i these cells utilize compensatory mechanisms, such as the increase of other glucose transporters (GLUT3, GLUT4) to sustain glucose import without impairing glycolysis, although this warrants further exploration.

We next focused our attention on investigating the effects of GLUT1i on tumor associated macrophages (TAMs), which are the dominant immune cell in Ccne1+ tumors. In a previous study, we have shown that Ccne1+ tumors are infiltrated by TAMs exist in several states within the tumor mass, namely MΦ Thbs1, MΦ C1qa, MΦ Spp1, and Mono/MΦ Ifi¹⁴ (**Extended Data Fig. 4d,e**). Therefore, we investigated if GLUT1i alters the proportions of these subtypes within the tumor mass. In the revised manuscript, we show that despite a trending increase of Spp1+ MO in GLUT1i treated tumors, which was largely driven by two outliers, the TAM heterogeneity is comparable between the two conditions (**Extended Data Fig. 4f**), further suggesting that cells other than the glyCAF are not particularly affected by the GLUT1i.

Importantly, in addition to characterizing the effect of GLUT1 inhibitor on the TME cells, we also want to highlight that since our initial submission, we have further investigated the molecular mechanisms used by the glyCAFs to block T cells migration into the tumor parenchyma, and further studied the role of the CXCL16- CXCR6 signaling axis between glyCAF and CD8+ T cells. We now show additional evidence that CXCL16 expressing glyCAF serves as a mechanism to retain CD8+ T cells in the tumor stroma in Figure 5 of the revised manuscript. Notably, glyCAF are particularly enriched for expression of CXCL16 (**Fig. 5f**). Knockdown of either *Cxcl16, Glut11* or GLUT1 inhibition is successful in abrogating CXCL16 expression in glyCAF (**Extended Data 5h**), and as a result can rescue the defects in CD8+ T cell migration in transwell (**Fig. 5i**). In the same line, *Cxcr6*^{-/-} T cells exhibit increased migration toward tumor cells relative to *Cxcr6*^{+/+} wildtype CD8+ T cells in the presence of glyCAF (**Fig. 5m,n**). We further corroborated this result in vivo with adoptive transfer experiments, in which we observed increased infiltration of *Cxcr6* deficient T cells in the tumor parenchyma relative to wildtype T cells (**Fig. 5o-q**).

Figure 5: Chemotherapy in UPS. The authors should correct text that indicates chemotherapy is standard first line treatment in UPS. This is often not the case. First line Radiation and surgery are primarily used to treat UPS with adjuvant chemotherapy sometimes being added in advanced disease with mixed results. As the authors have effectively demonstrated chemotherapy including doxorubicin is ineffective as first line therapy against UPS. The authors conclusions are important in spite of this error in rationale. The ability to sensitize UPS cells to chemotherapy using Gluti is an important discovery but should be framed in the text as such.

We apologize for this error. We have updated the text on **Page 3-4** the revised manuscript to highlight the distinction that chemotherapy is not generally first-line treatment, however it is occasionally used in high-grade STS tumors, although the clinical benefit of adjuvant/neo-adjuvant chemotherapy has been disputed^{35,36}.

References

1. Carbonetto, P. *et al.* GoM DE: interpreting structure in sequence count data with differential expression analysis allowing for grades of membership. *Genome Biology* **24**, 236 (2023).
2. Carbonetto, P., Sarkar, A., Wang, Z. & Stephens, M. Non-negative matrix factorization algorithms greatly improve topic model fits. Preprint at <https://doi.org/10.48550/arXiv.2105.13440> (2022).
3. Wei, J. *et al.* Characterization of Glycolysis-Associated Molecules in the Tumor Microenvironment Revealed by Pan-Cancer Tissues and Lung Cancer Single Cell Data. *Cancers* **12**, 1788 (2020).
4. Yin, X. *et al.* Hexokinase 2 couples glycolysis with the profibrotic actions of TGF- β . *Science Signaling* **12**, eaax4067 (2019).
5. Kitagawa, T., Masumi, A. & Akamatsu, Y. Transforming growth factor-beta 1 stimulates glucose uptake and the expression of glucose transporter mRNA in quiescent Swiss mouse 3T3 cells. *Journal of Biological Chemistry* **266**, 18066–18071 (1991).
6. Cords, L. *et al.* Cancer-associated fibroblast classification in single-cell and spatial proteomics data. *Nat Commun* **14**, 4294 (2023).
7. Petitprez, F. *et al.* B cells are associated with survival and immunotherapy response in sarcoma. *Nature* **577**, 556–560 (2020).
8. Hui, J. Y. C. Epidemiology and Etiology of Sarcomas. *Surg Clin North Am* **96**, 901–914 (2016).
9. Casares, N. *et al.* Caspase-dependent immunogenicity of doxorubicin-induced tumor cell death. *Journal of Experimental Medicine* **202**, 1691–1701 (2005).
10. Ma, Y. *et al.* Anticancer Chemotherapy-Induced Intratumoral Recruitment and Differentiation of Antigen-Presenting Cells. *Immunity* **38**, 729–741 (2013).
11. Ma, Y. *et al.* CCL2/CCR2-Dependent Recruitment of Functional Antigen-Presenting Cells into Tumors upon Chemotherapy. *Cancer Research* **74**, 436–445 (2014).
12. Puthenveetil, A. & Dubey, S. Metabolic reprogramming of tumor-associated macrophages. *Ann Transl Med* **8**, 1030 (2020).
13. van der Windt, G. J. W. & Pearce, E. L. Metabolic switching and fuel choice during T-cell differentiation and memory development. *Immunol Rev* **249**, 27–42 (2012).
14. Tessaro, F. H. G. *et al.* Single-cell RNA-seq of a soft-tissue sarcoma model reveals the critical role of tumor-expressed MIF in shaping macrophage heterogeneity. *Cell Reports* **39**, 110977 (2022).
15. Yang, J. *et al.* Glycolysis reprogramming in cancer-associated fibroblasts promotes the growth of oral cancer through the lncRNA H19/miR-675-5p/PFKFB3 signaling pathway. *Int J Oral Sci* **13**, 1–11 (2021).
16. Becker, L. M. *et al.* Epigenetic Reprogramming of Cancer-Associated Fibroblasts Deregulates Glucose Metabolism and Facilitates Progression of Breast Cancer. *Cell Reports* **31**, 107701 (2020).
17. Zhang, D. *et al.* Metabolic Reprogramming of Cancer-Associated Fibroblasts by IDH3 α Downregulation. *Cell Reports* **10**, 1335–1348 (2015).
18. Jerby-Arnon, L. *et al.* Opposing immune and genetic mechanisms shape oncogenic programs in synovial sarcoma. *Nat Med* **27**, 289–300 (2021).
19. Jerby-Arnon, L. *et al.* A Cancer Cell Program Promotes T Cell Exclusion and Resistance to Checkpoint Blockade. *Cell* **175**, 984–997.e24 (2018).
20. Goel, S. *et al.* CDK4/6 inhibition triggers anti-tumour immunity. *Nature* **548**, 471–475 (2017).
21. Schaer, D. A. *et al.* The CDK4/6 Inhibitor Abemaciclib Induces a T Cell Inflamed Tumor Microenvironment and Enhances the Efficacy of PD-L1 Checkpoint Blockade. *Cell Reports* **22**, 2978–2994 (2018).
22. van Wagenveld, L. *et al.* Homologous Recombination Deficiency and Cyclin E1 Amplification Are Correlated with Immune Cell Infiltration and Survival in High-Grade Serous Ovarian Cancer. *Cancers (Basel)* **14**, 5965 (2022).
23. Krysko, D. V. *et al.* Immunogenic cell death and DAMPs in cancer therapy. *Nat Rev Cancer* **12**, 860–875 (2012).
24. Pilon-Thomas, S. *et al.* Neutralization of tumor acidity improves anti-tumor responses to immunotherapies. *Cancer Res* **76**, 1381–1390 (2016).
25. Yin, L. *et al.* CA9-Related Acidic Microenvironment Mediates CD8⁺ T Cell Related Immunosuppression in Pancreatic Cancer. *Front Oncol* **11**, 832315 (2022).

26. Guarnerio, J. *et al.* A Genetic Platform to Model Sarcomagenesis from Primary Adult Mesenchymal Stem Cells. *Cancer Discov* **5**, 396–409 (2015).
27. Mito, J. K. *et al.* Cross Species Genomic Analysis Identifies a Mouse Model as Undifferentiated Pleomorphic Sarcoma/Malignant Fibrous Histiocytoma. *PLOS ONE* **4**, e8075 (2009).
28. Mayer, S. *et al.* The tumor microenvironment shows a hierarchy of cell-cell interactions dominated by fibroblasts. *Nat Commun* **14**, 5810 (2023).
29. Zhai, X. *et al.* Identification of the novel therapeutic targets and biomarkers associated of prostate cancer with cancer-associated fibroblasts (CAFs). *Frontiers in Oncology* **13**, (2023).
30. Wang, X. *et al.* Cancer-associated fibroblast-derived Lumican promotes gastric cancer progression via the integrin β 1-FAK signaling pathway. *International Journal of Cancer* **141**, 998–1010 (2017).
31. Yamauchi, N. *et al.* Stromal expression of cancer-associated fibroblast-related molecules, versican and lumican, is strongly associated with worse relapse-free and overall survival times in patients with esophageal squamous cell carcinoma. *Oncol Lett* **21**, 445 (2021).
32. Costa, A. *et al.* Fibroblast Heterogeneity and Immunosuppressive Environment in Human Breast Cancer. *Cancer Cell* **33**, 463-479.e10 (2018).
33. Kraman, M. *et al.* Suppression of Antitumor Immunity by Stromal Cells Expressing Fibroblast Activation Protein- α . *Science* **330**, 827–830 (2010).
34. Li, H. *et al.* Reference component analysis of single-cell transcriptomes elucidates cellular heterogeneity in human colorectal tumors. *Nat Genet* **49**, 708–718 (2017).
35. Pasquali, S. & Gronchi, A. Neoadjuvant chemotherapy in soft tissue sarcomas: latest evidence and clinical implications. *Ther Adv Med Oncol* **9**, 415–429 (2017).
36. Saponara, M., Stacchiotti, S., Casali, P. G. & Gronchi, A. (Neo)adjuvant treatment in localised soft tissue sarcoma: The unsolved affair. *European Journal of Cancer* **70**, 1–11 (2017).

REVIEWERS' COMMENTS

Reviewer #1 (Remarks to the Author):

The authors have adequately answered my concerns. I do not have additional questions. However, it would be informative to the readers if the authors can acknowledge the role of Tgfb signaling in differential accumulation of mCAFs and glyCAFs - at the very least in the discussion.

Reviewer #3 (Remarks to the Author):

In this revised study, Broz et al., have addressed the role of tumor CAFs in regulating chemotherapy through immune infiltrating lymphocytes specially cytotoxic CD8+ T cells. They show that GLUT1i specifically targets GlyCAFs and not the transcriptome of lymphoid or myeloid cells. My comments have been addressed and a minor comment remains:

1. In Fig 5, Authors are recommended to share representative images of CD8+ T cells migrating through 3um transmembrane pores.

Reviewer #4 (Remarks to the Author):

The authors have attempted to address many of the significant critiques. While not all of their subsequent data support the full conclusions regarding the prominent role of glyCAFs towards suppressing anti-tumor immune activity, their manuscript adds significant insights into CAF profiling and biological effects of their metabolic properties.

Reviewer #5 (Remarks to the Author):

I find the authors have made substantial efforts to respond appropriately to most of the critiques. However, they indicate that they cannot perform functional studies on the newly identified sarcoma associated fibroblasts because they are difficult to obtain from human samples. This response is reasonable, however i would suggest they use their mouse models to isolate the fibroblast population under investigation and perform functional assays of

fibroblast contraction. The authors have instead transplanted the fibroblast population into mice and seen that no tumors develop. This result is certainly valid but is at best a negative control. The underlying question is can the authors draw the conclusions they have about this population of cells without positive functional validation. T cell studies require co-culture killing assays and macrophage studies require phagocytosis or cytokine production elements. In my opinion Fibroblast identification studies should have some similar validation. However, I shall leave it to the editor to determine their requirements for the journal.

Reviewer #1 (Remarks to the Author):

The authors have adequately answered my concerns. I do not have additional questions. However, it would be informative to the readers if the authors can acknowledge the role of Tgfb signaling in differential accumulation of mCAFs and glyCAFs - at the very least in the discussion.

We agree with the reviewer that the discussion of TGFb signaling in glyCAF accumulation will be essential for further studies. As such, we have included this in the discussion.

Reviewer #3 (Remarks to the Author):

In this revised study, Broz et al., have addressed the role of tumor CAFs in regulating chemotherapy through immune infiltrating lymphocytes specially cytotoxic CD8+ T cells. They show that GLUT1i specifically targets GlyCAFs and not the transcriptome of lymphoid or myeloid cells. My comments have been addressed and a minor comment remains:

1. In Fig 5, Authors are recommended to share representative images of CD8+ T cells migrating through 3um transmembrane pores.

As recommended by Reviewer #3, below we show representative images of migrated CD8+ T cells that have migrated through the 3um pores into the bottom chamber of the transwell culture system containing Ccne1+ tumor cells. White arrows indicate CD8+ T cells and the cell-type labels correspond to the type of cells forming the monolayer in the top of the transwell.

Reviewer #4 (Remarks to the Author):

The authors have attempted to address many of the significant critiques. While not all of their subsequent data support the full conclusions regarding the prominent role of glyCAFs towards suppressing anti-tumor immune activity, their manuscript adds significant insights into CAF profiling and biological effects of their metabolic properties.

We thank the reviewer for the critical suggestions that helped strengthen our study's conclusions.

Reviewer #5 (Remarks to the Author):

I find the authors have made substantial efforts to respond appropriately to most of the critiques. However, they indicate that they cannot perform functional studies on the newly identified sarcoma associated fibroblasts because they are difficult to obtain from human samples. This response is reasonable, however i would suggest they use their mouse models to isolate the fibroblast population under investigation and perform functional assays of fibroblast contraction. The authors have instead transplanted the fibroblast population into mice and seen that no tumors develop. This result is certainly valid but is at best a negative control. The underlying question is can the authors draw the conclusions they have about this population of cells without positive functional validation. T cell studies require co-culture killing assays and macrophage studies require phagocytosis or cytokine production elements. In my opinion Fibroblast identification studies should have some similar validation. However, I shall leave it to the editor to determine their requirements for the journal.

We acknowledge that reviewer #5 has expressed a desire for functional data on Cancer-Associated Fibroblasts (CAFs). We agree that performing functional assays is crucial for comprehending the role of these cells in tumor growth and progression. However, we hold the view that the proposed contraction assays for CAFs would not yield substantial information and would not conclusively determine the functional role of these cells, at least in the context of our study.

Historically, contraction assays for CAFs served two main purposes: i) as an indicator of the tumor-activated nature of CAFs with matrix remodeling capability, and ii) as a proxy for fibroblasts or epithelial cells undergoing epithelial-to-mesenchymal transition in carcinomas, gauging the ability of fibroblasts (or EMT cells) to contract to a greater extent than tumor cells (epithelial cells).

Mechanistically, this work focuses on the Cxcl16/Cxcr6 interaction between glyCAFs and T cells that is responsible for blocking T cells at the tumor margin. Thus, we do not believe that probing CAFs' contraction capabilities would be particularly relevant to the current work, since contractile function is not related to the mechanism in this study. The assay would merely be performed as proof that the cells identified as CAFs are indeed CAFs. However, the advent of scRNAseq and the ability to comprehensively characterize cell expression profiles has, in our opinion, diminished the necessity for contraction assays, as CAFs can now be unequivocally identified based on the expression of unique genes. Consequently, many papers on CAFs published in the last five years rely on scRNAseq analysis, omitting contraction assays (examples listed below).

- LRRC15+ myofibroblasts dictate the stromal setpoint to suppress tumor immunity (Nature, 2022).
- Mesothelial cell-derived antigen-presenting cancer-associated fibroblasts induce the expansion of regulatory T cells in pancreatic cancer (Cancer Cell, 2022).

- Functionally distinct cancer-associated fibroblast subpopulations establish a tumor-promoting environment in squamous cell carcinoma (Nature Communications, 2023).
- TGF β -blockade uncovers stromal plasticity in tumors by revealing the existence of a subset of interferon-licensed fibroblasts (Nature Communications, 2020).
- Spatial Positioning and Matrix Programs of Cancer-Associated Fibroblasts Promote T-cell Exclusion in Human Lung Tumor (Cancer Discovery, 2022).

Furthermore, in the context of sarcoma, we believe that the contraction assay is even less informative when comparing CAFs and tumor cells. Given that sarcoma cells are also mesenchymal cells, like CAFs, they may possess a high contraction capability, at least compared to epithelial cells. Conducting contraction assays for both CAFs and sarcoma cells may result in similar outcomes, making it challenging to draw meaningful conclusions.

While we acknowledge the importance of functional assays, we think that the field should progress towards defining critical genes for CAF subtype manipulation and functionally profiling their activities in vivo through the generation of knock-out or transgenic mice.